# Diffusion Models and the Manifold Hypothesis: Log-Domain Smoothing is Geometry Adaptive

**Tyler Farghly** *
Department of Statistics
University of Oxford

**Peter Potaptchik** *
Department of Statistics
University of Oxford

**Samuel Howard** *
Department of Statistics
University of Oxford

**George Deligiannidis**
Department of Statistics
University of Oxford

**Jakiw Pidstrigach**
Department of Statistics
University of Oxford

## Abstract

Diffusion models have achieved state-of-the-art performance, demonstrating remarkable generalisation capabilities across diverse domains. However, the mechanisms underpinning these strong capabilities remain only partially understood. A leading conjecture, based on the manifold hypothesis, attributes this success to their ability to adapt to low-dimensional geometric structure within the data. This work provides evidence for this conjecture, focusing on how such phenomena could result from the formulation of the learning problem through score matching. We inspect the role of implicit regularisation by investigating the effect of smoothing minimisers of the empirical score matching objective. Our theoretical and empirical results confirm that smoothing the score function—or equivalently, smoothing in the log-density domain—produces smoothing tangential to the data manifold. In addition, we show that the manifold along which the diffusion model generalises can be controlled by choosing an appropriate smoothing.

## 1 Introduction: Diffusion, manifolds and generalisation

Diffusion models (Sohl-Dickstein et al., 2015; Song and Ermon, 2019; Ho et al., 2020; Song et al., 2021) have emerged as a powerful class of generative models, achieving state-of-the-art performance across domains (Dhariwal and Nichol, 2021; Kong et al., 2021; Liu et al., 2023; Ho et al., 2022). Beyond their ability to generate high-quality outputs, they are also capable of producing novel samples not present in the training data, indicating a surprising capacity for generalisation.

The goal of diffusion models is to produce samples from a target distribution $\mu_{\text{data}}$ on $\mathbb{R}^d$, given only a finite number of samples. They do this by learning to *reverse* a noising process, $X_t$, which begins with a random sample of the data distribution $X_0 \sim \mu_{\text{data}}$ and gradually transforms it into noise. This process is defined by the stochastic differential equation (SDE),

$$\mathrm{d}X_t = -\alpha X_t \, \mathrm{d}t + \sqrt{2} \, \mathrm{d}B_t, \quad X_0 \sim \mu_{\text{data}}, \tag{1}$$

for some $\alpha \geq 0$, where $B_t$ denotes the $d$-dimensional Brownian motion. It is well known (Haussmann and Pardoux, 1986) that the time reversal $Y_t := X_{T-t}$ of (1) satisfies

$$\mathrm{d}Y_t = \alpha Y_t \, \mathrm{d}t + 2\nabla \log p_{T-t}(Y_t) \, \mathrm{d}t + \sqrt{2} \, \mathrm{d}B_t, \tag{2}$$

where $p_t$ denotes the density of $X_t$. Therefore, the task of generating samples from $\mu_{\text{data}}$ can be solved by simulating paths of (2). To that end, the unknown *score function*, $\nabla \log p_t$ in (2), is approximated

---

*Authors contributed equally to this work. Correspondence to {last name}@stats.ox.ac.uk

39th Conference on Neural Information Processing Systems (NeurIPS 2025).

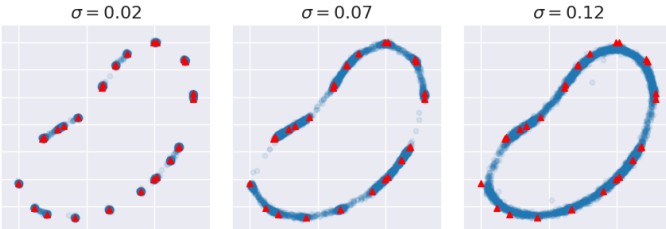

Figure 1: **Isotropic smoothing of the score function *identifies manifold structure*.** Training data (▲) is shown against generated samples (●) from a diffusion model using the smoothed score $\nabla \log \hat{p}_t * \mathcal{N}_\sigma$, where the scale of the kernel is increased from $\sigma = 0.02$ to $\sigma = 0.12$. As $\sigma$ increases, generated samples begin to fill out more of the manifold *without* having seen training samples in those regions.

by minimising the *(population) score matching loss* (see Hyvärinen (2005)):

$$\ell_{\text{sm}}(s) = \int_0^T \mathbb{E}\left[\|s(t, X_t) - \nabla \log p_t(X_t)\|^2\right] \mathrm{d}t. \tag{3}$$

In (3), the expectation is taken over samples from $X_t$ (see (1)), when started from the true data distribution $X_0 \sim \mu_{\text{data}}$. In practice, one has access only to a finite dataset of samples $\{x_i\}_{i=1}^N$ from $\mu_{\text{data}}$ and so $\ell_{\text{sm}}$ must be approximated empirically. Therefore, during training, the noising process is *not* started from the target distribution $\mu_{\text{data}}$, but is instead initialised from the *empirical measure*, $\hat{\mu}_{\text{data}} = \frac{1}{N} \sum_i \delta_{x_i}$. This gives rise to the *empirical score matching loss*,

$$\hat{\ell}_{\text{sm}}(s) := \int_0^T \mathbb{E}\left[\|s(t, \hat{X}_t) - \nabla \log \hat{p}_t(\hat{X}_t)\|^2\right] \mathrm{d}t, \tag{4}$$

where $\hat{p}_t$ is the density of the forward process, $\hat{X}_t$, which is initialised from $\hat{\mu}_{\text{data}}$.

One quirk of this objective is that it possesses a unique minimiser[2] identical to the *empirical score function*, $\nabla \log \hat{p}_t(x)$. As a result, if one were to reverse the noising process with this minimiser, one would arrive close to the empirical measure $\hat{\mu}_{\text{data}}$, reproducing the training data instead of generating novel samples from the target distribution. In fact, it has been shown that any approximation sufficiently close to $\nabla \log \hat{p}_t$ will produce samples belonging to the training dataset (Pidstrigach, 2022). However, in practice, diffusion models trained with this objective perform well and avoid memorisation, suggesting that regularisation is key to their generalisation capabilities.

The study of generalisation in diffusion models can be divided into three parts: (i) formulating the learning problem via score matching and its empirical approximation; (ii) the inductive bias of the training procedure and model architecture; and (iii) how regularising the minimiser of (4) affects the reverse SDE and generated samples. Of these, (ii) has been widely studied—spanning architectures, regularisation, and optimisation—and while far from completely understood, there are numerous studies into how neural network training promotes bias towards smooth functions interpolating the data (Rahaman et al., 2019; Mulayoff et al., 2021; Ma and Ying, 2021; Vardi, 2023). In contrast, (i) and (iii) are diffusion-specific and comparatively underexplored, so we focus on these two parts.

To account for the inductive bias during score matching, we propose a simple model built upon smooth approximations to the minimiser of $\hat{\ell}_{\text{sm}}$. In particular, we consider the score function $s^k$, which smooths the empirical score function $\nabla \log \hat{p}_t$, with a generic probability kernel $k$:

$$s^k(t, x) = \int \nabla \log \hat{p}_t(y) \, k_x(\mathrm{d}y). \tag{5}$$

While this significantly simplifies the possible inductive bias employed during training, it succeeds in capturing a defining property of diffusion models: as a result of the approach of score matching, any smoothing resulting from inductive bias occurs at the level of the score function—in the *log-domain*.

Beyond these considerations, understanding the generalisation of diffusion models also requires an analysis of the data distributions they successfully model. There is growing support for the

---

[2]Here we mean in the $L^2$ sense: any minimiser of $\hat{\ell}_{\text{sm}}$ is identical to $\nabla \log \hat{p}_t$ almost everywhere.

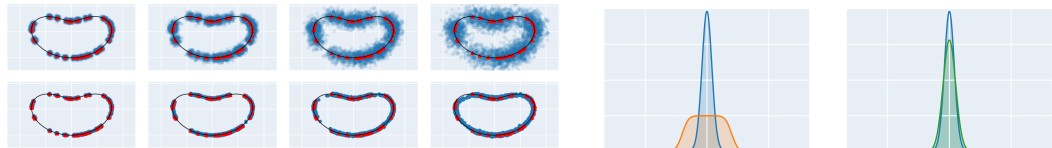

Figure 2: **Density smoothing generates samples off-manifold, whereas score smoothing generates samples that retain manifold structure.** Left: Comparing samples (•) drawn from a KDE (top) versus from a diffusion model with the smoothed score (bottom) from Figure 1 (training data is •). The scale of smoothing increases from left to right. Right: 1D intuition for data-domain versus log-domain smoothing. The left sub-figure shows the Gaussian (−) smoothed in data-domain (−), and the right sub-figure shows the Gaussian smoothed in log-domain (−) with the same kernel.

theory that diffusion models are particularly successful at modelling distributions adhering to the *manifold hypothesis*, wherein high-dimensional data concentrates on a lower-dimensional manifold (Tenenbaum et al., 2000; Bengio et al., 2012; Goodfellow et al., 2016). A standing conjecture is that generative models, including diffusion models and flow-based approaches, owe their success partly to their capacity to uncover these hidden structures (Pidstrigach, 2022; De Bortoli, 2022; Loaiza-Ganem et al., 2024; Farghly et al., 2025). This raises a critical question: what mechanism allows diffusion models to so effectively identify and leverage this underlying manifold structure? In this work, we argue that the practice of smoothing in the *logarithmic domain* plays a key role.

## 2 Log-domain smoothing retains geometric structure

This work contributes to a small but growing literature on the effect of score-smoothing in diffusion models (Scarvelis et al., 2025; Chen, 2025; Gabriel et al., 2025). In this section, we provide intuition for how diffusion models inherently perform smoothing in the log-density domain via their score matching objective. We then show that log-domain smoothing is crucial for preserving the underlying manifold structure of the data. Finally, we revisit the manifold hypothesis and explore how specific characteristics of the smoothing kernel can guide the model to generalise along different geometries.

### 2.1 Diffusion models smooth in the log-domain

As outlined in the introduction, we model the inductive bias of neural network training by a smoothing kernel $k$ (see (5)). Assuming that the nature of the inductive bias does not vary too rapidly over the spatial domain, we can treat the kernel as locally constant. In this case, the convolution will commute with the gradient operation, and we obtain the following simple but consequential equation:

$$s^k(t, x) = k * \nabla \log \hat{p}_t(x) = \nabla \left( k * \log \hat{p}_t(x) \right). \tag{6}$$

Therefore, smoothing the score function corresponds to smoothing the empirical density $\hat{p}_t$ in the *log-domain*, as opposed to smoothing at the density-level directly. Consequently, when a trained diffusion model generates samples by following the reverse SDE in (2), it effectively utilises scores derived from this log-smoothed version of the empirical density:

$$\hat{p}_t^k(\mathrm{d}x) \propto \exp \left( \int \log \hat{p}_t(y) k_x(\mathrm{d}y) \right) \mathrm{d}x, \tag{7}$$

where we use $k_x$ to denote the distribution of the smoothing kernel centred at $x$.

Sampling from a diffusion model involves discretising the backwards process in (2) using the learned score function approximation. To correct for discretisation and approximation error, so-called *corrector-steps* are interspersed between iterations (Song et al., 2021; Karras et al., 2022). This involves running Langevin Monte Carlo to correct the distribution of the diffusion model, maintaining correspondence between the diffusion model samples and the distribution associated with the score function. Furthermore, to account for instability near convergence, the technique of *early stopping* is often used, where the reverse process is terminated an amount of time $\epsilon > 0$ before convergence (Song et al., 2021). With this, we arrive at our approximation of the diffusion model output as the log-domain smoothed empirical measure, $\hat{p}_\epsilon^k$. Indeed, this is the density recovered by the diffusion model with score function $s^k$ with sufficient correction steps and sufficiently fine discretisation.

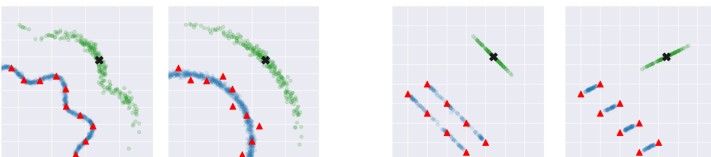

Figure 3: **The smoothing kernel influences the manifold on which generated samples lie.** The empirical score corresponding to training data (▲) is smoothed with different kernels. We visualise the kernels with samples (●) from $k_x$. We use the smoothed scores to generate samples (●). Despite using the same training data, different kernels generate samples that lie on different manifolds.

This characterisation of diffusion model output through smoothing in the log-domain identifies a distinction between diffusion models and classical density-level estimators. For example, the classical kernel density estimation (KDE) (Tsybakov, 2009) approximates the underlying data distribution by smoothing the empirical measure $\hat{\mu}_{\text{data}}$ with a kernel $k$, providing an estimator of the form,

$$\hat{q}^k_{\text{KDE}}(dx) = \int k_x(y)\hat{\mu}_{\text{data}}(dy)dx.$$

In words, the KDE also approximates the data distribution by smoothing the empirical data distribution, but it performs its smoothing in the *data-domain* as opposed to the log-domain.

## 2.2   Smoothing in log-domain preserves manifold structure

Capturing the geometry underlying the data distribution is a critical aspect of effective generative modelling. We briefly provide some intuition for why log-domain smoothing plays a vital role here. Consider a data distribution that is concentrated on a manifold within the larger data space. Data-domain smoothing techniques, such as the KDE, yield a positive probability density wherever the smoothing kernel overlaps with the manifold, leading to a *smearing* of the density away from the manifold. In contrast, smoothing in the log-domain offers a distinct advantage—when we transition to the log-domain, locations where the original density is zero are mapped to $-\infty$. Consequently, if a smoothing kernel extends into regions off-manifold, the resulting smoothed log-density in those regions is effectively $-\infty$.

In Section 3, we make this intuition more concrete, theoretically showing that smoothing the empirical density in the log-domain approximates smoothing along the data manifold. We start by analysing the case in which the data is supported on a linear manifold (see Section 3.1), where we obtain a perfect correspondence between smoothing the empirical density in the log-domain and smoothing along the (linear) data manifold. Then, in Section 3.2, we state our main theoretical result that generalises this to the curved manifold setting—showing that smoothing in the log-domain approximates a geometry-adapted smoothing, which generates samples close to the underlying manifold.

## 2.3   Choosing an interpolating manifold via geometric bias

The manifold hypothesis traditionally assumes that data lies on a low-dimensional *true* submanifold. However, in many real-world scenarios, this assumption is too rigid: rather than adhering to a single well-defined manifold, data likely exhibits geometric structure that different interpolating manifolds can approximate. This is especially true when the size of the dataset is small relative to the dimension and curvature of the space. In such settings, the focus is no longer on recovering a *true* manifold, but on choosing a *plausible* interpolating manifold. With this, we arrive at our next question of interest: how does the algorithm *choose* the interpolating manifold?

Returning to score smoothing, this reframes our central question to one of understanding how smoothing induces biases in the geometric structure of generated samples. As Figure 3 illustrates, the geometry of the output distribution depends entirely on the directions of smoothing. By choosing a kernel that aligns with certain geometric structures (e.g., tangent to a circle), the diffusion model is biased to interpolate along the corresponding manifold. We term this relationship between sample geometry and the smoothing kernel—or more broadly, the inductive bias of the score matching algorithm—as the *geometric bias* of the diffusion model. In Section 4, we develop theory and experiments that identify key structural properties of the smoothing kernel that dictate this bias.

# 3 Geometry-adaptivity of log-domain smoothing

In this section, we provide theoretical results that aim to capture and make concrete the intuition presented in Section 2.2, examining the smoothed density $\hat{p}_\epsilon^k$ in (7) as a tractable proxy for the diffusion model output. We also consider a *manifold-adapted* counterpart to $\hat{p}_\epsilon^k$, denoted by $\hat{p}_\epsilon^{k^{\mathcal{M}}}$. The kernel $k^{\mathcal{M}}$ acts similarly to $k$, but restricts the smoothing to occur only along level sets of the manifold, spreading mass along the manifold without destroying the geometric structure. Note that a priori, one may not have knowledge of the manifold structure and thus could not construct such a kernel $k^{\mathcal{M}}$—here, we use it merely as a theoretical tool to represent a desirable behaviour of a generative model: that interpolation identifies and preserves geometric structure. To show that smoothing with a generic kernel $k$ is geometry-adaptive, we wish to show that $\hat{p}_\epsilon^k$ is *close* to its manifold-adapted counterpart $\hat{p}_\epsilon^{k^{\mathcal{M}}}$. In this section, we provide theoretical results analysing this relationship, and the properties of the data manifold and diffusion model that influence it.

## 3.1 Warm-up: linear setting

We first restrict our analysis to the setting where the data distribution is supported on a $d^*$-dimensional affine subspace $\mathcal{M} = \{x \in \mathbb{R}^d : Ax = b\}$, where $A \in \mathbb{R}^{(d-d^*) \times d}$ is a row-orthonormal matrix and $b \in \mathbb{R}^{(d-d^*)}$. This allows us to provide intuition for our main result in Section 3.2 where we generalise the linear case. Consider the simplified setting where the kernel $k$ is location-independent

$$k_x := \text{law}(x + \xi),$$

where $\xi$ is a zero-mean random variable independent of $x$. In this case, we have the following result.

**Proposition 3.1.** *The log-domain smoothed density satisfies the property,*

$$\hat{p}_\epsilon^k = \hat{p}_\epsilon^{k^{\mathcal{M}}}, \qquad \text{where } k_x^{\mathcal{M}} := \text{law}(x + P\xi), \tag{8}$$

*where $P := I - A^T A$ is the projection onto $Null(A) = \{x \in \mathbb{R}^d : Ax = 0\}$.*

The kernel $k_x^{\mathcal{M}}$ is a modification of $k_x$ that smooths only along the plane parallel to $\mathcal{M}$ passing through $x$. From this proposition, we see that in the affine setting, smoothing in the log-domain with respect to a generic kernel $k$ is equivalent to smoothing with the geometry-adapted kernel $k^{\mathcal{M}}$. In other words, log-domain smoothing is fundamentally *geometry-adaptive*.

We now provide a brief exposition of the proof technique which also forms the basis of the proof in the more general setting. Given the training set $\{x_i\}_{i=1}^N$, we can directly compute the noised empirical densities $\hat{p}_t(x)$ and the corresponding score functions. Recall that the LogSumExp (LSE) function is defined on any finite set $\{r_i\}_i \subset \mathbb{R}$ and is given by $\text{LSE}(\{r_i\}_i) := \log(\sum_i \exp(r_i))$. Using this function, we can succinctly express the empirical log-density as

$$\log \hat{p}_t(x) = \text{LSE}\left(\left\{-\|x - \mu_t x_i\|^2/(2\sigma_t^2)\right\}_{i=1}^N\right) + C_t, \tag{9}$$

for data-independent quantities $C_t, \mu_t, \sigma_t$ given in Appendix B.1. We then use the following property.

**Fact 3.2.** *For any $\{r_i\}_i \subset \mathbb{R}$ and any constant $c \in \mathbb{R}$, $\text{LSE}(\{r_i + c\}_i) = \text{LSE}(\{r_i\}_i) + c$.*

Using this fact, we can decompose the log-density into directions tangent and normal to the data manifold. Indeed, using the fact that $x_i \in \mathcal{M}$ we obtain,

$$\log \hat{p}_t(x + \xi) = \text{LSE}\left(\left\{-(\|P(x + \xi - x_i)\|^2 + \|A^T A(x + \xi - x_i)\|^2)/(2\sigma_t^2)\right\}_i\right) + C_t$$

$$= \text{LSE}\left(\left\{-(\|P(x + \xi - x_i)\|^2 + \|A(x + \xi) - b\|^2)/(2\sigma_t^2)\right\}_i\right) + C_t$$

$$= \underbrace{\text{LSE}\left(\left\{-\|P(x + \xi - x_i)\|^2/(2\sigma_t^2)\right\}_i\right)}_{\text{tangent}} - \underbrace{\|A(x + \xi) - b\|^2/(2\sigma_t^2)}_{\text{normal}} + C_t,$$

In other words, interactions between the noise $\xi$ and the data occur only in the tangent direction, and the normal direction is constant with respect to the examples $\{x_i\}_i$. Once taking the expectation of the above expression, we obtain that the log-density of $\hat{p}_t^k$ is identical, up to a constant, to the log-density of $\hat{p}_t^{k^{\mathcal{M}}}$, which only applies smoothing in directions tangent to the manifold. We refer to Appendix C for the complete derivation.

## 3.2 The case of curved manifolds

In this section, we state the main theoretical contribution of this work in which we show that smoothing in log-density is fundamentally geometry-adaptive. Similar to the above analysis, we do this by deriving a relationship between $p_\epsilon^k$ using an uninformed kernel $k$, and $p_\epsilon^{k^{\mathcal{M}}}$ using its manifold-adapted counterpart $k^{\mathcal{M}}$. We consider curved manifolds satisfying the following assumption.

**Assumption 3.3.** Suppose that $\mu_{\text{data}}$ lies on a smooth compact submanifold $\mathcal{M} \subset \mathbb{R}^d$, and that $\mu_{\text{data}}$ restricted to $\mathcal{M}$ admits a density $p_\mu$ satisfying $c_\mu := \inf_{\mathcal{M}} p_\mu > 0$.[3]

Our approach to generalising the proof from the previous section is to use the defining feature of Riemannian manifolds—that locally the manifold behaves as if it were flat. The distance that one must be to the manifold depends on its curvature, which we control with an object from differential geometry called the *reach*. This object defines the maximum distance from the manifold for which the projection to the manifold, $\Pi_{\mathcal{M}}$, is well-defined, i.e. a unique element of the manifold is closest. For example, if $\mathcal{M}$ were a sphere, the reach would be the radius.

**Assumption 3.4** (Manifold reach)**.** The manifold $\mathcal{M}$ has a reach no smaller than $\tau > 0$, i.e. for all $x \in \mathbb{R}^d$ with $\text{dist}(x, \mathcal{M}) < \tau$, there exists a *unique* $x^\star \in \mathcal{M}$ such that $\text{dist}(x, \mathcal{M}) = \|x - x^\star\|$.

The reach is inversely related to the maximum curvature of the manifold. Our assumption effectively requires that the curvature is globally upper-bounded. This assumption, as well as the lower bound on the density, have been used in several recent works and is common in the manifold hypothesis and manifold learning literature (Aamari, 2017; Potaptchik et al., 2025; Azangulov et al., 2024). We refer to Appendix D.2 for further discussion and details regarding the reach of the manifold.

To generalise the manifold-adapted kernel in (8), consider the kernel's projection onto level sets of the manifold $\mathcal{M}_r = \{x \in \mathbb{R}^d : \text{dist}(x, \mathcal{M}) = r\}$. We define the manifold-adapted modification by

$$k_x^{\mathcal{M}} := (\Pi_{\mu_\epsilon \mathcal{M}_{r(x)}})_* k_x, \qquad r(x) := \mathbb{E}_{Y \sim k_x}[\text{dist}(Y, \mu_\epsilon \mathcal{M})^2]^{1/2}, \qquad (10)$$

where $\mu_\epsilon \mathcal{M}$ is the element-wise scaling of $\mathcal{M}$ by $\mu_\epsilon$ and $(\Pi_{\mu_\epsilon \mathcal{M}_{r(x)}})_*$ denotes the push-forward by the projection mapping $\Pi_{\mu_\epsilon \mathcal{M}_{r(x)}}$, that is, the distribution of $\Pi_{(\mu_\epsilon \mathcal{M})_{r(x)}}(Y), Y \sim k_x$. The function $r(x)$ approximates the distance of $x$ to the manifold, but with some correction according to the variance of the kernel in directions normal to the manifold. Therefore, similar to the definition in (8), the kernel $k_x^{\mathcal{M}}$ is a modification of $k_x$ adapted to the geometry of $\mathcal{M}$ by smoothing only in directions tangential to the manifold. We refer to Appendix B.3 for some additional details regarding the definition of $k^{\mathcal{M}}$, including a discussion on the well-posedness of the projection function.

The variance of the smoothing kernel $k$ in directions normal to the manifold will prove to be an important object in our bound, leading us to make the following assumption.

**Assumption 3.5.** There are constants $K, K_{\max} \geq 0$ such that for all $x \in \mathbb{R}^d$, $Y \sim k_x$,

$$\mathbb{E}\big[|\text{dist}(Y, \mathcal{M}) - \text{dist}(x, \mathcal{M})|^2\big] \leq K^2, \quad |\text{dist}(Y, \mathcal{M}) - \text{dist}(x, \mathcal{M})| \leq K_{\max}, \text{ almost surely.}$$

By measuring the change in distance to the manifold under smoothing, $K$ and $K_{\max}$ quantify noise in directions normal to the manifold and kernel adaptivity to the manifold structure. If $k$ places most mass tangentially, then $Y \sim k_x$ stays near the level set through $x$ and so $K$ is small. For example, in the Gaussian case $k_x = \mathcal{N}(x, \sigma^2 I_d)$, we have that $K^2 \approx (d - d^*)\sigma^2$ whenever $\sigma$ is taken small.

Unlike in the affine case, $\hat{p}_\epsilon^k$ and $p_\epsilon^{k^{\mathcal{M}}}$ are not the same in general, so instead we show that these distributions are *close*. We consider the Rényi divergence, $D_q$—a natural generalisation of the Kullback-Leibler divergence (which is the case $q = 1$). For the sake of brevity, we leave the definition and a brief exposition on the Rényi divergence to Appendix B.2 and we state our main result.

**Theorem 3.6.** *Under assumptions 3.3, 3.4, and 3.5 and if $K_{\max} < \tau/96$, then for any $q \in [1, 1 + \tau/96K], \delta \in (0, 1]$, whenever $N > N_{\min}(\delta), \epsilon < \epsilon_{\max}$ we obtain with probability at least $1 - \delta$ that*

$$D_q\big(\hat{p}_\epsilon^{k^{\mathcal{M}}} \big\| \hat{p}_\epsilon^k\big) \lesssim \frac{K}{\tau} \max\left\{d^* + 1, \ (c_\mu^2 N)^{-\frac{1}{d^*}} \epsilon^{-1}\right\},$$

*where the quantities $\epsilon_{\max}$ and $N_{\min}(\delta) \lesssim (d^* + \log(\delta^{-1}))\tau^{-2d^*}$ are defined in (53) and (45).*[4]

---

[3]Here, we take $p_\mu$ to be the density with respect to the volume measure of the manifold $\mathcal{M}$, which is itself inherited from the Lebesgue measure.

[4]Here, $\lesssim$ denotes an upper bound that ignores multiplicative logarithmic factors.

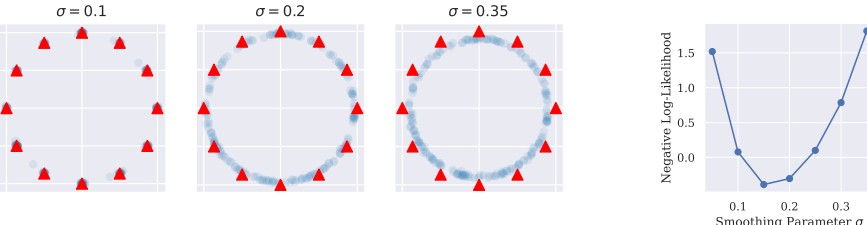

Figure 4: **Score smoothing can promote generalisation along curved manifolds, but too much smoothing can distort the desired structure.** Left: Training data (▲) against generated samples (●) using isotropic Gaussian score smoothing with variance $\sigma^2$. Right: Corresponding population negative log-likelihood, calculated for 1000 points on the true circular manifold. See Appendix G.1.

For large $N$, the right-hand side depends only on dimension, curvature and kernel scale. This bound shows that for log-domain smoothing to become geometry-adaptive, it is sufficient for the scale of smoothing normal to the manifold to be small relative to the manifold curvature and dimension. When $N$ is small relative to $\epsilon^{-d^*}$, the bound becomes $K/\tau\epsilon$, highlighting the role that early stopping plays in the data-sparse setting. The dependence on $K$ also provides insight for how the behaviour of $\hat{p}_\epsilon^k$ depends on the kernel's manifold-alignment—when $k$ is already more aligned with the manifold structure, $K$ is smaller, and the closer $\hat{p}_\epsilon^k$ is to its manifold-adapted counterpart $\hat{p}_\epsilon^{k^{\mathcal{M}}}$.

We once again emphasise the key difference between the log-domain smoothing that we consider, and the traditional KDE approach which instead smooths the empirical measure in the density-domain. As KDE bandwidth increases, samples rapidly leave the data manifold, whereas smoothed-score diffusion models produce new samples along the manifold structure without deviating far from it.

### 3.3 Log-domain smoothing and generalisation

So far, we have presented results pertaining to the similarity of $\hat{p}_\epsilon^k$ and its manifold-adapted counterpart $\hat{p}_\epsilon^{k^{\mathcal{M}}}$. While it is intuitively clear that smoothing with the manifold adapted kernel $k^{\mathcal{M}}$ will help promote generalisation, we provide two results to validate that this is indeed the case. The following result demonstrates that $\hat{p}_\epsilon^k$ preserves mass concentration around the manifold structure.

**Corollary 3.7.** *Consider the setting of Theorem 3.6, then for any $\delta \in (0,1]$, whenever $\epsilon < \epsilon_{\max}, N > \max\{N_{\min}(\delta), \epsilon^{-2}\}$, we obtain that with probability at least $1 - 2\delta$ that,*

$$\mathbb{P}_{Y \sim \hat{p}_\epsilon^k}(\text{dist}(Y, \mathcal{M}) \geq r + m | S) \leq 2\exp(-r^2/8\epsilon), \qquad \text{for all } r \geq 0,$$

*where $m > 0$ satisfies $m^2 \lesssim K^2 + \frac{K}{\tau} \max\left\{d^* + 1, \left(c_\mu^2 N\right)^{-1/d^*} \epsilon^{-1}\right\} + \epsilon d + (\epsilon/c_\mu)^{2/d^*}$.*

This corollary shows that the distance to the manifold decays exponentially fast, at nearly the same rate as the noised empirical measure $\hat{p}_\epsilon$, prior to smoothing. In other words, log-domain smoothing *preserves* concentration to the manifold. We note that when $K$ is large, the concentration bound becomes less strong. Next, we show that smoothing with $k^{\mathcal{M}}$ does indeed distribute mass along the manifold structure. We let $T_x\mathcal{M}$ denote the space of vectors tangent to $\mathcal{M}$ at $x$.

**Proposition 3.8.** *Consider the setting of Theorem 3.6, assuming further that $\mathbb{E}_{Y \sim k_x}[\text{dist}(Y, \mathcal{M})^2]$ is constant in $x \in \mathcal{M}$. Let $\delta \in (0, 1]$ and suppose that $N > N_{\min}(\delta)$, then, with probability at least $1 - \delta$, it holds that for any $x \in \mathcal{M}$,*

$$\hat{p}_\epsilon^k(x) \geq \hat{p}_\epsilon^k(x_i^\star) \exp\left(-\frac{16\mu_\epsilon^2 F \|x - x_i^\star\|}{\sigma_\epsilon^2 c_\mu^{1/(d^*+2)}}\right), \qquad x_i^\star := \text{argmin}_{\{x_i\}_{i=1}^N} \|x - x_i\|,$$

*where $F^2 := \sup_{x \in \mathcal{M}, v \in T_x\mathcal{M}} \frac{v^T \mathcal{I}(x)v}{\|v\|^2}$ and $\mathcal{I}(x) \in \mathbb{R}^{d \times d}$ is the Fisher information matrix of $k_x$.*

The quantity $F$ is an upper bound on the Fisher information of the kernel along the manifold. In the case where $k$ is a Gaussian kernel with variance $\sigma^2 I_d$, we have that $F = 1/\sigma$. Thus, whenever $\sigma$ is taken large, arbitrary points on the manifold receive similar density as the training data.

Together, propositions 3.8 and 3.7 show that log-domain smoothing distributes probability mass along the manifold while preserving geometric structure. While further work must be done to obtain rigorous generalisation bounds, the above results already suggest an interesting relationship between

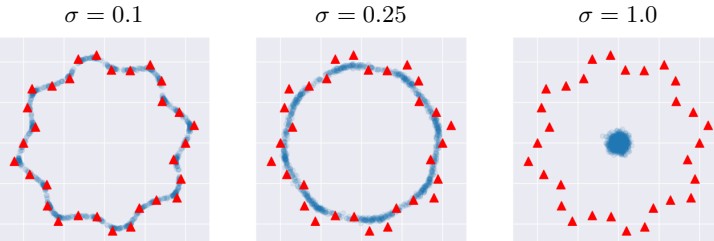

Figure 5: **Different smoothing kernels can isolate alternative manifolds, given the same training data.** Training data (▲) against generated samples (●) using isotropic Gaussian score smoothing. By changing the smoothing variance $\sigma^2$, different geometries are realised.

smoothing scale and generalisation: as smoothing grows, $K$ increases, and once it becomes large relative to $\tau/\epsilon d^*$, the strength of the bound in Corollary 3.7 weakens; meanwhile $F$ decreases, increasing the distribution of mass along the manifold. This suggests a possible trade-off in the generalisation error that is governed by the scale of the smoothing and its relationship to $\epsilon, d^*$ and $\tau$. We explore this in Figure 4, where we plot population error (given by the negative log-likelihood) against scale of smoothing and observe a U-shaped curve, suggesting moderate smoothing can improve over the empirical measure $\hat{p}_\epsilon$ while oversmoothing can worsen generalisation.

## 4 Rethinking the manifold hypothesis: geometry and inductive bias

So far, we have considered the traditional setting of the manifold hypothesis, where the goal is to recover the data's *true* geometry under a well-defined ground-truth manifold. Yet with finite data—especially in high dimensions—many plausible interpolations fit. In practice, "correctness" is task-dependent: if an interpolation meets application-specific criteria, the generative model is deemed successful. Consequently, practitioners are not recovering a single *true* interpolation; network architectures and training algorithms implicitly inject biases that steer models toward desirable behaviours. This motivates a shift in perspective in this section: rather than assuming a ground truth manifold, we study how inductive biases make the model *choose* an interpolating manifold. We refer to this form of bias as the model's *geometric bias*.

### 4.1 Geometric bias of log-domain smoothing

In Figure 3, we study a toy case where data (red triangles) admit several plausible interpolating manifolds. When the smoothing kernel is aligned with the wavy-circle level sets, the generated samples (blue) remain faithful to the wavy geometry, while aligning with the base circle yields circular samples. This indicates that smoothing the empirical score *parallel* or *tangentially* to a target manifold $\mathcal{M}$ induces a geometric bias toward it. The right of Figure 3 shows that tailored kernels can even alter dimension and connectivity. In Figure 5, we consider isotropic Gaussian smoothing. Here, the *scale* of the smoothing controls the bias: small bandwidths preserve fine waviness, larger bandwidths recover the broader circular shape, and excessive noise leads to eventual sample collapse.

### 4.2 Geometry-adaptivity and geometric bias

The theoretical analysis of Section 3.2 can also be extended to the present setting, allowing us to further elucidate the relationship between the smoothing kernel and geometric bias. In particular, we provide a modification of Theorem 3.6 that quantifies how well log-domain smoothing adapts to different manifolds, without the requirement for the data to belong to that manifold.

We define the set of *permissible manifolds* $\mathbb{M}_\mu$ to be the set of all smooth compact submanifolds $\mathcal{M} \subseteq \mathbb{R}^d$ with non-zero reach $\tau_\mathcal{M} > 0$, satisfying the property $c_\mathcal{M} := \operatorname{ess\,inf}_{\mu_{\text{data}}} p_{\mu,\mathcal{M}} > 0$ where $p_{\mu,\mathcal{M}}$ denotes the density of $(\Pi_\mathcal{M})_* \mu_{\text{data}}$ with respect to the volume measure on $\mathcal{M}$. In other words, $\mathbb{M}_\mu$ consists of all manifolds with bounded curvature, such that the projection of $\mu_{\text{data}}$ onto $\mathcal{M}$ has full support. Furthermore, given any $\mathcal{M} \in \mathbb{M}_\mu$, we let $d^*_\mathcal{M}$ denote the manifold dimension and define

$$K^2_\mathcal{M} := \sup_{x \in \mathbb{R}^d} \mathbb{E}_{Y \sim k_x}\big[|\operatorname{dist}(Y, \mathcal{M}) - \operatorname{dist}(x, \mathcal{M})|^2\big], \quad K_{\max,\mathcal{M}} := \|\operatorname{dist}(Y, \mathcal{M}) - \operatorname{dist}(x, \mathcal{M})\|_{L^\infty}.$$

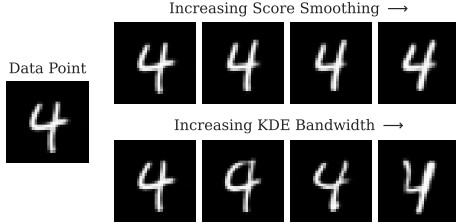

Increasing Score Smoothing ⟶

Data Point

Increasing KDE Bandwidth ⟶

Figure 6: As smoothing is increased, generations from the score-smoothed diffusion model remain in the manifold structure. In contrast, samples from the KDE quickly deviate, leading to poor reconstructions.

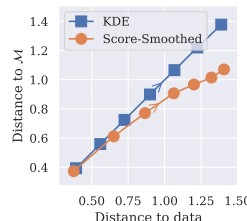

Figure 7: Comparison of $L^2$ distance to closest point in training data and closest point in $\mathcal{M}$. Arrows indicate increasing amounts of smoothing.

With this, we can state our second main result.

**Theorem 4.1.** *Let $\mathcal{M} \in \mathbb{M}_\mu$ and $\Delta_{\mathcal{M}} := \mathrm{dist}(\{x_i\}_{i=1}^N, \mathcal{M})$. Then, for any $\delta \in (0,1]$, whenever $K_{\mathcal{M},\max} + \Delta_{\mathcal{M}} \le \tau_{\mathcal{M}}/96$ and $N, \epsilon^{-1}$ is sufficiently large, with probability at least $1 - \delta$ we have*

$$D_2\big(\hat{p}_\epsilon^{k^{\mathcal{M}}}\big\|\hat{p}_\epsilon^k\big) \lesssim \frac{K_{\mathcal{M}}(d_{\mathcal{M}}^* + 1)}{\tau_{\mathcal{M}}} + \frac{K_{\mathcal{M}}\Delta_{\mathcal{M}}}{\epsilon} + \frac{K_{\mathcal{M}}^2 \Delta_{\mathcal{M}}^2 d_{\mathcal{M}}^*}{\tau_{\mathcal{M}}\,\epsilon}. \tag{11}$$

While Theorem 3.6 controls how closely log-domain smoothing adapts to an underlying true geometry, Theorem 4.1 instead bounds how well *any given* manifold describes the geometry induced by log-domain smoothing. If a manifold $\mathcal{M} \in \mathbb{M}_\mu$ makes the right-hand side of this bound small, then smoothing under a generic kernel $k$ is similar to smoothing under the the $\mathcal{M}$-adapted kernel $k^{\mathcal{M}}$. Indeed, such a manifold $\mathcal{M}$ *effectively captures the geometric bias of the smoothing kernel $k$.* Thus we can identify favoured manifolds by optimising the right-hand side with respect to $\mathcal{M} \in \mathbb{M}_\mu$.

Analysing (11) shows how the smoothing kernel drives geometric bias. The bound trades off curvature $\tau_{\mathcal{M}}^{-1}$ against interpolation error $\Delta_{\mathcal{M}}$, modulated by $K_{\mathcal{M}}$, $d_{\mathcal{M}}^*$ and $\epsilon$. For small $\epsilon$, the second term can dominate, so optimisation favours manifolds with small $\Delta_{\mathcal{M}}$, even at the expense of a larger $\tau_{\mathcal{M}}$. As smoothing increases, $K_{\mathcal{M}}$ grows and the $K_{\mathcal{M}}$ term dominates, shifting preference toward lower-curvature manifolds (as in figures 3 and 5). Moreover, if $k_x$ emphasises certain directions in its smoothing, then choosing $\mathcal{M}$ tangent to them keeps $K_{\mathcal{M}}$ small, yielding a low-dimensional manifold $\mathcal{M}$ aligned with those directions that minimises $\Delta_{\mathcal{M}}$ and $d_{\mathcal{M}}^*$. That is, a low-dimensional manifold that is tangent to the smoothing directions while best interpolating the data.

## 5 High-dimensional experiments

So far, our experiments have focused on illustrative low-dimensional settings to complement our theory. In this section, we consider higher-dimensional settings to investigate the extent to which the identified phenomena persist in scenarios more representative of practical applications.

### 5.1 Generation in latent space

We begin with MNIST and define a 32-dimensional VAE latent space (following Rombach et al. (2022)). We study the digit-4 manifold $\mathcal{M}$, which comprises a lower-dimensional structure in the latent space. This ground-truth manifold is approximated using all samples of the digit 4, from which we use a subset of 100 samples as our training dataset. We compare a smoothed-score diffusion model using an isotropic Gaussian kernel to KDE, which corresponds to density-level smoothing.

To assess how well the manifold structure is preserved, we visualize samples as smoothing increases in Figure 6. The top and bottom rows display samples from a score-smoothed diffusion model and KDE, respectively, at different smoothing scales. With score smoothing, generations perfectly recover training examples (plotted on the left) at small smoothing levels. As the amount of smoothing increases, the samples become progressively novel images that are not present in the dataset yet nonetheless decode to resemble 4's, indicating mass is spread primarily *along* the underlying geometry. KDE, in contrast, deteriorates more as bandwidth grows, as generated samples move substantially off-manifold. We provide a quantitative assessment of this behaviour by reporting the average distance of

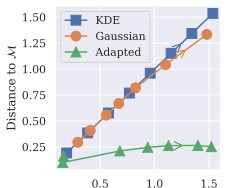 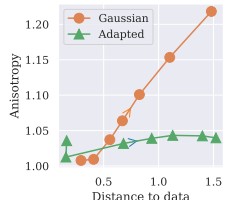 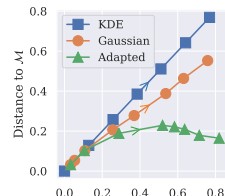 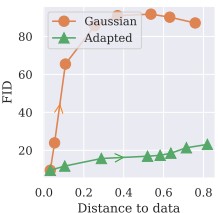

Figure 8: Comparison of smoothing kernels for the synthetic 'bump' image manifold. Left: Comparing $L_2$ distance to data and $\mathcal{M}$. Right: Anisotropy of generated 'bump' samples.

Figure 9: Comparison of smoothing kernels for the MNIST image manifold. Left: Comparing $L_2$ distance to data and $\mathcal{M}$. Right: FID of generated samples.

samples to the manifold against distance to the training set in Figure 7. As KDE smoothing increases, the distance to the manifold increases identically with the distance to the data, whereas the diffusion samples sometimes become closer to other 4s not used for training. See Appendix G.2 for details.

## 5.2 Generation in pixel space

While latent-space generation is common, diffusion models also succeed directly in pixel space. We therefore repeat the analysis there, focusing on $1d$ image manifolds where the geometry can be controlled and evaluated. Data is naturally more separated in pixel space, yielding many permissible manifolds; we leverage this to test how kernel choice affects the geometry of the sampling distribution.

**Synthetic image manifold**    We construct a closed $1d$ manifold $\phi : [0, 2\pi) \to \mathbb{R}^{64 \times 64}$ that maps an angle to an image of a bump function centred at the corresponding angle around a circle, and form a dataset of 16 equidistant points on this curve. For a visualisation of the manifold, see Appendix G.3. We compare isotropic Gaussian score smoothing against the KDE by plotting distance-to-manifold versus distance-to-data across smoothing scales in Figure 8. We further test a *manifold-adapted* score-smoothing kernel that translates the manifold $\mathcal{M}$ to pass through the current point and smooths along it. Crucially, the sampler "knows" the manifold *only* via the smoothing mechanism; the empirical score itself uses only the training dataset. The adapted kernel produces samples significantly closer to the true manifold than isotropic Gaussian smoothing, supporting the discussion in Section 4. As a measure of how visually "on-manifold" the generations are, we also report the anisotropy of the generated bump samples in Figure 8. The samples using the adapted smoothing exhibit lower anisotropies as the degree of smoothing increases, indicating that they appear visually closer to the data manifold than those obtained using Gaussian smoothing.

**MNIST manifold in pixel space**    We repeat the pixel-space study on MNIST by constructing an explicit image-space manifold $\phi : [0, 1] \to \mathbb{R}^{32 \times 32}$ by decoding a curve in VAE latent space; the VAE only defines the manifold, and the diffusion runs entirely in pixel space. In Figure 9, we plot distance to the manifold $\mathcal{M}$ versus distance to the training set, again finding that manifold-adapted smoothing stays closer to $\mathcal{M}$ than Gaussian smoothing. As a complementary measure of visual quality, Figure 9 also reports FID to a held-out test set and shows a consistent benefit for score-function smoothing. Gaussian smoothing is known to produce barycentres of training datapoints (Scarvelis et al., 2025), which in pixel space appears as increasing blurring and yields a steep increase in the FID at larger smoothing. The manifold-adapted kernel mitigates this blurring and avoids the same increase in FID value. Full details and additional plots for different curves $\phi$ appear in Appendix G.4.

## 6 Conclusion

In this work, we have investigated how implicit regularisation caused by smoothing of the empirical score function interacts with the manifold structure of data. In particular, we identify that smoothing at the level of the log-domain is implicitly geometry-adaptive, behaving similarly to a manifold-adapted kernel when given enough samples. Beyond the data-rich setting, we observe that the *choice* of smoothing kernel can shape the generated distribution. Future work could examine how inductive biases in deep learning architectures and training influence the smoothing that occurs in practice.

## Acknowledgements

Tyler Farghly was supported by Engineering and Physical Sciences Research Council (EPSRC) [grant number EP/T517811/1] and by the DeepMind scholarship. Peter Potaptchik is supported by the EPSRC CDT in Modern Statistics and Statistical Machine Learning [EP/S023151/1], a Google PhD Fellowship, and an NSERC Postgraduate Scholarship (PGS D). Samuel Howard is supported by the EPSRC CDT in Modern Statistics and Statistical Machine Learning [grant number EP/S023151/1]. George Deligiannidis and Jakiw Pidstrigach acknowledge support from EPSRC [grant number EP/Y018273/1]. The authors would like to thank Iskander Azangulov and Arya Akhavan for stimulating discussions and Ioannis Siglidis for valuable feedback.

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

# Technical Appendices and Supplementary Material

## A   Extended Discussion

**Related work**   In the manifold setting, Pidstrigach (2022) and De Bortoli (2022) provide precise convergence bounds for diffusion models. Recently, there has been a surge of interest in providing refined results in the manifold setting (Oko et al., 2023; Chen et al., 2023; Tang and Yang, 2024; Li and Yan, 2024; Azangulov et al., 2024; Potaptchik et al., 2025), or under other specific structural settings (Shah et al., 2023; Chen et al., 2024; Wang et al., 2024). Additionally, many works have focused on empirically validating the manifold hypothesis for data such as images (Fefferman et al., 2016; Pope et al., 2021; Stanczuk et al., 2024; Brown et al., 2023; Kamkari et al., 2024). Our work also shares similarities with wider literature regarding how manifold structure interacts with learning tasks, such as Genovese et al. (2012), Cheng and Wu (2013), Moscovich et al. (2017), and Gao et al. (2022).

Recently, there has been an increased interest in understanding generalisation and memorisation in diffusion models. Memorisation of training data has been observed empirically by Somepalli et al. (2023) and Carlini et al. (2023) when the capacity of the network is large relative to the number of training samples. Other works investigate how inductive biases of neural network architectures aid in generalisation (Kadkhodaie et al., 2024; Niedoba et al., 2025; Kamb and Ganguli, 2025). The recent work of Vastola (2025) examines the role of noise in the objective. The dichotomy between generalisation and memorisation has been investigated in Yoon et al. (2023), Gu et al. (2023), Wen et al. (2024), Zhang et al. (2024), and Baptista et al. (2025). The works of Raya and Ambrogioni (2023), Biroli et al. (2024), and Ventura et al. (2025) examine the roles of distinct regimes in the generative process.

This work contributes to a small but growing line of research into the effect of score-smoothing in diffusion models. Scarvelis et al. (2025) previously studied isotropic Gaussian and Gumbel smoothing of the score function, as a training-free alternative for running diffusion models, and show that this generates barycentres of training datapoints. Chen (2025) investigates the effect of score smoothing on generalisation in the $1d$ linear setting. Concurrently, the work of Gabriel et al. (2025) also studies the effect of a kernel-smoothed score function and its relation to preserving manifold structure, though with different analysis techniques.

**Limitations and further investigations**   Our argument that the log-domain smoothed measure $\hat{p}_\epsilon^k$ approximates the output of the diffusion model with score smoothing relies on the exchangeability of gradients (see (6)), a property that holds for location-independent kernels. Extending our framework to the more general case of location-dependent kernels is an important next step. Similarly, our key theorems (theorems 3.6 and 4.1) currently require the scale of noise normal to the manifold, $K$, to be small relative to its curvature, $\tau$. A more complete characterisation of the geometric bias would therefore require relaxing this assumption. Furthermore, a more robust characterisation of the geometric bias of log-domain smoothing would also benefit from a matching lower bound on the Rényi divergence. While we have demonstrated through heuristic arguments how our results show the generalisation potential of log-domain smoothing, our work stops short of deriving a formal generalisation error bound. Whether log-domain smoothing alone could produce optimal generalisation bounds is a question that we leave open for future investigation. Finally, we believe this theoretical framework could be a valuable tool for the related literature analysing the memorisation and privacy properties of diffusion models.

The experiments presented here illustrate that the *type* of smoothing can influence the geometric structure of the generated samples. However, they also highlight the challenges posed by high-curvature manifolds, where smoothing the empirical score alone may not suffice. Real-world image manifolds tend to be highly curved, yet diffusion models still generalize well from relatively few training samples (Kadkhodaie et al., 2024). Furthermore, in practical settings we do not have knowledge of the ground-truth manifold structure to explicitly apply manifold-adaptive smoothing. If such adaptation occurs, it must arise implicitly from the model's inductive biases. This suggests that other factors must also be at play, such as biases induced by neural architectural choices like convolutions and attention (Kamb and Ganguli, 2025). We do not examine the behaviour of such practical architectures in this work, but remark that understanding to what extent architectural designs choices interact with the ideas presented here is an interesting direction for future study.

## B Notation and omitted details

In this section, we include some technical details concerning the theoretical results of the paper that were omitted for the sake of readability.

### B.1 Properties of the forward process

In Section 1, we introduced the forward process in (1). Throughout the proofs, we use the following property of the forward process:

$$X_t | X_0 \sim N(\mu_t X_0, \sigma_t I_d), \qquad \mu_t = e^{-\alpha t}, \qquad \sigma_t^2 = \begin{cases} \alpha^{-1}(1 - \mu_t^2), & \text{if } \alpha > 0, \\ 2t, & \text{otherwise.} \end{cases} \qquad (12)$$

When $\alpha = 0$, this follows immediately from properties of the Wiener process and when $\alpha > 0$, $X_t$ becomes the Ornstein-Uhlenbeck process and the result follows from a standard analysis (e.g. see Pavliotis, 2014; Cohen and Elliott, 2015).

Using this fact, we have the following closed-form expression for $\hat{p}_t$, the density of the empirical forward process $\hat{X}_t$:

$$\log \hat{p}_t = \log \left( \frac{1}{N} \sum_{i=1}^{N} p_{X_t | X_0}(x | x_i) \right)$$

$$= \log \left( \frac{1}{N} \sum_{i=1}^{N} \exp(-\|x - \mu_t x_i\|^2 / 2\sigma_t^2) \right) - \frac{d}{2} \log(2\pi\sigma_t^2)$$

$$= \text{LSE} \left( \left\{ -\|x - \mu_t x_i\|^2 / 2\sigma_t^2 \right\}_{i=1}^{N} \right) + C_t,$$

where $C_t = -\log(N) - \frac{d}{2} \log(2\pi\sigma_t^2)$ and we recall the definition of the function $\text{LSE}(\{r_i\}_i) := \log(\sum_i \exp(r_i))$.

### B.2 Rényi divergence

We provide a brief exposition of the Rényi divergence, which is a measure of difference between two measures. Given two measures $\mu, \nu$ on $\mathbb{R}^d$ and $q \in (1, \infty)$ we define the $q$-Rényi divergence by

$$D_q(\mu\|\nu) = \begin{cases} \frac{1}{q-1} \log \int (\frac{d\mu}{d\nu}(x))^{q-1} \mu(dx), & \text{if } \mu \ll \nu, \\ \infty, & \text{otherwise.} \end{cases}$$

For the case of $q = 1$ we set $D_q$ to be the Kullback-Leibler (KL) divergence,

$$D_1(\mu\|\nu) = \begin{cases} \int \log \frac{d\mu}{d\nu}(x) \, \mu(dx), & \text{if } \mu \ll \nu, \\ \infty, & \text{otherwise.} \end{cases}$$

Indeed, whenever $D_q(\mu\|\nu) < \infty$ for some $q > 1$, it can be shown that $\lim_{q \to 1^+} D_q(\mu\|\nu) = D_1(\mu\|\nu)$. Furthermore, whenever $\frac{d\mu}{d\nu}$ is bounded $\mu$-almost surely, we obtain,

$$\lim_{q \to \infty} D_q(\mu\|\nu) = \log \left( \text{ess sup}_\mu \frac{d\mu}{d\nu} \right),$$

which is taken to be $D_\infty(\mu\|\nu)$. Thus, the Rényi divergence provides a natural interpolation between the KL divergence and the worst-case regret, with $D_q$ increasing in $q$. This measure of distance recently gained popularity in the sampling (Vempala and Wibisono, 2019; Chewi et al., 2022; Erdogdu et al., 2022; Mousavi-Hosseini et al., 2023) and privacy (Mironov, 2017) literatures as a stronger alternative to traditional divergences. We refer to (Erven and Harremoes, 2014) and (Chewi et al., 2022) for further properties of this divergence.

## B.3 Projections

Throughout this work, we frequently utilise the projection mapping $\Pi_{\mathcal{M}} : \mathbb{R}^d \to \mathcal{M}$ which maps $x \in \mathbb{R}^d$ to the nearest element of $\mathcal{M}$. In cases where $\mathcal{M}$ is curved, we run in to the issue that the projection is not well-defined as there could be multiple elements that are equally close to $x$. In most places in the proof we consider quantities $x$ that are sufficiently close that the projection function is uniquely defined (see reach in Appendix D.2) but, for example, when we define the manifold adapted kernel in (10), we use the projection for all $x \in \mathbb{R}^d$.

Throughout the proofs of this work we do not utilise any property of the projection aside from the fact that it maps $x$ to some element of the manifold that is of distance $\text{dist}(x, \mathcal{M})$ away from $x$. For that reason, $\Pi_{\mathcal{M}}$ can be taken to be any mapping onto $\mathcal{M}$ such that $\|x - \Pi_{\mathcal{M}}(x)\| = \text{dist}(x, \mathcal{M})$. Since $\mathcal{M}$ is taken to be a closed set, such a mapping always exists and we will take this choice of mapping to be fixed throughout the work.

When $\alpha > 0$ the samples generated at early stopping time $\epsilon$ are slightly biased due to contractions of the Ornstein-Uhlenbeck process. For this reason, we will frequently consider the contracted manifold,

$$\mu_\epsilon \mathcal{M} = \{\mu_\epsilon x : x \in \mathcal{M}\},$$

where $\mu_\epsilon$ is as defined in (12), and we will frequently use the shorthand $\mathcal{M}^\epsilon := \mu_\epsilon \mathcal{M}$. Given a projection mapping $\Pi_{\mathcal{M}}$ onto $\mathcal{M}$, we take the projection mapping $\Pi_{\mathcal{M}^\epsilon}$ onto $\mathcal{M}^\epsilon$ to be given by

$$\Pi_{\mathcal{M}^\epsilon}(x) = \mu_\epsilon \Pi_{\mathcal{M}}(x/\mu_\epsilon).$$

## C  Manifold-adaptivity in the affine setting

We begin with the proof of Proposition 3.1 concerning the affine setting. Recall the assumption that the support of $\mu_{\text{data}}$ is restricted to the affine subspace $\mathcal{M} = \{x \in \mathbb{R}^d : Ax = b\}$, where $A \in \mathbb{R}^{(d-d^*) \times d}$ is row-orthonormal and $b \in \mathbb{R}^{(d-d^*)}$, and we write the smoothing kernel in the following form:

$$k_x := \text{law}(x + \xi),$$

where $\xi$ is a centred random variable independent of $x$. Throughout the proof, we use the null space projection matrix $P := I - A^T A$.

*Proof of Proposition 3.1.* Since $P$ is the projection matrix onto $\text{Null}(A)$, any $z \in \mathbb{R}^d$ can be decomposed as

$$\begin{aligned} \|z\|^2 &= \|Pz\|^2 + \|(I - P)z\|^2 \\ &= \|Pz\|^2 + \|A^T Az\|^2 \\ &= \|Pz\|^2 + \|Az\|^2, \end{aligned}$$

where the final line follows from the fact that $A$ is row-orthonormal and so $AA^T = I_{(d-d^*)}$. Using fact 3.2 and the assumption that $Ax_i = b$ for every $i \in [N]$, we obtain the identity

$$\log \hat{p}_\epsilon(z) = \text{LSE}\left(\left\{-\frac{\|z - x_i\|^2}{2\sigma_\epsilon^2}\right\}_i\right) + C_t \tag{13}$$

$$= \text{LSE}\left(\left\{-\frac{\|P(z - x_i)\|^2 + \|A(z - x_i)\|^2}{2\sigma_\epsilon^2}\right\}_i\right) + C_t \tag{14}$$

$$= \text{LSE}\left(\left\{-\frac{\|P(z - x_i)\|^2}{2\sigma_\epsilon^2}\right\}_i\right) - \frac{\|Az - b\|^2}{2\sigma_\epsilon^2} + C_t. \tag{15}$$

This decomposition separates the influence of $z$ into normal and tangent directions with respect to $\mathcal{M}$.

Now, let $x \in \mathbb{R}^d$ and define $Y_x = (x + \xi) \sim k_x$ and $\widetilde{Y}_x = (x + P\xi) \sim k_x^{\mathcal{M}}$. Since $P = P^2$, we observe that

$$\|P(Y_x - x_i)\| = \|P(\tilde{Y}_x - x_i)\|.$$

Furthermore, using the fact that $\xi_x$ is centred, we also have,

$$\mathbb{E}[\|AY_x - b\|^2] = \|Ax - b\|^2 + \mathbb{E}[\|A\xi\|^2], \qquad \mathbb{E}[\|A\tilde{Y}_x - b\|^2] = \|Ax - b\|^2 + \mathbb{E}[\|AP\xi\|^2].$$

In particular, substituting into (15) and taking the expectation, we conclude that

$$\mathbb{E}[\log \hat{p}_\epsilon(Y_x)] = \mathbb{E}\left[ \mathrm{LSE}\left( \left\{ -\frac{\|P(Y_x - x_i)\|^2}{2\sigma_\epsilon^2} \right\}_i \right) \right] - \frac{\|Ax - b\|^2}{2\sigma_\epsilon^2} + C,$$

and

$$\mathbb{E}[\log \hat{p}_\epsilon(\tilde{Y}_x)] = \mathbb{E}\left[ \mathrm{LSE}\left( \left\{ -\frac{\|P(Y_x - x_i)\|^2}{2\sigma_\epsilon^2} \right\}_i \right) \right] - \frac{\|Ax - b\|^2}{2\sigma_\epsilon^2} + \tilde{C},$$

for constants $C, \tilde{C}$ independent of $x$. Therefore, the log-density of $\hat{p}_\epsilon^k$ and $\hat{p}_\epsilon^{k \mathcal{M}}$ are identical up to a constant. $\qquad \square$

# D  Lemmata

For proving the results for the more the general manifold setting, we require several additional properties of the log-sum-exp function, smooth manifolds and tubular neighbourhoods. In this section we collect these results.

## D.1  Stability of the LSE function

The following lemma provides stability bounds for the LSE function which we make use of throughout our analysis. It can be seen as a generalisation of Fact 3.2 which is heavily used in the analysis of the affine case.

**Lemma D.1.** *For any $\{x_i\}_{i=1}^N \subset \mathbb{R}$ and $\{\varepsilon_i\}_{i=1}^N \subset \mathbb{R}$, we have*

$$\mathrm{LSE}(\{x_i + \epsilon_i\}_{i=1}^N) - \mathrm{LSE}(\{x_i\}_{i=1}^N) = \int_0^1 \frac{\sum_{i=1}^N \exp(x_i + r\epsilon_i)\epsilon_i}{\sum_{i=1}^N \exp(x_i + r\epsilon_i)} dr. \tag{16}$$

*In particular, we have that*

$$\mathrm{LSE}(\{x_i + \epsilon_i\}_{i=1}^N) - \mathrm{LSE}(\{x_i\}_{i=1}^N) \le \max\{\epsilon_i\}. \tag{17}$$

*Proof.* From the chain rule, we compute the partial derivatives,

$$\frac{\partial}{\partial x_j} \mathrm{LSE}(\{x_i\}_i) = \frac{\exp(x_j)}{\sum_{i=1}^N \exp(x_i)}.$$

Therefore, by the fundamental theorem of calculus, we obtain that

$$\mathrm{LSE}(\{x_i + \epsilon_i\}_i) - \mathrm{LSE}(\{x_i\}_i) = \int_0^1 \sum_{j=1}^N \epsilon_j \frac{\partial}{\partial x_j} \mathrm{LSE}(\{x_i + r\epsilon_i\}_i)\, dr$$

$$= \int_0^1 \sum_{j=1}^N \frac{\exp(x_j + r\epsilon_j)\epsilon_j}{\sum_{i=1}^N \exp(x_i + r\epsilon_i)} dr,$$

completing the proof of (16). To obtain (17), we use the fact that the sum is a weighted average of the sequence $\{\epsilon_i\}_i$, and so it has the property that,

$$\sum_{i=1}^N \frac{\exp(x_i + r\epsilon_i)\epsilon_i}{\sum_{j=1}^N \exp(x_j + r\epsilon_j)} \le \max_i \epsilon_i.$$

$\qquad \square$

We also note that if all $\epsilon_i$ are identical, we readily recover Fact 3.2 from (16).

## D.2 Manifold reach

Next, we collect some facts about the *reach* of the manifold, which quantifies how far one can extend from the manifold before the projection onto it ceases to be unique. We refer to (Aamari, 2017) for a more detailed exposition. We begin with the rigorous definition.

**Definition D.2.** The reach of a set $A \subset \mathbb{R}^d$, is defined by $\tau_A = \inf_{p \in A} \operatorname{dist}(p, Med(A))$, where we define the set,

$$Med(A) = \left\{ z \in \mathbb{R}^d : \exists p, q \in A \text{ s.t. } p \neq q, \|p - z\| = \|q - z\| = \operatorname{dist}(z, A) \right\}.$$

The reach defines the maximum distance at which the projection to the set is unique. In the case where the set $A$ is a smooth submanifold of $\mathbb{R}^d$, the reach captures the curvature of the manifold and provides an upper bound on the distance at which the manifold appears approximately flat. The following lemma demonstrates this, controlling the curvature of paths along the manifold using the reach. We use the notation $N_x \mathcal{M}$ to denote the *normal space* of the manifold $\mathcal{M}$ at $x \in \mathcal{M}$ which consists of all vectors perpendicular to the tangent space at $x$.

**Lemma D.3.** *Suppose that the manifold $\mathcal{M}$ has reach $\tau_{\mathcal{M}} > 0$, then for any $x, y \in \mathcal{M}$, $v \in N_x \mathcal{M}$,*

$$|\langle v, x - y \rangle| \leq \frac{\|v\|}{2\tau_{\mathcal{M}}} \|x - y\|^2.$$

*Proof.* Let $x, y \in \mathcal{M}$, $v \in N_x \mathcal{M}$ and define $z = x + rv/\|v\|$ for some $r \in (0, \tau_{\mathcal{M}})$ so that $z \notin Med(\mathcal{M})$. Since the projection is uniquely defined with $\Pi_{\mathcal{M}}(z) = x$, we have,

$$\|z - y\| \geq \operatorname{dist}(z, M) = r.$$

On the other hand, we have,

$$\|z - y\|^2 = \|x - y\|^2 + r^2 + \frac{2r}{\|v\|} \langle x - y, v \rangle.$$

Combining these two and rearranging, we obtain the bound,

$$\langle y - x, v \rangle \leq \frac{\|v\|}{2r} \|x - y\|^2.$$

By replacing $v$ with $-v$ which is also in the normal space, we obtain the opposite direction, hence obtaining,

$$|\langle x - y, v \rangle| \leq \frac{\|v\|}{2r} \|x - y\|^2.$$

Since this bound holds for all $r \in (0, \tau_{\mathcal{M}})$, we can take $r \to \tau_{\mathcal{M}}^-$ to obtain the bound in the statement. $\square$

With this, we can control the geodesic distance on the manifold by the standard Euclidean distance.

**Lemma D.4.** *Suppose that the manifold $\mathcal{M}$ has reach $\tau_{\mathcal{M}} > 0$. Let $x, y \in \mathcal{M}$ such that $\|x - y\| \leq \tau_{\mathcal{M}}/2$ and let $\gamma_t$ be a geodesic (shortest path) between $x, y$ on $\mathcal{M}$. Then, we have the bound,*

$$\int \|\partial_t \gamma_t\| dt \leq 2\|x - y\|.$$

Therefore, if we are close enough to the manifold, its inherited metric behaves roughly like the Euclidean one. The proof of this lemma can be found in Lemma III.21 of (Aamari, 2017).

## D.3 Concentration under the manifold hypothesis

We now turn to results concerning probability measures supported on submanifolds with bounded reach. The next lemma controls the mass of a small ball centred on the manifold, showing that the rates are similar to the affine case.

**Lemma D.5.** *Suppose that the measure $\mu_{\text{data}}$ is supported on a smooth compact submanifold $\mathcal{M}$ with reach $\tau_{\mathcal{M}} > 0$ and dimension $d^*$. Then, for any $r \leq \pi\tau_{\mathcal{M}}/2\sqrt{2}$, we have*

$$\mu_{\text{data}}(B_r(x)) \geq c_\mu r^{d^*}, \qquad c_\mu = \inf_{B_r(x)} p_\mu,$$

*where $p_\mu$ denotes the density of $\mu_{\text{data}}$ with respect to the volume measure on $\mathcal{M}$.*

For the proof of this lemma, we refer to the proof of Proposition 4.3 of (Aamari et al., 2019) or Lemma III.23 of (Aamari, 2017).

We use this bound, to obtain a result concerning the concentration of the empirical measure on the manifold. To this end, we recall a bound on the covering number of the manifold. Given $r > 0$, the covering number $N_{\text{cov}}(\mathcal{M}, r)$ is defined as the minimum number of Euclidean balls of radius $r$ required to cover the subset of $\mathbb{R}^d$ defined by the space $\mathcal{M}$. The following lemma is from Proposition III.11 of Aamari, 2017.

**Lemma D.6.** *Consider the setting of Lemma D.5 and suppose that $c_\mu > 0$, then for any $\varepsilon \in (0, \tau_{\mathcal{M}}/2)$ we have,*

$$N_{cov}(\mathcal{M}, 2\varepsilon) \leq \frac{1}{c_\mu \varepsilon^{d^*}}.$$

We can now prove a bound on the concentration of the empirical measure.

**Lemma D.7.** *Suppose that the measure $\mu_{\text{data}}$ is supported on a compact smooth submanifold $\mathcal{M}$ with reach $\tau_{\mathcal{M}} > 0$ and dimension $d^*$ and $c_\mu > 0$. Then, for any $r \in (0, \tau_{\mathcal{M}}]$, $\delta \in (0, 1)$, we have*

$$\mathbb{P}\left(\inf_{x \in M} \hat{\mu}_{\text{data}}(B_r(x)) \geq c_\mu \frac{r^{d^*}}{2^{d^*+1}}\right) \geq 1 - \delta,$$

*whenever,*

$$N \geq 16\Big(\big(144d^* \log(4/r) + 144\log(1/c_\mu)\big) \vee \frac{\log(\delta^{-1})}{2}\Big)(r/2)^{-2d^*} c_\mu^{-2}. \tag{18}$$

*Proof.* According to Lemma D.6, there exists a set $\mathcal{C} \subset \mathcal{M}$ such that $\{B_{r/2}(c) : c \in \mathcal{C}\}$ forms a covering of $\mathcal{M}$ with $|\mathcal{C}| \leq c_\mu^{-1} 4^{d^*} r^{-d^*}$. Thus, for any $x \in M$ there exists $c \in \mathcal{C}$ such that $x \in B_{r/2}(c)$ and therefore $B_{r/2}(c) \subset B_r(x)$. From this we deduce the following bound

$$\inf_{x \in \mathcal{M}} \hat{\mu}_{\text{data}}(B_r(x)) \geq \inf_{c \in \mathcal{C}} \hat{\mu}_{\text{data}}(B_{r/2}(c)).$$

Therefore, it suffices to lower bound this object on the right-hand side.

Next, using a rudimentary bound from the empirical processes literature (for example, see Section 4.2 of (Wainwright, 2019) or Section 7.1 of (Handel, 2014)), we obtain the bound,

$$\mathbb{P}\left(\sup_{c \in \mathcal{C}} |\hat{\mu}_{\text{data}}(B_{r/2}(c)) - \mu_{\text{data}}(B_{r/2}(c))| \geq 12\sqrt{\frac{\log |\mathcal{C}|}{N}} + \varepsilon\right) \leq \exp(-2N\varepsilon^2).$$

Thus, choosing $\varepsilon = c_\mu (r/2)^{d^*}/4$, we obtain that under (18), it follows that

$$\mathbb{P}\left(\sup_{C \in \mathcal{C}} |\hat{\mu}_{\text{data}}(C) - \mu_{\text{data}}(C)| \geq c_\mu (r/2)^{d^*}/2\right) \leq \delta. \tag{19}$$

To conclude the proof, we use Lemma D.5 to obtain that,

$$\inf_{c \in \mathcal{C}} \hat{\mu}_{\text{data}}(B_{r/2}(c)) \geq \inf_{c \in \mathcal{C}} \hat{\mu}_{\text{data}}(B_{r/2}(c)) - \sup_{c \in M} |\hat{\mu}_{\text{data}}(B_{r/2}(c)) - \mu_{\text{data}}(B_{r/2}(c))|$$

$$\geq c_\mu (r/2)^{d^*} - \sup_{x \in M} |\hat{\mu}_{\text{data}}(B_{r/2}(x)) - \mu_{\text{data}}(B_{r/2}(x))|.$$

Combining this with (19), we arrive at the bound in the statement. $\qquad\square$

## D.4 Weyl's tube formula

The sets $\mathcal{M}_r$ and $\mathcal{M}_r^\epsilon$ are related to the notion of *tubes* that have been investigated in the differential geometry literature (Gray, 2004; Weyl, 1939). We borrow a result from Weyl, 1939 that computes the volume enclosing these sets. Let $\mathcal{M}_{\leq r}^\epsilon := \{x \in \mathbb{R}^d : \mathrm{dist}(x, \mathcal{M}^\epsilon) \leq r\}$.

**Proposition D.8** (Weyl's Tube Formula)**.** *Suppose Assumption 3.3 holds, then for all $r \geq 0$,*

$$\lambda(\mathcal{M}_{\leq r}^\epsilon) = \sum_{p=0}^{\lfloor d^*/2 \rfloor} \tilde{k}_{2p}(\mathcal{M}^\epsilon) r^{d-d^*+2p},$$

*for some quantities $\tilde{k}_{2p}(\mathcal{M}^\epsilon) \geq 0$.*

The quantities $\tilde{k}$ are related to the integrated mean curvature of $\mathcal{M}$ and further details about these quantities can be found in (Gray, 2004) where the result is stated in Section 1.1. In this work, we develop upper bounds in such a way that the final result does not depend on these quantities.

Note that using this result, we can obtain estimates for the integrals of functions depending on $\lambda(\mathcal{M}_{\leq r}^\epsilon)$ using the expression,

$$\int_{\mathbb{R}^d} f(\mathrm{dist}(x, \mathcal{M}^\epsilon)) dx = \int_0^\infty f(r) \frac{d}{dr} \lambda(\mathcal{M}_{\leq r}^\epsilon) dr$$

$$= \sum_{p=0}^{\lfloor d^*/2 \rfloor} (d - d^* + 2p) \tilde{k}_{2p}(\mathcal{M}^\epsilon) \int_0^\infty f(r) r^{d-d^*-1+2p} dr. \qquad (20)$$

# E  Proofs for the main results

This section of the appendix provides the proofs for theorems 3.6 and 4.1. These theorems establish the core result that smoothing in the log-domain is approximately geometry-adaptive, meaning that smoothing with a generic kernel $k$ behaves similarly to smoothing with a manifold-adapted kernel $k^{\mathcal{M}}$. We begin by proving some lemmas that are involved in the proof of both of these theorems.

## E.1  Controlling the log-density ratio

To establish the proximity between $\hat{p}_\epsilon^k$ and $\hat{p}_\epsilon^{k^{\mathcal{M}}}$ in divergence, we must control the ratio of their densities. In this section, we fix a permissible manifold $\mathcal{M} \in \mathbb{M}_\mu$, and use $K_{\mathcal{M}}, K_{\max,\mathcal{M}}, \tau_{\mathcal{M}}$ and $d_{\mathcal{M}}^*$ as in Section 4.1. We also fix $x \in \mathbb{R}^d$ and let $Y \sim k_x, \tilde{Y} = \Pi_{\mathcal{M}_{r(x)}}(Y)$, so that $\tilde{Y} \sim k_x^{\mathcal{M}}$. Using the expression in (9), we can express the density ratio by,

$$\log \frac{d\hat{p}_\epsilon^k}{d\hat{p}_\epsilon^{k^{\mathcal{M}}}}(x)$$

$$= \mathbb{E}\left[ \mathrm{LSE}\left( \left\{ -\frac{\|Y - \mu_\epsilon x_i\|^2}{2\sigma_\epsilon^2} \right\}_{i=1}^N \right) - \mathrm{LSE}\left( \left\{ -\frac{\|\tilde{Y} - \mu_\epsilon x_i\|^2}{2\sigma_\epsilon^2} \right\}_{i=1}^N \right) \Big| S \right] + \log\left( \frac{C^{\mathcal{M}}}{C} \right), \tag{21}$$

where we define the normalising constants,

$$C = \int \exp\left( \int \log \hat{p}_\epsilon(y) k_x(dy) \right) dx, \qquad C^{\mathcal{M}} = \int \exp\left( \int \log \hat{p}_\epsilon(y) k_x^{\mathcal{M}}(dy) \right) dx. \tag{22}$$

We proceed similarly to the proof in the affine case (see Appendix C), decomposing the LSE function into normal and perpendicular components. We use the decomposition,

$$\|y - \mu_\epsilon x_i\|^2 = \|y - \Pi_{\mathcal{M}^\epsilon}(y)\|^2 + 2\langle y - \Pi_{\mathcal{M}^\epsilon}(y), \Pi_{\mathcal{M}^\epsilon}(y) - \mu_\epsilon x_i \rangle + \|\Pi_{\mathcal{M}^\epsilon}(y) - \mu_\epsilon x_i\|^2,$$

along with Fact 3.2 to obtain that,

$$\mathrm{LSE}\left( \left\{ -\frac{\|Y - \mu_\epsilon x_i\|^2}{2\sigma_\epsilon^2} \right\}_{i\in[N]} \right)$$

$$= \mathrm{LSE}\left( \left\{ -\frac{\|\Pi_{\mathcal{M}^\epsilon}(Y) - \mu_\epsilon x_i\|^2 + 2\langle Y - \Pi_{\mathcal{M}^\epsilon}(Y), \Pi_{\mathcal{M}^\epsilon}(Y) - \mu_\epsilon x_i \rangle}{2\sigma_\epsilon^2} \right\}_i \right) - \frac{\|Y - \Pi_{\mathcal{M}^\epsilon}(Y)\|^2}{2\sigma_\epsilon^2}.$$

It follows from the definition of $\tilde{Y}$ and $r(x)$ that,

$$\|\tilde{Y} - \Pi_{\mathcal{M}^\epsilon}(Y)\|^2 = \|\tilde{Y} - \Pi_{\mathcal{M}^\epsilon}(\tilde{Y})\|^2$$
$$= r(x)^2$$
$$= \mathbb{E}[\|Y - \Pi_{\mathcal{M}^\epsilon}(Y)\|^2].$$

Therefore we may apply Fact 3.2 once more to obtain,

$$\mathbb{E}\left[\operatorname{LSE}\left(\left\{-\frac{\|Y - \mu_\epsilon x_i\|^2}{2\sigma_\epsilon^2}\right\}_{i \in [N]}\right)\Big| S\right]$$

$$= \mathbb{E}\left[\operatorname{LSE}\left(\left\{-\frac{\|\Pi_{\mathcal{M}^\epsilon}(Y) - \mu_\epsilon x_i\|^2 + 2\langle Y - \Pi_{\mathcal{M}^\epsilon}(Y), \Pi_{\mathcal{M}^\epsilon}(Y) - \mu_\epsilon x_i\rangle}{2\sigma_\epsilon^2}\right\}_i\right.\right.$$
$$\left.\left. -\frac{\|\tilde{Y} - \Pi_{\mathcal{M}^\epsilon}(Y)\|^2}{2\sigma_\epsilon^2}\Big| S\right]\right.$$

$$= \mathbb{E}\left[\operatorname{LSE}\left(\left\{-\frac{\|\Pi_{\mathcal{M}^\epsilon}(Y) - \mu_\epsilon x_i\|^2 + 2\langle Y - \Pi_{\mathcal{M}^\epsilon}(Y), \Pi_{\mathcal{M}^\epsilon}(Y) - \mu_\epsilon x_i\rangle}{2\sigma_\epsilon^2}\right.\right.\right.$$
$$\left.\left.\left. -\frac{\|\tilde{Y} - \Pi_{\mathcal{M}^\epsilon}(Y)\|^2}{2\sigma_\epsilon^2}\right\}_i\right)\Big| S\right]$$

$$= \mathbb{E}\left[\operatorname{LSE}\left(\left\{-\frac{\|\tilde{Y} - \mu_\epsilon x_i\|^2 + \Delta_i}{2\sigma_\epsilon^2}\right\}_{i \in [N]}\right)\Big| S\right],$$

where we define the quantity $\Delta_i := 2\langle Y - \tilde{Y}, \Pi_{\mathcal{M}^\epsilon}(Y) - \mu_\epsilon x_i\rangle$. Therefore, we obtain the simple expression,

$$\log \frac{d\hat{p}_\epsilon^k}{d\hat{p}_\epsilon^{k,\mathcal{M}}}(x) = \mathbb{E}[\Delta \operatorname{LSE}_{\mathcal{M}}(x)|S] + \log\left(\frac{C^{\mathcal{M}}}{C}\right), \tag{23}$$

$$\Delta \operatorname{LSE}_{\mathcal{M}}(x) := \operatorname{LSE}\left(\left\{-\frac{\|\tilde{Y} - \mu_\epsilon x_i\|^2 + \Delta_i}{2\sigma_\epsilon^2}\right\}_{i \in [N]}\right) - \operatorname{LSE}\left(\left\{-\frac{\|\tilde{Y} - \mu_\epsilon x_i\|^2}{2\sigma_\epsilon^2}\right\}_{i=1}^N\right).$$

Having expressed the log-density ratio in terms of $\Delta \operatorname{LSE}_{\mathcal{M}}$, our next task is to bound this quantity. For the sake of intuition, we can consider the linear setting: In this case, $\tilde{Y} - Y$ is normal to the manifold and $\Pi_{\mathcal{M}^\epsilon}(Y) - \mu_\epsilon x_i$ is tangent to the manifold, so it would follow that $\Delta_i = 0$ and thus $\Delta \operatorname{LSE}_{\mathcal{M}}(x) = 0$. In the case where the manifold is curved, it is no longer necessarily true that $\Delta_i$ is 0 and so we control $\Delta \operatorname{LSE}_{\mathcal{M}}$ using the curvature of the manifold and the stability of the LSE function.

We proceed with a simple lemma.

**Lemma E.1.** *Suppose that* $\Delta_{\mathcal{M}} := \operatorname{dist}(\{x_i\}_{i=1}^N, \mathcal{M}) < \infty$ *and* $\tau_{\mathcal{M}} > 0$. *Then, for any* $x \in \mathbb{R}^d, i \in [N]$ *we have,*

$$|\Delta_i| \leq \mu_\epsilon|\zeta|\left(\left(\frac{1}{2\tau_{\mathcal{M}}}d_i^2\right) \wedge d_i + \Delta_{\mathcal{M}}\right), \qquad d_i = \|\Pi_{\mathcal{M}}(Y/\mu_\epsilon) - \Pi_{\mathcal{M}}(x_i)\|,$$

*where we define the quantity,*

$$\zeta := \|Y - \Pi_{\mathcal{M}^\epsilon}(Y)\| - \mathbb{E}\left[\|Y - \Pi_{\mathcal{M}^\epsilon}(Y)\|^2\right]^{1/2}.$$

*Proof.* Using the definition of $\tilde{Y}$, we obtain that,

$$Y - \tilde{Y} = (Y - \Pi_{\mathcal{M}^\epsilon}(Y)) - (\tilde{Y} - \Pi_{\mathcal{M}^\epsilon}(Y))$$
$$= (Y - \Pi_{\mathcal{M}^\epsilon}(Y)) - \frac{Y - \Pi_{\mathcal{M}^\epsilon}(Y)}{\|Y - \Pi_{\mathcal{M}^\epsilon}(Y)\|}\mathbb{E}\left[\|Y - \Pi_{\mathcal{M}^\epsilon}(Y)\|^2\right]^{1/2}$$
$$= \frac{Y - \Pi_{\mathcal{M}^\epsilon}(Y)}{\|Y - \Pi_{\mathcal{M}^\epsilon}(Y)\|}\left(\|Y - \Pi_{\mathcal{M}^\epsilon}(Y)\| - \mathbb{E}\left[\|Y - \Pi_{\mathcal{M}^\epsilon}(Y)\|^2\right]^{1/2}\right).$$

With this, we can write $\Delta_i$ in the following form:

$$\Delta_i = 2\zeta \left\langle \frac{Y - \Pi_{\mathcal{M}^\epsilon}(Y)}{\|Y - \Pi_{\mathcal{M}^\epsilon}(Y)\|}, \Pi_{\mathcal{M}^\epsilon}(Y) - \mu_\epsilon x_i \right\rangle. \tag{24}$$

To control $\Delta_i$, we use Lemma D.3 as well as the Cauchy-Schwarz inequality to obtain,

$$|\Delta_i| = 2\left| \zeta \left( \left\langle \frac{Y - \Pi_{\mathcal{M}^\epsilon}(Y)}{\|Y - \Pi_{\mathcal{M}^\epsilon}(Y)\|}, \Pi_{\mathcal{M}^\epsilon}(Y) - \mu_\epsilon \Pi_{\mathcal{M}}(x_i) \right\rangle \right. \right. \tag{25}$$
$$\left. \left. + \left\langle \frac{Y - \Pi_{\mathcal{M}^\epsilon}(Y)}{\|Y - \Pi_{\mathcal{M}^\epsilon}(Y)\|}, \mu_\epsilon \Pi_{\mathcal{M}}(x_i) - \mu_\epsilon x_i \right\rangle \right) \right|$$
$$\leq 2\mu_\epsilon |\zeta| \left( \frac{\|\Pi_{\mathcal{M}}(Y/\mu_\epsilon) - \Pi_{\mathcal{M}}(x_i)\|^2}{2\tau_{\mathcal{M}}} \wedge \|\Pi_{\mathcal{M}}(Y/\mu_\epsilon) - \Pi_{\mathcal{M}}(x_i)\| + \Delta_{\mathcal{M}} \right), \tag{26}$$

completing the proof of the lemma. $\qquad\square$

Since $\zeta$ is a random variable, we next find ways of controlling it using $K_{\mathcal{M}}$ and $K_{\max,\mathcal{M}}$.

**Lemma E.2.** *Let $\zeta$ be as in Lemma E.1 and suppose that $K_{\mathcal{M}}, K_{\max,\mathcal{M}} < \infty$, then we have that,*

$$\mathbb{E}[|\zeta|^2|S]^{1/2} \leq 2K_{\mathcal{M}}, \qquad |\zeta| \leq K_{\mathcal{M}}^{\max} + K_{\mathcal{M}},$$

*almost surely.*

*Proof.* For the first bound, we use the $L^2$-triangle inequality to obtain,

$$\mathbb{E}[|\zeta|^2|S]^{1/2} \leq \mathbb{E}\left[ \left| \|Y - \Pi_{\mathcal{M}^\epsilon}(Y)\| - \|x - \Pi_{\mathcal{M}^\epsilon}(x)\| \right|^2 \Big| S \right]^{1/2}$$
$$+ \left| \|x - \Pi_{\mathcal{M}^\epsilon}(x)\| - \mathbb{E}[\|Y - \Pi_{\mathcal{M}^\epsilon}(Y)\|^2|S]^{1/2} \right|$$
$$\leq \mathbb{E}\left[ \left| \|Y - \Pi_{\mathcal{M}^\epsilon}(Y)\| - \|x - \Pi_{\mathcal{M}^\epsilon}(x)\| \right|^2 \Big| S \right]^{1/2}$$
$$+ \mathbb{E}\left[ \left( \|Y - \Pi_{\mathcal{M}^\epsilon}(Y)\| - \|x - \Pi_{\mathcal{M}^\epsilon}(x)\| \right)^2 \Big| S \right]^{1/2}$$
$$\leq 2K_{\mathcal{M}}.$$

Similarly, we can obtain $L^\infty$ bounds via,

$$\|\zeta\|_{L^\infty} = \left\| \|Y - \Pi_{\mathcal{M}^\epsilon}(Y)\| - \|x - \Pi_{\mathcal{M}^\epsilon}(x)\| \right\|_{L^\infty}$$
$$+ \left\| \|x - \Pi_{\mathcal{M}^\epsilon}(x)\| - \mathbb{E}[\|Y - \Pi_{\mathcal{M}^\epsilon}(Y)\|^2|S]^{1/2} \right\|_{L^\infty}$$
$$\leq K_{\max,\mathcal{M}} + K_{\mathcal{M}}.$$

$\qquad\square$

We now state the bound for $\Delta \operatorname{LSE}_{\mathcal{M}}$ that we use for our two main theorems.

**Lemma E.3.** *Suppose that $\Delta_{\mathcal{M}} := \operatorname{dist}(\{x_i\}_{i=1}^N, \mathcal{M}) < \infty$ and $\tau_{\mathcal{M}} > 0$, then for any $x \in \mathbb{R}^d$, it holds that*

$$\mathbb{E}[|\Delta \operatorname{LSE}_{\mathcal{M}}(x)|S]$$
$$\leq \frac{8K_{\mathcal{M}}}{\tau_{\mathcal{M}}} \left( 1 + 2\log\left(\tfrac{\tau_{\mathcal{M}}}{K_{\mathcal{M}}}\right)_+ + (\Delta_{\mathcal{M}} + \tau_{\mathcal{M}} + K_{\mathcal{M}} + D_x)\frac{12\Delta_{\mathcal{M}}}{\sigma_\epsilon^2} \right.$$
$$+ \left( 5K_{\max,\mathcal{M}}^{1/2} K_{\mathcal{M}}^{1/2} + D_x \right)^2 \frac{\mathbb{P}(\mathcal{E}_x|S)^{1/2}}{\sigma_\epsilon^2}$$
$$\left. + 2\inf_{\varepsilon_0 > 0} \left\{ \log\left( \mathbb{E}_{Y \sim k_x}[\hat{\mu}_{\text{data}}(B_{\varepsilon_0}(\Pi_{\mathcal{M}}(Y/\mu_\epsilon)))^{-1}|S] \right)_+ + \left( 1 + \tfrac{K_{\mathcal{M}} + D_x}{\tau_{\mathcal{M}}} \right)\frac{\varepsilon_0^2}{\sigma_\epsilon^2} \right\} \right),$$

*where we define $\mathcal{E}_x = \{K_{\mathcal{M}} + D_x + 2|\zeta| \geq \mu_\epsilon \tau_{\mathcal{M}}/4\}$, $D_x = \|x - \Pi_{\mathcal{M}^\epsilon}(x)\|$.*

*Proof.* For this bound, we begin with (16) of Lemma D.1 to obtain

$$|\Delta \operatorname{LSE}_{\mathcal{M}}(x)| \leq \int_0^1 \frac{\sum_{i \in [N]} \exp(-\|\tilde{Y} - \mu_\epsilon x_i\|^2/2\sigma_\epsilon^2 - r\Delta_i/2\sigma_\epsilon^2)|\Delta_i|/2\sigma_\epsilon^2}{\sum_{i \in [N]} \exp(-\|\tilde{Y} - \mu_\epsilon x_i\|^2/2\sigma_\epsilon^2 - r\Delta_i/2\sigma_\epsilon^2)} dr.$$

We decompose this further as,

$$\begin{aligned}
|\Delta \operatorname{LSE}_{\mathcal{M}}(x)| &\leq \int_0^1 \frac{\sum_{i \in I_1} \exp(-\|\tilde{Y} - \mu_\epsilon x_i\|^2/2\sigma_\epsilon^2 - r\Delta_i/2\sigma_\epsilon^2)|\Delta_i|/2\sigma_\epsilon^2}{\sum_{i \in I_1} \exp(-\|\tilde{Y} - \mu_\epsilon x_i\|^2/2\sigma_\epsilon^2 - r\Delta_i/2\sigma_\epsilon^2)} dr \\
&\quad + \int_0^1 \frac{\sum_{i \in I_1^{\mathsf{C}}} \exp(-\|\tilde{Y} - \mu_\epsilon x_i\|^2 2\sigma_\epsilon^2 - r\Delta_i/2\sigma_\epsilon^2)|\Delta_i|/2\sigma_\epsilon^2}{\sum_{i \in I_0} \exp(-\|\tilde{Y} - \mu_\epsilon x_i\|^2/2\sigma_\epsilon^2 - r\Delta_i/2\sigma_\epsilon^2)} dr \\
&\leq \mathbb{1}_{|I_1|>0} \max_{i \in I_1} \left\{ \frac{|\Delta_i|}{2\sigma_\epsilon^2} \right\} \\
&\quad + \int_0^1 \frac{\sum_{i \in I_1^{\mathsf{C}}} \exp(-\|\tilde{Y} - \mu_\epsilon x_i\|^2/2\sigma_\epsilon^2 - r\Delta_i/2\sigma_\epsilon^2)|\Delta_i|/2\sigma_\epsilon^2}{\sum_{i \in I_0} \exp(-\|\tilde{Y} - \mu_\epsilon x_i\|^2/2\sigma_\epsilon^2 - r\Delta_i/2\sigma_\epsilon^2)} dr \\
&=: \mathtt{A} + \mathtt{B},
\end{aligned}$$

where we define the sets,

$$I_0 = \{i \in [N] : \|\Pi_{\mathcal{M}}(Y/\mu_\epsilon) - \Pi_{\mathcal{M}}(x_i)\| \leq \varepsilon_0\}, \quad I_1 = \{i \in [N] : \|\Pi_{\mathcal{M}}(Y/\mu_\epsilon) - \Pi_{\mathcal{M}}(x_i)\| \leq \varepsilon_1\}$$

for some random quantities $\varepsilon_0, \varepsilon_1 > 0$.

The quantity $\mathtt{A}$ can be bounded directly using Lemma E.1. From this, we obtain,

$$\mathtt{A} \leq \frac{\mu_\epsilon}{\sigma_\epsilon^2} |\zeta| \left( \tfrac{1}{2\tau_{\mathcal{M}}} \varepsilon_1^2 + \Delta_{\mathcal{M}} \right).$$

To bound $\mathtt{B}$, we proceed with the following upper bound,

$$\begin{aligned}
\mathtt{B} &\leq \frac{\sum_{j \in I_1^{\mathsf{C}}} \exp(-\|\tilde{Y} - \mu_\epsilon x_j\|/2\sigma_\epsilon^2 + |\Delta_j|/2\sigma_\epsilon^2)|\Delta_j|/2\sigma_\epsilon^2}{\sum_{i \in I_0} \exp(-\|\tilde{Y} - \mu_\epsilon x_i\|/2\sigma_\epsilon^2 - |\Delta_i|/2\sigma_\epsilon^2)} \\
&\leq |I_0|^{-1} \max_{i \in I_0} \sum_{j \in I_1^{\mathsf{C}}} \exp(\|\tilde{Y} - \mu_\epsilon x_i\|^2/2\sigma_\epsilon^2 - \|\tilde{Y} - \mu_\epsilon x_j\|^2/2\sigma_\epsilon^2 + |\Delta_i|/2\sigma_\epsilon^2 + |\Delta_j|/\sigma_\epsilon^2), \quad (27)
\end{aligned}$$

where we have used the fact that $r \leq \exp(r)$. To further control $\mathtt{B}$, we control the quantity inside of the exponential function by choosing $\varepsilon_0$ sufficiently small and $\varepsilon_1$ sufficiently large so that the quantity in the exponential becomes negative. This allows for control of $\mathtt{B}$ by taking $\epsilon$ sufficiently small.

We start by controlling $\|\tilde{Y} - \mu_\epsilon x_i\|^2 - \|\tilde{Y} - \mu_\epsilon x_j\|^2$. For any $i \in I_0, j \in [N]$, we have that

$$\begin{aligned}
&\|\tilde{Y} - \mu_\epsilon x_i\|^2 - \|\tilde{Y} - \mu_\epsilon x_j\|^2 \\
&= \|\Pi_{\mathcal{M}^\epsilon}(Y) - \mu_\epsilon x_i\|^2 + 2\langle \tilde{Y} - \Pi_{\mathcal{M}^\epsilon}(Y), \Pi_{\mathcal{M}^\epsilon}(Y) - \mu_\epsilon x_i \rangle - \|\Pi_{\mathcal{M}^\epsilon}(Y) - \mu_\epsilon x_j\|^2 \\
&\quad - 2\langle \tilde{Y} - \Pi_{\mathcal{M}^\epsilon}(Y), \Pi_{\mathcal{M}^\epsilon}(Y) - \mu_\epsilon x_j \rangle. \quad (28)
\end{aligned}$$

The first term is bounded using the fact that $\|\Pi_{\mathcal{M}^\epsilon}(Y) - \mu_\epsilon x_i\| \leq \mu_\epsilon \varepsilon_0 + \mu_\epsilon \Delta_{\mathcal{M}}$. The second term is controlled using the technique from the proof of Lemma E.1 to deduce the bound,

$$\begin{aligned}
&\langle \tilde{Y} - \Pi_{\mathcal{M}^\epsilon}(Y), \Pi_{\mathcal{M}^\epsilon}(Y) - \mu_\epsilon x_i \rangle &(29) \\
&\leq \mu_\epsilon \|\tilde{Y} - \Pi_{\mathcal{M}^\epsilon}(Y)\| \left( \left( \tfrac{1}{2\tau_{\mathcal{M}}} \|\Pi_{\mathcal{M}}(Y/\mu_\epsilon) - \Pi_{\mathcal{M}}(x_i)\|^2 \right) \wedge \|\Pi_{\mathcal{M}}(Y/\mu_\epsilon) - \Pi_{\mathcal{M}}(x_i)\| + \Delta_{\mathcal{M}} \right) \\
&\leq \mu_\epsilon (K_{\mathcal{M}} + D_x) \left( \left( \tfrac{1}{2\tau_{\mathcal{M}}} \varepsilon_0^2 \right) \wedge \varepsilon_0 + \Delta_{\mathcal{M}} \right), &(30)
\end{aligned}$$

where in the second line, we use that

$$\begin{aligned}
\|\tilde{Y} - \Pi_{\mathcal{M}^\epsilon}(Y)\| &= \mathbb{E}[\|Y - \Pi_{\mathcal{M}^\epsilon}(Y)\|^2]^{1/2} \\
&\leq \mathbb{E}\left[ \left| \|Y - \Pi_{\mathcal{M}^\epsilon}(Y)\| - \|x - \Pi_{\mathcal{M}^\epsilon}(x)\| \right|^2 \right]^{1/2} + \|x - \Pi_{\mathcal{M}^\epsilon}(x)\| \\
&\leq K_{\mathcal{M}} + D_x.
\end{aligned}$$

Similarly, the fourth term of (28) is bounded by,

$$- \langle \tilde{Y} - \Pi_{\mathcal{M}^\epsilon}(Y), \Pi_{\mathcal{M}^\epsilon}(Y) - \mu_\epsilon x_j \rangle$$
$$\leq \mu_\epsilon (K_{\mathcal{M}} + D_x)\Big(\big(\tfrac{1}{2\tau_{\mathcal{M}}}\|\Pi_{\mathcal{M}}(Y/\mu_\epsilon) - \Pi_{\mathcal{M}}(x_j)\|^2\big) \wedge \|\Pi_{\mathcal{M}}(Y/\mu_\epsilon) - \Pi_{\mathcal{M}}(x_j)\| + \Delta_{\mathcal{M}}\Big),$$
$$(31)$$

and finally, the third term of (28) is controlled using Young's inequality to obtain,

$$\|\Pi_{\mathcal{M}^\epsilon}(Y) - \mu_\epsilon x_j\|^2 = \|\Pi_{\mathcal{M}^\epsilon}(Y) - \Pi_{\mathcal{M}^\epsilon}(\mu_\epsilon x_j)\|^2 + \|\Pi_{\mathcal{M}^\epsilon}(\mu_\epsilon x_j) - \mu_\epsilon x_j\|^2$$
$$+ 2\langle \Pi_{\mathcal{M}^\epsilon}(Y) - \Pi_{\mathcal{M}^\epsilon}(\mu_\epsilon x_j), \Pi_{\mathcal{M}^\epsilon}(\mu_\epsilon x_j) - \mu_\epsilon x_j \rangle$$
$$\geq \frac{3}{4}\|\Pi_{\mathcal{M}^\epsilon}(Y) - \Pi_{\mathcal{M}^\epsilon}(\mu_\epsilon x_j)\|^2 - 3\|\Pi_{\mathcal{M}^\epsilon}(\mu_\epsilon x_j) - \mu_\epsilon x_j\|^2$$
$$\geq \frac{3}{4}\|\Pi_{\mathcal{M}^\epsilon}(Y) - \Pi_{\mathcal{M}^\epsilon}(\mu_\epsilon x_j)\|^2 - 3\mu_\epsilon^2 \Delta_{\mathcal{M}}^2, \qquad (32)$$

Thus, substituting (30), (31) and (32) in to (28) leads to the bound,

$$\|\tilde{Y} - \mu_\epsilon x_i\|^2 - \|\tilde{Y} - \mu_\epsilon x_j\|^2$$
$$\leq \mu_\epsilon^2 (\varepsilon_0 + \Delta_{\mathcal{M}})^2 + 2\mu_\epsilon (K_{\mathcal{M}} + D_x)\Big(\big(\tfrac{1}{2\tau_{\mathcal{M}}}\varepsilon_0^2\big) \wedge \varepsilon_0 + \Delta_{\mathcal{M}}\Big)$$
$$- \frac{3}{4}\|\Pi_{\mathcal{M}^\epsilon}(Y) - \mu_\epsilon \Pi_{\mathcal{M}}(x_j)\|^2 + 3\mu_\epsilon^2 \Delta_{\mathcal{M}}^2$$
$$+ 2\mu_\epsilon (K_{\mathcal{M}} + D_x)\Big(\big(\tfrac{1}{2\tau_{\mathcal{M}}}\|\Pi_{\mathcal{M}}(Y/\mu_\epsilon) - \mu_\epsilon \Pi_{\mathcal{M}}(x_j)\|^2\big) \wedge \|\Pi_{\mathcal{M}}(Y/\mu_\epsilon) - \Pi_{\mathcal{M}}(x_j)\| + \Delta_{\mathcal{M}}\Big)$$
$$\leq \mu_\epsilon \Big(2\mu_\epsilon + \tfrac{1}{\tau_{\mathcal{M}}}K_{\mathcal{M}} + \tfrac{1}{\tau_{\mathcal{M}}}D_x\Big)\varepsilon_0^2 + \mu_\epsilon \Big(5\mu_\epsilon \Delta_{\mathcal{M}} + 4K_{\mathcal{M}} + 4D_x\Big)\Delta_{\mathcal{M}}$$
$$- \frac{3}{4}\|\Pi_{\mathcal{M}^\epsilon}(Y) - \mu_\epsilon \Pi_{\mathcal{M}}(x_j)\|^2$$
$$+ 2(K_{\mathcal{M}} + D_x)\big(\tfrac{1}{2\mu_\epsilon \tau_{\mathcal{M}}}\|\Pi_{\mathcal{M}^\epsilon}(Y) - \mu_\epsilon \Pi_{\mathcal{M}}(x_j)\|^2\big) \wedge \|\Pi_{\mathcal{M}^\epsilon}(Y) - \mu_\epsilon \Pi_{\mathcal{M}}(x_j)\|. \qquad (33)$$

Continuing with bounding the contents of the exponential function in (27), we next control $|\Delta_i| + 2|\Delta_j|$, using Lemma E.1 to obtain,

$$|\Delta_i| + 2|\Delta_j| \leq 2\mu_\epsilon |\zeta|\Big(\tfrac{1}{2\tau_{\mathcal{M}}}\varepsilon_0^2 + 2\big(\tfrac{1}{2\tau_{\mathcal{M}}}\|\Pi_{\mathcal{M}}(Y/\mu_\epsilon) - \Pi_{\mathcal{M}}(x_j)\|^2\big) \wedge \|\Pi_{\mathcal{M}}(Y/\mu_\epsilon) - \Pi_{\mathcal{M}}(x_j)\| + 3\Delta_{\mathcal{M}}\Big). \qquad (34)$$

Therefore, combining (33) and (34), we obtain the bound,

$$\|\tilde{Y} - \mu_\epsilon x_i\|^2 - \|\tilde{Y} - \mu_\epsilon x_j\|^2 + |\Delta_i| + 2|\Delta_j|$$
$$\leq \mu_\epsilon \Big(2\mu_\epsilon + \tfrac{1}{\tau_{\mathcal{M}}}(|\zeta| + K_{\mathcal{M}} + D_x)\Big)\varepsilon_0^2 + \mu_\epsilon \Big(5\mu_\epsilon \Delta_{\mathcal{M}} + 6|\zeta| + 4K_{\mathcal{M}} + 4D_x\Big)\Delta_{\mathcal{M}}$$
$$- \frac{3}{4}\|\Pi_{\mathcal{M}^\epsilon}(Y) - \mu_\epsilon \Pi_{\mathcal{M}}(x_j)\|^2$$
$$+ 2(K_{\mathcal{M}} + D_x + 2|\zeta|)\big(\tfrac{1}{2\mu_\epsilon \tau_{\mathcal{M}}}\|\Pi_{\mathcal{M}^\epsilon}(Y) - \mu_\epsilon \Pi_{\mathcal{M}}(x_j)\|^2\big) \wedge \|\Pi_{\mathcal{M}^\epsilon}(Y) - \mu_\epsilon \Pi_{\mathcal{M}}(x_j)\|$$
$$\leq \mu_\epsilon \Big(2\mu_\epsilon + \tfrac{1}{\tau_{\mathcal{M}}}(|\zeta| + K_{\mathcal{M}} + D_x)\Big)\varepsilon_0^2 + \mu_\epsilon \Big(5\mu_\epsilon \Delta_{\mathcal{M}} + 6|\zeta| + 4K_{\mathcal{M}} + 4D_x\Big)\Delta_{\mathcal{M}}$$
$$- \frac{1}{2}\|\Pi_{\mathcal{M}^\epsilon}(Y) - \mu_\epsilon \Pi_{\mathcal{M}}(x_j)\|^2$$
$$+ 2\mathbb{1}_{\mathcal{E}_x}(K_{\mathcal{M}} + D_x + 2|\zeta|)\|\Pi_{\mathcal{M}^\epsilon}(Y) - \mu_\epsilon \Pi_{\mathcal{M}}(x_j)\|,$$

where the indicator function contains the event $\mathcal{E}_x = \{K_{\mathcal{M}} + D_x + 2|\zeta| \geq \mu_\epsilon \tau_{\mathcal{M}}/4\}$.

Using this bound, we choose a value of $\varepsilon_1$ that guarantees that the contents of the exponential function in (27) is negative. By solving the quadratic, it follows that to have $\|\tilde{Y} - \mu_\epsilon x_i\|^2 - \|\tilde{Y} - \mu_\epsilon x_j\|^2 + |\Delta_i| + 2|\Delta_j| \leq -\mu_\epsilon \kappa$, for some $\kappa > 0$, it is sufficient to have $\|\Pi_{\mathcal{M}^\epsilon}(Y) - \Pi_{\mathcal{M}^\epsilon}(\mu_\epsilon x_j)\| \geq \mu_\epsilon \varepsilon_1$ with,

$$\mu_\epsilon^2 \varepsilon_1^2 = 6\mu_\epsilon \kappa + 6\mu_\epsilon \Big(2\mu_\epsilon + \tfrac{1}{\tau_{\mathcal{M}}}(|\zeta| + K_{\mathcal{M}} + D_x)\Big)\varepsilon_0^2 + 6\mu_\epsilon \Big(5\mu_\epsilon \Delta_{\mathcal{M}} + 6|\zeta| + 4K_{\mathcal{M}} + 4D_x\Big)\Delta_{\mathcal{M}}$$
$$+ 8\mathbb{1}_{\mathcal{E}_x}(K_{\mathcal{M}} + D_x + 2|\zeta|)^2. \qquad (35)$$

Substituting this into (27), we then obtain the following bound for B,

$$\mathbb{E}[\mathsf{B}|S] \leq \mathbb{E}\left[\frac{|I_1^{\mathsf{C}}|}{|I_0|}\bigg|S\right] \exp(-\mu_\epsilon \kappa / 2\sigma_\epsilon^2).$$

We now return to bounding A with this choice of $\varepsilon_1$ to obtain that,

$$\mathbb{E}[\mathtt{A}|S] \leq \frac{\mu_\epsilon K_\mathcal{M}}{\sigma_\epsilon^2}\left(\frac{1}{\tau_\mathcal{M}\mu_\epsilon^2}\left(6\mu_\epsilon\kappa + 6\mu_\epsilon\left(2\mu_\epsilon + \tfrac{3}{\tau_\mathcal{M}}K_\mathcal{M} + \tfrac{1}{\tau_\mathcal{M}}D_x\right)\varepsilon_0^2\right.\right.$$
$$+ 6\mu_\epsilon\left(5\mu_\epsilon\Delta_\mathcal{M} + 16K_\mathcal{M} + 4D_x\right)\Delta_\mathcal{M}$$
$$\left.\left.+ 8\mathbb{P}(\mathcal{E}_x|S)^{1/2}(5K_{\max,\mathcal{M}}^{1/2}K_\mathcal{M}^{1/2} + D_x)^2\right) + 2\Delta_\mathcal{M}\right),$$

where we utilise the bounds in Lemma E.2 to control $|\zeta|$. We then optimise $\kappa$ by choosing,

$$\kappa = \frac{2\sigma_\epsilon^2}{\mu_\epsilon}\log\left(\mathbb{E}\left[|I_1^\complement|/|I_0|\Big|S\right]\frac{\tau_\mathcal{M}}{K_\mathcal{M}}\right)_+,$$

which produces the bound,

$$\mathbb{E}[|\Delta\,\mathrm{LSE}(x)|S] \leq \frac{K_\mathcal{M}}{\tau_\mathcal{M}}\left(1 + 12\log\left(\mathbb{E}\left[|I_1^\complement|/|I_0|\Big|S\right]\frac{\tau_\mathcal{M}}{K_\mathcal{M}}\right)_+ + \left(2 + \tfrac{3}{\tau_\mathcal{M}}K_\mathcal{M} + \tfrac{1}{\tau_\mathcal{M}}D_x\right)\frac{6\varepsilon_0^2}{\sigma_\epsilon^2}\right.$$
$$\left.+ \left(5\Delta_\mathcal{M} + 16K_\mathcal{M} + 4D_x\right)\frac{6\Delta_\mathcal{M}}{\sigma_\epsilon^2} + \frac{8\mathbb{P}(\mathcal{E}_x|S)^{1/2}}{\sigma_\epsilon^2}(5K_{\max,\mathcal{M}}^{1/2}K_\mathcal{M}^{1/2} + D_x)^2\right)\right)$$
$$+ \frac{2K_\mathcal{M}\Delta_\mathcal{M}}{\sigma_\epsilon^2},$$

where we have used that $\mu_\epsilon \leq 1$ to simplify the expression.

To conclude the proof, we use the fact that $|I_0| = N\hat{\mu}_{\mathrm{data}}(B_{\varepsilon_0}(\Pi_\mathcal{M}(Y/\mu_\epsilon))$ and $|I_1^\complement| \leq N$. Then, optimising over $\varepsilon_0$ leads to the bound in the statement. $\qquad\square$

## E.2 Manifold concentration under log-domain smoothing

The bound on $\Delta\,\mathrm{LSE}_\mathcal{M}$ developed in the previous section depends on the distance to the manifold, $D_x$. Since, to control the Rényi divergence, we must integrate $\Delta\,\mathrm{LSE}_\mathcal{M}$ with respect to $\hat{p}_\epsilon^{k^\mathcal{M}}$, we must develop some bounds on the concentration of this measure to the manifold. Due to the complexity of log-domain smoothing, this is non-trivial and relies on Weyl's formula for the volume of tubular neighbourhoods. In the following lemma, we develop such a concentration inequality.

**Lemma E.4.** *Let $\delta, \varepsilon > 0$ such that $\operatorname{ess\,inf}_{x\in\mathcal{M}}(\Pi_\mathcal{M})_*\hat{\mu}_{\mathrm{data}}(B_\varepsilon(x)) \geq \delta$, then for all $r^2 \geq 2\sigma_\epsilon^2 d$, we obtain the bound,*

$$\mathbb{P}_{Z\sim\hat{p}_\epsilon^{k^\mathcal{M}}}(\mathrm{dist}(Z,\mathcal{M}^\epsilon) \geq r|S) \leq \exp\left(d\log(8) + \frac{5(K_\mathcal{M}^2 + \mu_\epsilon^2\Delta_\mathcal{M})^2 + 4\mu_\epsilon^2\varepsilon^2}{2\sigma_\epsilon^2}\right.$$
$$\left.+ 2\delta^{-1} - \frac{(r - \sqrt{2\sigma_\epsilon^2 d})^2}{4\sigma_\epsilon^2}\right).$$

*Proof.* We begin by expressing the probability in integral form,

$$\mathbb{P}(\mathrm{dist}(Z,\mathcal{M}^\epsilon) \geq r) = C_\mathcal{M}^{-1}\int_{\mathrm{dist}(\cdot,\mathcal{M})\geq r}\exp\left(\int\log\hat{p}_\epsilon(y)k_x^\mathcal{M}(dy)\right)dx.$$

From the formulation of $\log\hat{p}_\epsilon$ in (9), we readily obtain that

$$\log\hat{p}_\epsilon(y) \leq -\frac{\min_i\|y - \mu_\epsilon x_i\|^2}{2\sigma_\epsilon^2} + \log(N) + C_\epsilon$$
$$\leq -\frac{\min_i\{(\|y - \mu_\epsilon\Pi_\mathcal{M}(x_i)\| - \|\mu_\epsilon\Pi_\mathcal{M}(x_i) - \mu_\epsilon x_i\|)_+\}^2}{2\sigma_\epsilon^2} + \log(N) + C_\epsilon$$
$$\leq -\frac{(\mathrm{dist}(y,\mathcal{M}^\epsilon) - \mu_\epsilon\Delta_\mathcal{M})_+^2}{2\sigma_\epsilon^2} - \frac{d}{2}\log(2\pi\sigma_\epsilon^2), \tag{36}$$

Letting $Y \sim k_x^{\mathcal{M}}$, we use the fact that $Y \in \mathcal{M}_{r(x)}^{\epsilon}$ to obtain the following lower bound:

$$
\begin{aligned}
\mathrm{dist}(Y, M^{\epsilon}) & \\
&= \mathbb{E}_{Y' \sim k_x}[\mathrm{dist}(Y', \mathcal{M}^{\epsilon})^2]^{1/2} \\
&\geq \mathrm{dist}(x, \mathcal{M}^{\epsilon}) - \mathbb{E}_{Y' \sim k_x}[(\mathrm{dist}(Y', \mathcal{M}^{\epsilon}) - \mathrm{dist}(x, \mathcal{M}^{\epsilon}))^2]^{1/2} \\
&\geq \mathrm{dist}(x, \mathcal{M}^{\epsilon}) - K_{\mathcal{M}}.
\end{aligned} \tag{37}
$$

Thus, combining (36) and (37), we obtain the bound,

$$
\begin{aligned}
\int_{\mathrm{dist}(\cdot, \mathcal{M}) \geq r} & \exp \left( \int \log \hat{p}_{\epsilon}(y) k_x^{\mathcal{M}}(dy) \right) dx \\
&\leq (2\pi\sigma_{\epsilon}^2)^{-d/2} \int_{\mathrm{dist}(\cdot, \mathcal{M}) \geq r} \exp \left( -\frac{(\mathrm{dist}(x, \mathcal{M}^{\epsilon}) - \mu_{\epsilon}\Delta_{\mathcal{M}} - K_{\mathcal{M}})_+^2}{2\sigma_{\epsilon}^2} \right) dx \\
&\leq (2\pi\sigma_{\epsilon}^2)^{-d/2} \int_{\mathrm{dist}(\cdot, \mathcal{M}) \geq r} \exp \left( -\frac{\mathrm{dist}(x, \mathcal{M}^{\epsilon})^2}{4\sigma_{\epsilon}^2} + \frac{(K_{\mathcal{M}} + \mu_{\epsilon}\Delta_{\mathcal{M}})^2}{2\sigma_{\epsilon}^2} \right) dx. \quad (38)
\end{aligned}
$$

Next, we lower bound $C_{\mathcal{M}}$. For this, we use Lemma D.1 with the parameters,

$$
\epsilon_i = -\frac{(\mathrm{dist}(y, \mathcal{M}^{\epsilon}) - \|y - \mu_{\epsilon} x_i\|)^2}{\sigma_{\epsilon}^2},
$$

which produces the expression,

$$
\begin{aligned}
\log \hat{p}_{\epsilon}(y) &\geq \mathrm{LSE} \left( \left\{ -\frac{(\mathrm{dist}(y, \mathcal{M}^{\epsilon}) + (\|y - \mu_{\epsilon}\Pi_{\mathcal{M}}(x_i)\| - \mathrm{dist}(y, \mathcal{M}^{\epsilon})) + \mu_{\epsilon}\Delta_{\mathcal{M}})^2}{2\sigma_{\epsilon}^2} \right\}_i \right) + C_{\epsilon} \\
&\geq -\frac{(\mathrm{dist}(y, \mathcal{M}^{\epsilon}) + \mu_{\epsilon}\Delta_{\mathcal{M}})^2}{\sigma_{\epsilon}^2} + \int_0^1 \frac{\sum_{i=1}^N \exp(r\epsilon_i)\epsilon_i}{\sum_{i=1}^N \exp(r\epsilon_i)} + \log(N) + C_{\epsilon}.
\end{aligned}
$$

We control this further using a similar technique to that used in the proof of Theorem 3.6. We define the sets $I_0 = \{i \in [N] : \epsilon_i \geq -\mu_{\epsilon}^2 \varepsilon^2 / \sigma_{\epsilon}^2\}, I_1 = \{i \in [N] : \epsilon_i \geq -2\mu_{\epsilon}^2 \varepsilon^2 / \sigma_{\epsilon}^2\}$ for any quantity $\varepsilon > 0$. With this, we obtain the following bound,

$$
\begin{aligned}
\log \hat{p}_{\epsilon}(y) &= -\frac{(\mathrm{dist}(y, \mathcal{M}^{\epsilon}) + \mu_{\epsilon}\Delta_{\mathcal{M}})^2}{\sigma_{\epsilon}^2} + \int_0^1 \frac{\sum_{i \in I_1} \exp(r\epsilon_i)\epsilon_i}{\sum_{i \in I_1} \exp(r\epsilon_i)} dr + \int_0^1 \frac{\sum_{i \in I_1^\complement} \exp(r\epsilon_i)\epsilon_i}{\sum_{i \in I_0} \exp(r\epsilon_i)} dr \\
&\quad - \frac{d}{2} \log(2\pi\sigma_{\epsilon}^2) \\
&\geq -\frac{(\mathrm{dist}(y, \mathcal{M}^{\epsilon}) + \mu_{\epsilon}\Delta_{\mathcal{M}})^2}{\sigma_{\epsilon}^2} + \min_{i \in I_1} \epsilon_i + \frac{|I_1^\complement|}{|I_0|} \int_0^1 \min_{i \in I_1^\complement, j \in I_0} \exp(r(\epsilon_i - \epsilon_j))\epsilon_i dr \\
&\quad - \frac{d}{2} \log(2\pi\sigma_{\epsilon}^2) \\
&\geq -\frac{(\mathrm{dist}(y, \mathcal{M}^{\epsilon}) + \mu_{\epsilon}\Delta_{\mathcal{M}})^2}{\sigma_{\epsilon}^2} - \frac{2\mu_{\epsilon}^2\varepsilon^2}{\sigma_{\epsilon}^2} - \frac{|I_1^\complement|}{|I_0|} \int_0^1 \exp \left( -\frac{r\mu_{\epsilon}^2\varepsilon^2}{\sigma_{\epsilon}^2} \right) \frac{2\mu_{\epsilon}^2\varepsilon^2}{\sigma_{\epsilon}^2} dr \\
&\quad - \frac{d}{2} \log(2\pi\sigma_{\epsilon}^2).
\end{aligned}
$$

This is further controlled using the fact that $I_0 \supseteq \{i \in [N] : \|\Pi_{\mathcal{M}}(x_i) - \Pi_{\mathcal{M}}(y/\mu_{\epsilon})\|^2 \leq \varepsilon^2\}$ and $|I_1^\complement| \leq N$ and hence

$$
\frac{|I_1^\complement|}{|I_0|} \leq \frac{N}{|\{\Pi_{\mathcal{M}}(x_i)\}_{i=1}^N \cap B_{\varepsilon}(\Pi_{\mathcal{M}}(y/\mu_{\epsilon}))|} \leq \hat{\mu}_{\mathrm{data}}(B_{\varepsilon}(\Pi_{\mathcal{M}}(y/\mu_{\epsilon})))^{-1} \leq \delta^{-1},
$$

where the last inequality holds almost surely. In combination with the bound $\mathbb{E}[\mathrm{dist}(Z, M^{\epsilon})^2]^{1/2} \leq \mathrm{dist}(x, \mathcal{M}^{\epsilon}) + K_{\mathcal{M}}$, we obtain that,

$$
\int \log \hat{p}_{\epsilon}(y) k_x^{\mathcal{M}}(dy) \geq -\frac{(\mathrm{dist}(x, \mathcal{M}^{\epsilon}) + K_{\mathcal{M}} + \mu_{\epsilon}\Delta_{\mathcal{M}})^2}{\sigma_{\epsilon}^2} - \frac{2\mu_{\epsilon}^2\varepsilon^2}{\sigma_{\epsilon}^2} - 2\delta^{-1} - \frac{d}{2} \log(2\pi\sigma_{\epsilon}^2). \quad (39)
$$

With this, we lower bound $C_\mathcal{M}$ by,

$$C_\mathcal{M} = \int \exp\left(\int \log \hat{p}_\epsilon(y) k_x^\mathcal{M}(dy)\right) dx$$

$$\geq (2\pi\sigma_\epsilon^2)^{-d/2} \int \exp\left(-\frac{2\operatorname{dist}(x,\mathcal{M}^\epsilon)^2 + 2(K_\mathcal{M} + \mu_\epsilon\Delta_\mathcal{M})^2}{\sigma_\epsilon^2} - \frac{2\mu_\epsilon^2\varepsilon^2}{\sigma_\epsilon^2} - 2\delta^{-1}\right) dx. \tag{40}$$

Before combining the bounds in (38) and (40) we further simplify their expressions using Weyl's formula (see Section D.4). Combining the bound in (38) with the integral formula in (20), we obtain the bound,

$$\int_{\operatorname{dist}(\cdot,\mathcal{M})\geq r} \exp\left(\int \log \hat{p}_\epsilon(y) k_x^\mathcal{M}(dy)\right) dx$$

$$\leq (2\pi\sigma_\epsilon^2)^{-d/2} \exp\left(\frac{(K_\mathcal{M} + \mu_\epsilon\Delta_\mathcal{M})^2}{2\sigma_\epsilon^2}\right) \sum_{p=0}^{\lfloor d^*/2\rfloor} (d - d^* + 2p)\tilde{k}_{2p}(\mathcal{M}^\epsilon) \int_r^\infty e^{-\frac{s^2}{4\sigma_\epsilon^2}} s^{d-d^*-1+2p} ds,$$

where we use the shorthand $d^* = d_\mathcal{M}^*$. The integral on the right-hand side can be analysed by relating it to the measure of a related spherically symmetric measure (e.g. see equation (4) in Bobkov, 2003). With this we relate the integral to the concentration of a Gaussian random variable:

$$\int_r^\infty s^{k-1} \exp(-s^2/4\sigma_\epsilon^2) ds = \frac{\Gamma(k/2+1)}{k\pi^{k/2}} \int_{\{x\in\mathbb{R}^k: \|x\|\geq r\}} \exp(-\|x\|^2/4\sigma_\epsilon^2) dx$$

$$= \frac{\Gamma(k/2+1)(4\sigma_\epsilon^2)^{k/2}}{k} \mathbb{P}_{\xi\sim N(0,2\sigma_\epsilon^2 I_k)}(\|\xi\| \geq r).$$

Thus, we obtain the expression,

$$\int_{\operatorname{dist}(\cdot,\mathcal{M})\geq r} \exp\left(\int \log \hat{p}_\epsilon(y) k_x^\mathcal{M}(dy)\right) dx$$

$$= (2\pi\sigma_\epsilon^2)^{-d/2} \exp\left(\frac{(K_\mathcal{M}+\mu_\epsilon\Delta_\mathcal{M})^2}{2\sigma_\epsilon^2}\right) \sum_{p=0}^{\lfloor d^*/2\rfloor} w_p \mathbb{P}_{\xi\sim N(0,2\sigma_\epsilon^2 I_{d-d^*+2p})}(\|\xi\| \geq r),$$

$$w_p := \frac{\Gamma((d-d^*+2p)/2+1)(4\sigma_\epsilon^2)^{(d-d^*+2p)/2}}{d-d^*+2p}(d-d^*+2p)\tilde{k}_{2p}(\mathcal{M}^\epsilon).$$

By a similar argument, we also obtain

$$C_\mathcal{M} \geq (2\pi\sigma_\epsilon^2)^{-d/2} \exp\left(-\frac{4(K_\mathcal{M}+\mu_\epsilon\Delta_\mathcal{M})^2 + 4\mu_\epsilon^2\varepsilon^2}{2\sigma_\epsilon^2} - 2\delta^{-1}\right) \sum_{p=0}^{\lfloor d^*/2\rfloor} 64^{-(d-d^*+2p)/2} w_p.$$

Dividing the two, we obtain the bound,

$$\mathbb{P}_{Z\sim\hat{p}_\epsilon^{k,\mathcal{M}}}(\operatorname{dist}(Z,\mathcal{M}^\epsilon) \geq r)$$

$$\leq \frac{\sum_{p=0}^{\lfloor d^*/2\rfloor} w_p \mathbb{P}_{\xi\sim N(0,2\sigma_\epsilon^2 I_{d-d^*+2p})}(\|\xi\| \geq r)}{\sum_{p=0}^{\lfloor d^*/2\rfloor} 64^{-(d-d^*+2p)/2} w_p} \exp\left(\frac{5(K_\mathcal{M}^2+\mu_\epsilon^2\Delta_\mathcal{M})^2}{2\sigma_\epsilon^2} + \frac{2\mu_\epsilon^2\varepsilon^2}{\sigma_\epsilon^2} + 2\delta^{-1}\right)$$

$$\leq 8^d \exp\left(\frac{5(K_\mathcal{M}^2+\mu_\epsilon^2\Delta_\mathcal{M})^2}{2\sigma_\epsilon^2} + \frac{2\mu_\epsilon^2\varepsilon^2}{\sigma_\epsilon^2} + 2\delta^{-1}\right) \max_{p\in\{0,\ldots,\lfloor d^*/2\rfloor\}} \mathbb{P}_{\xi\sim N(0,2\sigma_\epsilon^2 I_{d-d^*+2p})}(\|\xi\| \geq r).$$

We then bound this further using the concentration of the chi-squared distribution (see Example 2.28 of (Wainwright, 2019)), obtaining,

$$\mathbb{P}_{\xi\sim N(0,2\sigma_\epsilon^2 I_k)}(\|\xi\| \geq r) \leq \exp\left(-\frac{(r-\sqrt{2\sigma_\epsilon^2 k})_+^2}{4\sigma_\epsilon^2 d}\right)$$

$$\leq \exp\left(-\frac{(r-\sqrt{2\sigma_\epsilon^2 d})_+^2}{4\sigma_\epsilon^2 d}\right),$$

completing the proof of the bound. $\qquad\square$

## E.3 Proof of Theorem 3.6

With the pointwise bound on $\mathbb{E}[\Delta \operatorname{LSE}_{\mathcal{M}}(x)|S]$ from the previous subsections, we are now prepared to derive the Rényi divergence bound in Theorem 3.6.

*Proof of Theorem 3.6.* To bound the Rényi divergence, we begin with the following expression which follows from (23):

$$
D_q(\hat{p}_\epsilon^{k^{\mathcal{M}}} \| \hat{p}_\epsilon^k) = \frac{1}{q-1} \log \int \left( \frac{\hat{p}_\epsilon^{k^{\mathcal{M}}}(x)}{\hat{p}_\epsilon^k(x)} \right)^{q-1} \hat{p}_\epsilon^{k^{\mathcal{M}}}(dx)
$$

$$
= \frac{1}{q-1} \log \int \exp((q-1)\mathbb{E}[\Delta \operatorname{LSE}_{\mathcal{M}}(x)|S]) \, \hat{p}_\epsilon^{k^{\mathcal{M}}}(dx) + \log(C^{\mathcal{M}}/C),
$$

where the normalisation constants, $C$ and $C^{\mathcal{M}}$, are as defined in (22). Furthermore, we obtain the following relationship between the normalisation constants:

$$
C = \int \exp \left( \int \log \hat{p}_\epsilon(y) k_x(dy) - \int \log \hat{p}_\epsilon(y) k_x^{\mathcal{M}}(dy) + \int \log \hat{p}_\epsilon(y) k_x^{\mathcal{M}}(dy) \right) dx
$$

$$
= C^{\mathcal{M}} \int \exp(\mathbb{E}[\Delta \operatorname{LSE}_{\mathcal{M}}(x)|S]) \hat{p}_\epsilon^{k^{\mathcal{M}}}(dx).
$$

Therefore, using Jensen's inequality, we deduce the bound,

$$
D_q(\hat{p}_\epsilon^k \| \hat{p}_\epsilon^{k^{\mathcal{M}}}) \leq \frac{2}{\beta} \log \int \exp(\beta|\mathbb{E}[\Delta \operatorname{LSE}_{\mathcal{M}}(x)|S]|) \hat{p}_\epsilon^{k^{\mathcal{M}}}(dx)
$$

$$
= \frac{2}{\beta} \log \mathbb{E}[\exp(\beta|\mathbb{E}[\Delta \operatorname{LSE}_{\mathcal{M}}(Z)|S, Z]|)|S], \tag{41}
$$

where, for the sake of brevity, we use the shorthand $\beta = (q-1) \vee 1$ and $Z \sim \hat{p}_\epsilon^{k^{\mathcal{M}}}(dx)$.

We proceed by applying the bound on $\Delta \operatorname{LSE}_{\mathcal{M}}$ developed in Lemma E.3. The assumptions of Lemma E.3 hold with $\Delta_{\mathcal{M}} = 0$ and hence, we have the bound,

$$
\mathbb{E}[|\Delta \operatorname{LSE}_{\mathcal{M}}(Z)|S,Z] \leq \frac{8K}{\tau} \left( 1 + 2\log\left(\frac{\tau}{K}\right)_+ + \frac{(5K_{\max}^{1/2} K^{1/2} + D_Z)^2}{\sigma_\epsilon^2} \mathbb{1}_{5K_{\max}+D_Z \geq \mu_\epsilon \tau/4} \right.
$$

$$
\left. + 2 \inf_{\varepsilon_0 > 0} \left\{ \log \left( \mathbb{E}[\hat{\mu}_{\mathrm{data}}(B_{\varepsilon_0}(\Pi_{\mathcal{M}}(Y/\mu_\epsilon)))^{-1}|S, Z] \right)_+ + \left(1 + \frac{K+D_Z}{\tau}\right)\frac{\varepsilon_0^2}{\sigma_\epsilon^2} \right\} \right), \tag{42}
$$

where we have used the fact that $|\zeta| \leq 2K_{\max}$ (see Lemma E.2) and therefore, $\mathbb{P}(\mathcal{E}_Z|S, Z)^{1/2} \leq \mathbb{1}_{5K_{\max}+D_Z \geq \mu_\epsilon \tau/4}$. To control the infimum term, we utilise the bound on balls of $\hat{\mu}_{\mathrm{data}}$ given in Lemma D.7. In this lemma, it is shown that whenever $\varepsilon_0 \leq \tau$, with probability $1 - \delta$, we have,

$$
\sup_{y \in \mathcal{M}} \hat{\mu}_{\mathrm{data}}(B_{\varepsilon_0}(y))^{-1} \leq c_\mu^{-1}\varepsilon_0^{-d^*}/4, \tag{43}
$$

once $N$ is sufficiently large so that the condition in (18) is satisfied with $r = \varepsilon_0$. If we set $\varepsilon_0^2 = d^*\sigma_\epsilon^2$ and require that $\sigma_\epsilon^2 \leq (\tau/64)^2/d^*$, then once $N$ is sufficiently large so that (18) is satisfied, we obtain the bound,

$$
\inf_{\varepsilon_0 > 0} \left\{ \log \left( \mathbb{E}_{Y \sim k_Z}[\hat{\mu}_{\mathrm{data}}(B_{\varepsilon_0}(\Pi_{\mathcal{M}}(Y/\mu_\epsilon)))^{-1}|S, Z] \right)_+ + \left(1 + \frac{K+D_Z}{\tau}\right)\frac{\varepsilon_0^2}{\sigma_\epsilon^2} \right\}
$$

$$
\leq \inf_{\varepsilon_0 \in (0,\tau_{\mathcal{M}}/64]} \sup_{y \in \mathcal{M}} \left\{ \log \left( \hat{\mu}_{\mathrm{data}}(B_{\varepsilon_0}(y))^{-1} \right)_+ + \left(1 + \frac{K+D_Z}{\tau}\right)\frac{\varepsilon_0^2}{\sigma_\epsilon^2} \right\}
$$

$$
\lesssim \left(1 + \frac{K+D_Z}{\tau}\right)d^*. \tag{44}
$$

Indeed, this would require $N \gtrsim (d^* + 1)c_\mu^{-2}(d^*\sigma_\epsilon^2)^{-d^*}$. If $N$ does not satisfy this, then we instead set $\varepsilon_0$ to be the smallest $r$ such that (18) is satisfied. We have that such a quantity exists and satisfies $\varepsilon_0 \in (0, \tau/64]$ as soon as we assume that,

$$
N \geq 16\left(\left(144d^* \log(256/\tau) + 144\log(1/c_\mu)\right) \vee \tfrac{\log(\delta^{-1})}{2}\right)(\tau/128)^{-2d^*}c_\mu^{-2} =: N_{\min}(d^*, \tau, \delta, c_\mu). \tag{45}
$$

With this, we arrive at a quantity with $(c_\mu^2 N)^{-1/d^*} \lesssim \varepsilon_0^2 \lesssim (c_\mu^2 N)^{-1/d^*}$. Thus, we obtain the bound,

$$\inf_{\varepsilon_0 > 0} \left\{ \log\left( \mathbb{E}_{Y \sim k_Z}[\hat{\mu}_{\text{data}}(B_{\varepsilon_0}(\Pi_{\mathcal{M}}(Y/\mu_\epsilon)))^{-1}|S, Z]\right)_+ + \left(1 + \tfrac{K + D_Z}{\tau}\right)\tfrac{\varepsilon_0^2}{\sigma_\epsilon^2} \right\}$$

$$\lesssim 1 + \left(1 + \tfrac{K+D_Z}{\tau}\right) \tfrac{(c_\mu^2 N)^{-1/d^*}}{\sigma_\epsilon^2} \mathbb{1}_{d^* > 0}. \tag{46}$$

We can then combine the bounds in (44) and (46) to obtain that there exists a quantity $C_2 > 0$ that depends only logarithmically on structural parameters and satisfies,

$$2 \inf_{\varepsilon_0 > 0} \left\{ \log\left( \mathbb{E}[\hat{\mu}_{\text{data}}(B_{\varepsilon_0}(\Pi_{\mathcal{M}}(Y/\mu_\epsilon)))^{-1}|S, Z]\right)_+ + \left(1 + \tfrac{K + D_Z}{\tau}\right)\tfrac{\varepsilon_0^2}{\sigma_\epsilon^2} \right\}$$

$$\leq C_2\left(1 + \left(1 + \tfrac{D_Z}{\tau}\right)d^* \vee ((c_\mu^2 N)^{-\frac{1}{d^*}}\sigma_\epsilon^{-2})\right), \tag{47}$$

where we have also used the fact that $K \leq K_{\max} \leq \tau/96$.

Returning to bounding (41), we use (42) and (47) to derive the upper bound,

$$\frac{2}{\beta} \log \mathbb{E}[\exp(\beta|\mathbb{E}[\Delta\,\text{LSE}_{\mathcal{M}}(Z)|S, Z]|)|S]$$

$$\leq \frac{2}{\beta} \log \mathbb{E}\left[ \exp\left( \tfrac{8\beta K}{\tau}\left(1 + 2\log\left(\tfrac{\tau}{K}\right)_+ + \tfrac{(5K_{\max}^{1/2}K^{1/2} + D_Z)^2}{\sigma_\epsilon^2}\mathbb{1}_{5K_{\max} + D_Z \geq \mu_\epsilon \tau/4} + C_2 \right.\right.\right.$$

$$\left.\left.\left. + C_2\left(1 + \tfrac{D_Z}{\tau}\right)d^* \vee \tfrac{(c_\mu^2 N)^{-\frac{1}{d^*}}}{\sigma_\epsilon^2}\right)\right)\Big| S\right]$$

$$\leq \frac{16K}{\tau}\left(1 + 2\log\left(\tfrac{\tau}{K}\right) + C_2 + C_2 d^* \vee \tfrac{(c_\mu^2 N)^{-\frac{1}{d^*}}}{\sigma_\epsilon^2}\right)$$

$$+ \frac{1}{20\beta} \log \mathbb{E}\left[ \exp\left( \tfrac{40\beta K}{\tau}\tfrac{C_2 D_Z}{\tau}d^* \vee \tfrac{(c_\mu^2 N)^{-\frac{1}{d^*}}}{\sigma_\epsilon^2}\right)\Big| S\right]$$

$$+ \frac{1}{5\beta} \log \mathbb{E}\left[ \exp\left( \tfrac{10\beta K}{\tau}\tfrac{(5K^{1/2}K_{\max}^{1/2} + D_Z)^2}{\sigma_\epsilon^2}\mathbb{1}_{5K_{\max} + D_Z \geq \mu_\epsilon \tau/4}\right)\Big| S\right], \tag{48}$$

where in the second inequality, we use Hölder's inequality.

We now bound the last two terms, starting with the second. We do this by utilising Lemma E.4 which we apply with $\varepsilon = \tau/64$. As a result of (43) and the assumed lower bound on $N$, the assumptions of Lemma E.4 are satisfied with $\delta = c_\mu(\tau/64)^{d^*}/4$, and so, for any $r^2 \geq 4\sigma_\epsilon^2 d$,

$$\mathbb{P}(D_Z \geq r|S) \leq \exp\left(C - \tfrac{(r - \sqrt{2\sigma_\epsilon^2 d})^2}{4\sigma_\epsilon^2}\right), \quad C := \log(8)d + \frac{5K^2 + 2^{-10}\mu_\epsilon^2 \tau^2}{2\sigma_\epsilon^2} + 8c_\mu^{-1}(\tau/64)^{-d^*}.$$

Thus, for any $c, R > 0$, we have the bound,

$$\mathbb{E}\big[\exp\left(cD_Z\right)\big|S\big] = \int_0^\infty \mathbb{P}(D_Z \geq \log(r)/c)dr$$

$$\leq \exp(cR) + \int_{\exp(cR)}^\infty \mathbb{P}(D_Z \geq \log(r)/c)dr$$

$$\leq \exp(cR) + \int_{\exp(cR)}^\infty \exp\left( C - \tfrac{(\log(r)/c - \sqrt{2\sigma_\epsilon^2 d})^2}{4c^2\sigma_\epsilon^2}\right)dr.$$

This is simplified using the change of variables,

$$\int_{\exp(cR)}^\infty \exp\left( C - \tfrac{(\log(r)/c - \sqrt{2\sigma_\epsilon^2 d})^2}{4c^2\sigma_\epsilon^2}\right)dr$$

$$= c\int_0^\infty \exp\left( C - \tfrac{(u + R - \sqrt{2\sigma_\epsilon^2 d})^2}{4\sigma_\epsilon^2} + c(u + R)\right)du.$$

We then choose $R := \sqrt{2\sigma_\epsilon^2 d} + 16c\sigma_\epsilon^2 + \sqrt{256c^2\sigma_\epsilon^4 + 32\sigma_\epsilon^2 C}$ to further simplify the expression, obtaining,

$$\int_{\exp(cR)}^\infty \exp\left(C - \frac{(\log(r)/c - \sqrt{2\sigma_\epsilon^2 d})^2}{32c^2\sigma_\epsilon^2}\right) dr$$

$$= c\int_0^\infty \exp\left(-\frac{u^2}{32\sigma_\epsilon^2} + u\left(c - \frac{R}{16\sigma_\epsilon^2}\right) + c\sqrt{2\sigma_\epsilon^2 d}\right) du$$

$$\leq c\exp(c\sqrt{2\sigma_\epsilon^2 d})\int_0^\infty \exp\left(-\frac{u^2}{32\sigma_\epsilon^2}\right) du$$

$$= (32\pi)^{1/2}\exp(c\sqrt{2\sigma_\epsilon^2 d})c\sigma_\epsilon,$$

where the last line follows from the Gaussian integral. With this, we obtain a bound on the MGF:

$$\log \mathbb{E}\left[\exp\left(cD_Z\right)\big|S\right] \leq \log\left(\exp(cR) + (32\pi)^{1/2}\exp(c\sqrt{2\sigma_\epsilon^2 d})c\sigma_\epsilon\right)$$

$$\leq cR + (32\pi)^{1/2}c\sigma_\epsilon$$

$$\lesssim c\sqrt{\sigma_\epsilon^2 d} + c^2\sigma_\epsilon^2 + c(K + \tau) + c\sigma_\epsilon(\sqrt{d} + c_\mu^{-1/2}(\tau/64)^{-d^*/2}).$$

Substituting values for $c$ into the bound, we obtain,

$$\frac{1}{20\beta}\log\mathbb{E}\left[\exp\left(\frac{40\beta K}{\tau}\frac{C_2 D_Z}{\tau}d^* \vee \frac{(c_\mu^2 N)^{-\frac{1}{d^*}}}{\sigma_\epsilon^2}\right)\bigg|S\right]$$

$$\lesssim \sqrt{\sigma_\epsilon^2 d} + \frac{K}{\tau^2}\left(d^* \vee \frac{(c_\mu^2 N)^{-\frac{1}{d^*}}}{\sigma_\epsilon^2}\right)\left(\tau + \sigma_\epsilon(\sqrt{d} + c_\mu^{-1/2}(\tau/64)^{-d^*/2}) + \sigma_\epsilon^2\frac{K}{\tau^2}\left(d^* \vee \frac{(c_\mu^2 N)^{-\frac{1}{d^*}}}{\sigma_\epsilon^2}\right)\right)$$

$$\lesssim \sqrt{\sigma_\epsilon^2 d} + \frac{K}{\tau}\left(d^* \vee \frac{(c_\mu^2 N)^{-\frac{1}{d^*}}}{\sigma_\epsilon^2}\right)\left(1 + \frac{\sigma_\epsilon}{\tau}(\sqrt{d} + c_\mu^{-1/2}(\tau/64)^{-d^*/2}) + \frac{K}{\tau^3}\left((\sigma_\epsilon^2 d^*) \vee \tau^2\right)\right).$$

$$\tag{49}$$

Thus, as soon as we require that,

$$\sigma_\epsilon^2 \leq \left(\frac{K^2}{d\tau^2}\right) \wedge \frac{\tau^2}{(\sqrt{d} + c_\mu^{-1/2}(\tau/64)^{-d^*/2})^2} \wedge \left(\frac{\tau^2}{d^*}\right) =: \sigma_{\max,1}^2(d, d^*, \tau, K, c_\mu),$$

we obtain the bound,

$$\frac{1}{20\beta}\log\mathbb{E}\left[\exp\left(\frac{40\beta K}{\tau}\frac{C_2 D_Z}{\tau}d^* \vee \frac{(c_\mu^2 N)^{-\frac{1}{d^*}}}{\sigma_\epsilon^2}\right)\bigg|S\right] \lesssim \frac{K}{\tau}\left((d^* + 1) \vee \frac{(c_\mu^2 N)^{-\frac{1}{d^*}}}{\sigma_\epsilon^2}\right). \tag{50}$$

We now bound the third term of (48). We once again use the integral formula for the expectation to obtain,

$$\mathbb{E}\left[\exp\left(\frac{10\beta K}{\tau}\frac{(5K^{1/2}K_{\max}^{1/2} + D_Z)^2}{\sigma_\epsilon^2}\mathbb{1}_{5K_{\max} + D_Z \geq \mu_\epsilon\tau/4}\right)\bigg|S\right]$$

$$= \int_0^\infty \mathbb{P}\left(\exp\left(\frac{10\beta K}{\tau}\frac{(5K^{1/2}K_{\max}^{1/2} + D_Z)^2}{\sigma_\epsilon^2}\mathbb{1}_{5K_{\max} + D_Z \geq \mu_\epsilon\tau/4}\right) \geq r\bigg|S\right) dr$$

$$\leq \int_0^\infty \mathbb{P}\left((5K_{\max} + D_Z)^2\mathbb{1}_{5K_{\max} + D_Z \geq \mu_\epsilon\tau/4} \geq \frac{\sigma_\epsilon^2\tau}{10\beta K}\log(r)\bigg|S\right) dr$$

$$= 1 + \left(\exp\left(\frac{10\beta K}{\sigma_\epsilon^2\tau}(\mu_\epsilon\tau/4)^2\right) - 1\right)\mathbb{P}(D_Z \geq \mu_\epsilon\tau/4 - 5K_{\max}|S)$$

$$+ \int_{\exp\left(\frac{10\beta K}{\sigma_\epsilon^2\tau}(\mu_\epsilon\tau/4)^2\right)}^\infty \mathbb{P}\left((5K^{1/2}K_{\max}^{1/2} + D_Z)^2 \geq \frac{\sigma_\epsilon^2\tau}{10\beta K}\log(r)\bigg|S\right) dr. \tag{51}$$

We further bound the last term using $K \leq K_{\max}$ along with a change of variables, to derive,

$$\int_{\exp\left(\frac{10\beta K}{\sigma_\epsilon^2 \tau}(\mu_\epsilon\tau/4)^2\right)}^{\infty} \mathbb{P}\left((5K_{\max} + D_Z)^2 \geq \frac{\sigma_\epsilon^2 \tau}{10\beta K}\log(r)\Big|S\right)dr$$

$$= \frac{20\beta K}{\sigma_\epsilon^2 \tau}\int_{\mu_\epsilon\tau/4}^{\infty} \mathbb{P}(5K_{\max} + D_Z \geq u|S)\exp\left(\frac{10\beta K}{\sigma_\epsilon^2 \tau}u^2\right)u\,du$$

$$\leq \frac{20\beta K}{\sigma_\epsilon^2 \tau}\int_{\mu_\epsilon\tau/4}^{\infty} \exp\left(C - \frac{1}{4\sigma_\epsilon^2}\left(u - 5K_{\max} - \sqrt{2\sigma_\epsilon^2 d}\right)^2 + \frac{10\beta K}{\sigma_\epsilon^2 \tau}u^2\right)u\,du$$

Setting $c = 1 - \frac{40\beta K}{\tau} > 1/2$, we can simplify this expression by,

$$\frac{20\beta K}{\sigma_\epsilon^2 \tau}\int_{\mu_\epsilon\tau/4}^{\infty} \exp\left(C - \frac{1}{4\sigma_\epsilon^2}\left(u - 5K_{\max} - \sqrt{2\sigma_\epsilon^2 d}\right)^2 + \frac{10\beta K}{\sigma_\epsilon^2 \tau}u^2\right)u\,du$$

$$= \frac{20\beta K}{\sigma_\epsilon^2 \tau}\int_{\mu_\epsilon\tau/4}^{\infty} \exp\left(C - \frac{c}{4\sigma_\epsilon^2}\left(u - (5K_{\max} + \sqrt{2\sigma_\epsilon^2 d})c^{-1}\right)^2\right.$$

$$\left. + \frac{1 - c^{-1}}{4\sigma_\epsilon^2}(5K_{\max} + \sqrt{2\sigma_\epsilon^2 d})^2\right)u\,du$$

$$\leq \frac{20\beta K}{\sigma_\epsilon^2 \tau}(4\sigma_\epsilon^2 \pi/c)^{1/2}\exp\left(C - \frac{c}{4\sigma_\epsilon^2}\left(\mu_\epsilon\tau/4 - (5K_{\max} + \sqrt{2\sigma_\epsilon^2 d})c^{-1}\right)^2\right)\sqrt{2\sigma_\epsilon^2/c},$$

where the final inequality follows from the concentration of the Gaussian random variable. Using the fact that $K_{\max} \leq \tau/96$, $c^{-1} \leq 2$, and assuming that $\sigma_\epsilon^2 \leq \frac{K_{\max}^2}{2d}$,

$$\frac{20\beta K}{\sigma_\epsilon^2 \tau}\int_{\mu_\epsilon\tau/4}^{\infty} \exp\left(C - \frac{1}{4\sigma_\epsilon^2}\left(u - 5K_{\max} - \sqrt{2\sigma_\epsilon^2 d}\right)^2 + \frac{10\beta K}{\sigma_\epsilon^2 \tau}u^2\right)u\,du$$

$$\leq \frac{40\sqrt{2\pi}\beta K}{\tau c}\exp\left(C - \frac{1}{8\sigma_\epsilon^2}(\mu_\epsilon\tau/8)^2\right)$$

Similarly, we can bound the second term of (51), in total, obtaining,

$$\frac{1}{5\beta}\log\mathbb{E}\left[\exp\left(\frac{10\beta K}{\tau}\frac{(5K^{1/2}K_{\max}^{1/2} + D_Z)^2}{\sigma_\epsilon^2}\mathbb{1}_{5K^{1/2}K_{\max}^{1/2} + D_Z \geq \mu_\epsilon\tau/4}\right)\Big|S\right]$$

$$\lesssim \left(\exp\left(\frac{10\beta K}{\sigma_\epsilon^2 \tau}(\mu_\epsilon\tau/4)^2\right) - 1\right)\mathbb{P}(D_Z \geq \mu_\epsilon\tau/4 - 5K_{\max}|S)$$

$$+ \int_{\exp\left(\frac{10\beta K}{\sigma_\epsilon^2 \tau}(\mu_\epsilon\tau/4)^2\right)}^{\infty} \mathbb{P}\left((5K^{1/2}K_{\max}^{1/2} + D_Z)^2 \geq \frac{\sigma_\epsilon^2 \tau}{10\beta K}\log(r)\Big|S\right)dr$$

$$\lesssim \exp\left(C + \frac{10\beta K}{\sigma_\epsilon^2 \tau}(\mu_\epsilon\tau/4)^2 - \frac{1}{4\sigma_\epsilon^2}(3\mu_\epsilon\tau/16)^2\right) + \frac{K}{\tau}\exp\left(C - \frac{1}{8\sigma_\epsilon^2}(\mu_\epsilon\tau/8)^2\right)$$

$$\lesssim \exp\left(C - \frac{1}{8\sigma_\epsilon^2}(\mu_\epsilon\tau/8)^2\right).$$

Thus, by requiring that,

$$\sigma_\epsilon^2 \leq 2^{10}\left(\log(8)d + 8c_\mu^{-1}(\tau/64)^{-d^*}\right)\log(\tau/K) =: \sigma_{\max,2}^2(d^*, d, \tau, K, c_\mu),$$

we obtain the upper bound,

$$\frac{1}{5\beta}\log\mathbb{E}\left[\exp\left(\frac{10\beta K}{\tau}\frac{(5K^{1/2}K_{\max}^{1/2} + D_Z)^2}{\sigma_\epsilon^2}\mathbb{1}_{5K^{1/2}K_{\max}^{1/2} + D_Z \geq \mu_\epsilon\tau/4}\right)\Big|S\right] \lesssim \frac{K}{\tau}. \qquad (52)$$

Thus, by substituting (50) and (52) into (48) we obtain that, with a probability of at least $1 - \delta$, we have,

$$\frac{2}{\beta}\log\mathbb{E}[\exp(\beta|\mathbb{E}[\Delta\,\mathrm{LSE}_{\mathcal{M}}(Z)|S, Z]|)|S] \lesssim \frac{K}{\tau}\max\left\{d^* + 1, \frac{(c_\mu^2 N)^{-\frac{1}{d^*}}}{\sigma_\epsilon^2}\right\},$$

completing the proof. We collect the required upper bounds on $\sigma_\epsilon^2$, which when combined with the fact that $\sigma_\epsilon^2 \leq \epsilon$, leads to the sufficient condition,

$$\epsilon \leq \sigma_{\max,1}^2(d,d^*,\tau,K,c_\mu) \wedge \sigma_{\max,2}^2(d^*,d,\tau,K,c_\mu) \wedge \frac{K_{\max}^2}{2d} \wedge \frac{\tau^2}{(64)^2 d^*} =: \epsilon_{\max}(d,d^*,\tau,K,K_{\max},c_\mu).$$

(53)

$\square$

## E.4 Proof of Theorem 4.1

*Proof.* Let $\mathcal{M} \in \mathbb{M}_\mu$ and let $\tau_\mathcal{M}, d_\mathcal{M}^*, K_\mathcal{M}, K_{\max,\mathcal{M}}, \Delta_\mathcal{M}$ and $c_{\mu,\mathcal{M}}$ be as defined in Section 4.1 and assume that $\Delta_\mathcal{M} < \infty$ and $K_{\max,\mathcal{M}} \leq \tau/96$. Using the same argument that produced (42), we obtain the bound,

$$D_2(\hat{p}_\epsilon^{k^\mathcal{M}} \| \hat{p}_\epsilon^k) \leq 2\log \mathbb{E}[\exp(|\mathbb{E}[\Delta\,\mathrm{LSE}_\mathcal{M}(Z)|S,Z]|)|S],$$

where $Z \sim \hat{p}_\epsilon^{k^\mathcal{M}}(dx)$. Using Lemma E.3, we obtain the pointwise bound,

$$\mathbb{E}[|\Delta\,\mathrm{LSE}_\mathcal{M}(Z)|S,Z]$$

$$\leq \frac{8K_\mathcal{M}}{\tau_\mathcal{M}}\left(1 + 2\log\left(\tfrac{\tau_\mathcal{M}}{K_\mathcal{M}}\right) + \frac{(5K_{\max,\mathcal{M}}^{1/2}K_\mathcal{M}^{1/2} + D_Z)^2}{\sigma_\epsilon^2}\mathbb{1}_{D_Z \leq \mu_\epsilon\tau/4}\right.$$

$$+ \left(\Delta_\mathcal{M} + \tau_\mathcal{M} + K_\mathcal{M} + D_Z\right)\frac{12\Delta_\mathcal{M}}{\sigma_\epsilon^2}$$

$$\left.+ 2\inf_{\varepsilon_0 > 0}\left\{\log\left(\mathbb{E}[\hat{\mu}_{\mathrm{data}}(B_{\varepsilon_0}(\Pi_\mathcal{M}(Y/\mu_\epsilon)))^{-1}|S,Z]\right) + \left(1 + \tfrac{K_\mathcal{M}+D_Z}{\tau}\right)\frac{\varepsilon_0^2}{\sigma_\epsilon^2}\right\}\right).$$

To bound the term with the infimum, we proceed similarly to the proof of Theorem 3.6, bounding it using Lemma D.7. Given that $N$ is sufficiently large, we obtain from this lemma that,

$$\sup_{y \in \mathcal{M}} \hat{\mu}_{\mathrm{data}}(B_{\varepsilon_0}(x))^{-1} \leq c_\mu^{-1}\varepsilon_0^{-d_\mathcal{M}^*}/4,$$

with probability $1 - \delta$. With this we can choose $\varepsilon_0^2 = d^*\sigma_\epsilon^2$ to obtain the upper bound,

$$2\inf_{\varepsilon_0 > 0}\left\{\log\left(\mathbb{E}_{Y \sim k_Z}[\hat{\mu}_{\mathrm{data}}(B_{\varepsilon_0}(\Pi_\mathcal{M}(Y/\mu_\epsilon)))^{-1}|S,Z]\right) + \left(1 + \tfrac{K_\mathcal{M}+D_Z}{\tau_\mathcal{M}}\right)\frac{\varepsilon_0^2}{\sigma_\epsilon^2}\right\}$$

$$\leq C_3\left(1 + \left(1 + \tfrac{K_\mathcal{M}+D_Z}{\tau_\mathcal{M}}\right)d_\mathcal{M}^*\right),$$

for some quantity $C_3 > 0$ which depends only logarithmically on structural parameters. With this, we obtain the bound,

$$2\log\mathbb{E}[\exp(|\mathbb{E}[\Delta\,\mathrm{LSE}_\mathcal{M}(Z)|S,Z]|)|S]$$

$$\lesssim \frac{K_\mathcal{M}}{\tau_\mathcal{M}}\left(1 + \left(\Delta_\mathcal{M} + \tau_\mathcal{M} + K_\mathcal{M}\right)\frac{\Delta_\mathcal{M}}{\sigma_\epsilon^2} + d_\mathcal{M}^*\right) + \log\mathbb{E}\left[\exp\left(\frac{40K_\mathcal{M}}{\tau_\mathcal{M}}\left(\frac{C_3 d_\mathcal{M}^*}{\tau_\mathcal{M}} + \frac{12\Delta_\mathcal{M}^2}{\sigma_\epsilon^2}\right)D_Z\right)\Big|S\right]$$

$$+ \log\mathbb{E}\left[\exp\left(\frac{10K}{\tau}\frac{(5K^{1/2}K_{\max}^{1/2} + D_Z)^2}{\sigma_\epsilon^2}\mathbb{1}_{5K_{\max}+D_Z \geq \mu_\epsilon\tau/4}\right)\Big|S\right].$$

(54)

We bound the second and third terms similarly to as in the proof of Theorem 3.6. However, the concentration of $D_Z$ differs slightly due to the additional error from $\Delta_\mathcal{M} > 0$. We apply Lemma E.4 with $\varepsilon = K/2$ to obtain that for any $r^2 \geq 2\sigma_\epsilon^2 d$,

$$\mathbb{P}(D_Z \geq r|S) \leq \exp\left(C - \frac{(r - \sqrt{2\sigma_\epsilon^2 d})^2}{4\sigma_\epsilon^2}\right),$$

where the constant is given by

$$C = \log(8)d + \frac{5(K + \mu_\epsilon\Delta_\mathcal{M})^2 + K^2}{2\sigma_\epsilon^2} + 8c_\mu^{-1}(\tau/64)^{-d_\mathcal{M}^*}$$

$$\leq \log(8)d + \frac{6(K + \mu_\epsilon\Delta_\mathcal{M})^2}{2\sigma_\epsilon^2} + 8c_\mu^{-1}(\tau/64)^{-d_\mathcal{M}^*}.$$

To bound the third term of (54), we can directly use the argument in the proof of Theorem 3.6 that produces (51). Indeed, taking $\sigma_\epsilon^2$ sufficiently small, we obtain,

$$\log \mathbb{E}\left[\exp\left(\frac{10 K_\mathcal{M}}{\tau_\mathcal{M}} \frac{(5K^{1/2} K_{\max}^{1/2} + D_Z)^2}{\sigma_\epsilon^2} \mathbb{1}_{D_Z \leq \mu_\epsilon \tau_\mathcal{M}/4}\right)\Big| S\right] \lesssim \frac{K_\mathcal{M}}{\tau_\mathcal{M}}.$$

Similarly, we can borrow the argument that produces (49), to obtain

$$\log \mathbb{E}\left[\exp\left(\frac{40 K_\mathcal{M}}{\tau_\mathcal{M}}\left(\frac{C_3}{\tau_\mathcal{M}} + \frac{\Delta_\mathcal{M}^2}{\sigma_\epsilon^2}\right) d_\mathcal{M}^* D_Z\right)\Big| S\right]$$
$$\lesssim c\sqrt{\sigma_\epsilon^2 d} + c^2 \sigma_\epsilon^2 + c K_\mathcal{M} + c\sigma_\epsilon(\sqrt{d} + c_\mu^{-1/2}(\tau/64)^{-d^*/2})$$
$$\lesssim \frac{K_\mathcal{M}}{\tau_\mathcal{M}}\left(\frac{1}{\tau_\mathcal{M}} + \frac{\Delta_\mathcal{M}^2}{\sigma_\epsilon^2}\right) d_\mathcal{M}^*\left(K_\mathcal{M} + \Delta_\mathcal{M} + \sigma_\epsilon(\sqrt{d} + c_\mu^{-1/2}(\tau/64)^{-d^*/2})\right.$$
$$\left. + \sigma_\epsilon^2 \frac{K_\mathcal{M}}{\tau_\mathcal{M}}\left(\frac{1}{\tau_\mathcal{M}} + \frac{\Delta_\mathcal{M}^2}{\sigma_\epsilon^2}\right) d_\mathcal{M}^*\right).$$

With this, we obtain that as soon as $\sigma_\epsilon^2$ is sufficiently small, we obtain,

$$\log \mathbb{E}\left[\exp\left(\frac{40 K_\mathcal{M}}{\tau_\mathcal{M}}\left(\frac{C_3}{\tau_\mathcal{M}} + \frac{\Delta_\mathcal{M}^2}{\sigma_\epsilon^2}\right) d_\mathcal{M}^* D_Z\right)\Big| S\right] \lesssim \frac{K_\mathcal{M}^2}{\tau_\mathcal{M}}\left(\frac{1}{\tau_\mathcal{M}} + \frac{\Delta_\mathcal{M}^2}{\sigma_\epsilon^2}\right) d_\mathcal{M}^*.$$

Therefore, returning to (54) we obtain the bound,

$$2 \log \mathbb{E}[\exp(|\mathbb{E}[\Delta \operatorname{LSE}_\mathcal{M}(Z)|S, Z]|)|S]$$
$$\leq \frac{K_\mathcal{M}}{\tau_\mathcal{M}}\left(1 + \left(\Delta_\mathcal{M} + \tau_\mathcal{M} + K_\mathcal{M}\right)\frac{\Delta_\mathcal{M}}{\sigma_\epsilon^2} + d^*\right) + \frac{K_\mathcal{M}^2}{\tau_\mathcal{M}}\left(\frac{1}{\tau_\mathcal{M}} + \frac{\Delta_\mathcal{M}^2}{\sigma_\epsilon^2}\right) d_\mathcal{M}^*$$
$$\lesssim \frac{K_\mathcal{M}(d^* + 1)}{\tau_\mathcal{M}} + \frac{K_\mathcal{M}\Delta_\mathcal{M}}{\sigma_\epsilon^2}\left(1 + \frac{K_\mathcal{M} d_\mathcal{M}^*}{\tau_\mathcal{M}}\Delta_\mathcal{M}\right).$$

$\square$

# F  Proofs of other results

This appendix contains the proofs for Corollary 3.7 and Proposition 3.8. These results elucidate some of the generalisation properties of log-domain smoothing, specifically regarding how it concentrates mass near the manifold and distributes mass along it.

## F.1  Proof of Corollary 3.7

To prove this corollary, we utilise Lemma E.4. As remarked in the proof of Theorem 3.6, there exists $\varepsilon_0 \in [0, \tau/64]$ such that with a probability of at least $1 - \delta$, we obtain the bound,

$$\mathbb{P}_{Y \sim \hat{p}_\epsilon^{k\mathcal{M}}}\left(\operatorname{dist}(Y, \mathcal{M}) \geq r | S\right) \leq \exp\left(C - \frac{(r - \sqrt{2\sigma_\epsilon^2 d})^2}{4\sigma_\epsilon^2}\right), \quad C := \log(8)d + \frac{5K^2 + 4\mu_\epsilon^2 \varepsilon_0^2}{2\sigma_\epsilon^2} + 8c_\mu^{-1}\varepsilon_0^{-d^*},$$

for any $r^2 \geq 4\sigma_\epsilon^2 d$, where $\varepsilon_0$ must be chosen according to the size of $N$ and the condition in (18). In particular, we have the bound,

$$\mathbb{P}_{Y \sim \hat{p}_\epsilon^{k\mathcal{M}}}\left(\operatorname{dist}(Y, \mathcal{M}) \geq r + \sqrt{2\sigma_\epsilon^2 d} + \sqrt{4\sigma_\epsilon^2 C^2} | S\right) \leq \exp\left(-\frac{r^2}{4\sigma_\epsilon^2}\right).$$

When $N$ is large enough, we choose the optimal value of $\varepsilon_0^2 = \sigma_\epsilon^{\frac{4}{d^*+2}} c_\mu^{-\frac{2}{d^*+2}}$, so that,

$$\sigma_\epsilon^2\left(\frac{4\mu_\epsilon^2 \varepsilon_0^2}{2\sigma_\epsilon^2} + 8c_\mu^{-1}\varepsilon_0^{-d^*}\right) \lesssim \sigma_\epsilon^{\frac{4}{d^*+2}} c_\mu^{-\frac{2}{d^*+2}}.$$

If $N$ is not sufficiently large, we choose $\varepsilon_0$ to be the smallest value such that (18) is satisfied, yielding $(c_\mu^2 N)^{-1/d^*} \lesssim \varepsilon_0^2 \lesssim (c_\mu^2 N)^{-1/d^*}$ and also,

$$\sigma_\epsilon^2\left(\frac{4\mu_\epsilon^2 \varepsilon_0^2}{2\sigma_\epsilon^2} + 8c_\mu^{-1}\varepsilon_0^{-d^*}\right) \lesssim (c_\mu^2 N)^{-1/d^*}.$$

With these choices of $\varepsilon_0$, we obtain the bound,

$$\sqrt{4\sigma_\epsilon^2 C^2} \lesssim \sqrt{\sigma_\epsilon^2 d} + K + \max\{(c_\mu^2 N)^{-1/d^*}, \sigma_\epsilon^{\frac{2}{d^*+2}} c_\mu^{-\frac{1}{d^*+2}}\}.$$

Next we transfer this concentration property to the measure $\hat{p}_\epsilon^k$ by utilising the Rényi divergence bound in Theorem 3.6. By utilising Lemma 21 of Chewi et al., 2022, we obtain the bound,

$$\mathbb{P}_{Y \sim \hat{p}_\epsilon^k}\Big( \text{dist}(Y, \mathcal{M}) \geq r + \sqrt{2\sigma_\epsilon^2 d} + \sqrt{4\sigma_\epsilon^2 C^2} + \sqrt{4\sigma_\epsilon^2 D_2(\hat{p}_\epsilon^k \| \hat{p}_\epsilon^{k^\mathcal{M}})} \Big| S \Big) \leq 2 \exp\Big( -\frac{r^2}{8\sigma_\epsilon^2} \Big).$$

Finally we use Theorem 3.6 to bound the 2-Rényi divergence with probability at least $1 - \delta$.

## F.2 Proof of Proposition 3.8

Fix $x \in \mathcal{M}$ and let $\gamma_t^i$ be the shortest constant-velocity geodesic on $\mathcal{M}$ connecting the points $x$ and $x_i$. Then the density ratio at $x$ compared with $x_i$ can be expressed as

$$\frac{\hat{p}_\epsilon^k(x)}{\hat{p}_\epsilon^k(x_i)} = \exp\Big( \int \log \hat{p}_\epsilon(y)\, k_x(dy) - \int \log \hat{p}_\epsilon(y)\, k_{x_i}(dy) \Big)$$

$$= \exp\Big( \int \big( \log \hat{p}_\epsilon(y) - C_\epsilon + \|y - \Pi_{\mathcal{M}^\epsilon}(y)\|^2/2\sigma_\epsilon^2 \big)\, k_x(dy)$$

$$- \int \big( \log \hat{p}_\epsilon(y) - C_\epsilon + \|y - \Pi_{\mathcal{M}^\epsilon}(y)\|^2/2\sigma_\epsilon^2 \big)\, k_{x_i}(dy) \Big),$$

where we have used that $\mathbb{E}_{Y \sim k_x}[\text{dist}(Y, \mathcal{M})^2]$ is constant in $x$. Thus, by the fundamental theorem of calculus applied along the path $t \mapsto \gamma_t^i$, we obtain

$$\frac{\hat{p}_\epsilon^k(x)}{\hat{p}_\epsilon^k(x_i)} = \exp\Big( \int_0^1 \int \big( \log \hat{p}_\epsilon(y) - C_\epsilon + \|y - \Pi_{\mathcal{M}^\epsilon}(y)\|^2/2\sigma_\epsilon^2 \big) \langle \nabla_x k_x(y)\big|_{x=\gamma_t^i}, \partial_t \gamma_t^i \rangle\, dy\, dt \Big).$$

To control this term further, we introduce the Fisher information matrix

$$\mathcal{I}(x) = \int \big( \nabla_x \log k_x(y) \big) \big( \nabla_x \log k_x(y) \big)^\top k_x(dy),$$

and define

$$F^2 = \sup_{x \in \mathcal{M}} \sup_{\substack{v \in T_x \mathcal{M} \\ \|v\|=1}} v^\top \mathcal{I}(x)\, v.$$

From Cauchy–Schwarz, for each fixed $t$ we have

$$\int \big( \log \hat{p}_\epsilon(y) - C_\epsilon + \|y - \Pi_{\mathcal{M}^\epsilon}(y)\|^2/2\sigma_\epsilon^2 \big) \langle \nabla_x k_x(y)\big|_{x=\gamma_t^i}, \partial_t \gamma_t^i \rangle\, dy$$

$$= \int \big( \log \hat{p}_\epsilon(y) - C_\epsilon + \|y - \Pi_{\mathcal{M}^\epsilon}(y)\|^2/2\sigma_\epsilon^2 \big) \langle \nabla_x \log k_x(y)\big|_{x=\gamma_t^i}, \partial_t \gamma_t^i \rangle\, k_{\gamma_t^i}(dy)$$

$$\leq \Big( \int \big| \langle \nabla_x \log k_x(y)\big|_{x=\gamma_t^i}, \partial_t \gamma_t^i \rangle \big|^2 k_{\gamma_t^i}(dy) \Big)^{1/2}$$

$$\times \Big( \int \big( \log \hat{p}_\epsilon(y) - C_\epsilon + \|y - \Pi_{\mathcal{M}^\epsilon}(y)\|^2/2\sigma_\epsilon^2 \big)^2 k_{\gamma_t^i}(dy) \Big)^{1/2}$$

$$\leq \big( (\partial_t \gamma_t^i)^\top \mathcal{I}(\gamma_t^i)\, (\partial_t \gamma_t^i) \big)^{1/2} \Big( \int \big( \log \hat{p}_\epsilon(y) - C_\epsilon + \|y - \Pi_{\mathcal{M}^\epsilon}(y)\|^2/2\sigma_\epsilon^2 \big)^2 k_{\gamma_t^i}(dy) \Big)^{1/2}$$

$$\leq F \|\partial_t \gamma_t^i\| \Big( \int \big( \log \hat{p}_\epsilon(y) - C_\epsilon + \|y - \Pi_{\mathcal{M}^\epsilon}(y)\|^2/2\sigma_\epsilon^2 \big)^2 k_{\gamma_t^i}(dy) \Big)^{1/2}.$$

Next we bound the term

$$\Delta_\epsilon(y) := \log \hat{p}_\epsilon(y) - C_\epsilon + \|y - \Pi_{\mathcal{M}^\epsilon}(y)\|^2/2\sigma_\epsilon^2.$$

Recall that

$$\log \hat{p}_\epsilon(y) = \mathrm{LSE}\left(\left\{-\|y - \mu_\epsilon x_j\|^2/2\sigma_\epsilon^2\right\}_{j=1}^N\right),$$

so that

$$
\begin{aligned}
|\Delta_\epsilon(y)| &= \left|\mathrm{LSE}\left(\left\{-\|y - \mu_\epsilon x_j\|^2/2\sigma_\epsilon^2\right\}_{j=1}^N\right) + \|y - \Pi_{\mathcal{M}^\epsilon}(y)\|^2/2\sigma_\epsilon^2\right| \\
&\leq \left|\mathrm{LSE}\left(\left\{-\left(\|y - \Pi_{\mathcal{M}^\epsilon}(y)\|^2 + 2\langle y - \Pi_{\mathcal{M}^\epsilon}(y), \Pi_{\mathcal{M}^\epsilon}(y) - \mu_\epsilon x_j\rangle\right.\right.\right.\right. \\
&\qquad\qquad \left.\left.\left.\left. + \|\Pi_{\mathcal{M}^\epsilon}(y) - \mu_\epsilon x_j\|^2\right)/2\sigma_\epsilon^2\right\}_{j=1}^N\right) + \|y - \Pi_{\mathcal{M}^\epsilon}(y)\|^2/2\sigma_\epsilon^2\right| \\
&\leq -\mathrm{LSE}\left(\left\{-\left(\|\|y - \Pi_{\mathcal{M}^\epsilon}(y)\|\|\Pi_{\mathcal{M}^\epsilon}(y) - \mu_\epsilon x_j\|^2/\tau + \|\Pi_{\mathcal{M}^\epsilon}(y) - \mu_\epsilon x_j\|^2\right)/2\sigma_\epsilon^2\right\}_{j=1}^N\right) \\
&\leq -\mathrm{LSE}\left(\left\{-\|\Pi_{\mathcal{M}^\epsilon}(y) - \mu_\epsilon x_j\|^2/\sigma_\epsilon^2\right\}_{j=1}^N\right).
\end{aligned}
$$

Using the bound $\|y - \Pi_{\mathcal{M}^\epsilon}(y)\| \leq K_{\max} \leq \tau$ and the same argument as in the proof of Corollary 3.7, we obtain

$$|\Delta_\epsilon(y)| \leq \frac{2\mu_\epsilon^2 \varepsilon^2}{\sigma_\epsilon^2} + 2\,\hat{\mu}_{\mathrm{data}}\left(B_\varepsilon(\Pi(y/\mu_\epsilon))\right)^{-1}.$$

Since $N \geq N_{\min}$, we have that with

$$\hat{\mu}_{\mathrm{data}}\left(B_\varepsilon(\Pi(y/\mu_\epsilon))\right) \geq c_\mu \varepsilon^{d^*}.$$

In particular, this implies the uniform bound

$$|\Delta_\epsilon(y)| \leq 4\left(\frac{2\mu_\epsilon^2}{\sigma_\epsilon^2}\right)^{\frac{d^*+1}{d^*+2}} c_\mu^{-\frac{1}{d^*+2}} \leq \frac{8\mu_\epsilon^2}{\sigma_\epsilon^2 c_\mu^{\frac{1}{d^*+2}}}.$$

Combining this with the previous estimate yields the following bound on the density ratio:

$$\frac{\hat{p}_\epsilon^k(x)}{\hat{p}_\epsilon^k(x_i)} \geq \exp\left(-\frac{8\mu_\epsilon^2 F}{\sigma_\epsilon^2 c_\mu^{\frac{1}{d^*+2}}} \int_0^1 \|\partial_t \gamma_t^i\|\,dt\right).$$

Finally, we control the length of the path $\gamma_t^i$. Let $x_i^\star$ be a nearest neighbour of $x$ among the sample points, so that

$$\|x - x_i^\star\| = \inf_{i \in [N]} \|x - x_i\|.$$

By the probability bound, we have $\inf_{i \in [N]} \|x - x_i\| \leq \tau/2$ with high probability. Together with Lemma D.4, this yields

$$\int_0^1 \|\partial_t \gamma_t^i\|\,dt \leq 2\|x - x_i^\star\|,$$

which completes the proof of the proposition.

# G  Experimental details

In this section, we provide detailed descriptions of the experimental settings used in the paper. Code to reproduce the experiments is available at `https://github.com/samuel-howard/log_smoothing`.

## G.1  2-dimensional circle example

The plots in Figure 4 illustrate the trade-off that score-smoothing can provide for generalisation, to complement the theoretical results and discussion in Section 3.3. We consider an empirical dataset of 12 uniformly spaced points on the unit circle, and generate samples using the smoothed score function with an isotropic Gaussian kernel. We use a variance exploding diffusion model with $T = 9$ and a geometric noise schedule, an Euler-Maruyama discretisation with 100 steps, and 1000 samples in the smoothing evaluation. In Figure 4 we show how the resulting samples behave for different smoothing parameter $\sigma$. Too little smoothing generates only training data, while too much smoothing causes generated samples to move towards the centre of the circle. There is a good choice of smoothing that balances between the two, promoting generalisation along the manifold (a phenomenon noticed by Scarvelis et al. (2025)).

To further illustrate this trade-off, we also plot how the population negative log-likelihood changes as the degree of smoothing increases, averaged over 1000 points on the true circular manifold. We recall that one can calculate the log-likelihood of a point by integrating the divergence of the probability-flow ODE drift function along the probability flow ODE trajectory (Chen et al., 2018). In our case, the drift of the probability flow ODE is a smoothed empirical score function. As we are considering isotropic Gaussian smoothing, the divergence and the kernel convolution can be interchanged, allowing us to compute the log-likelihood by integrating the smoothed divergence of the empirical score function. The resulting plot exhibits a U-shape, clearly demonstrating the generalisation trade-off that arises from varying the smoothing level.

## G.2  Comparing Gaussian smoothing and KDE

We here describe the experimental setup used in Section 5.1, which aims to illustrate how score-function smoothing can preserve the geometry of the data, in contrast to density-level smoothing which quickly loses such structure. In order to consider a well-structured manifold, we follow Rombach et al. (2022) and use a 32-dimensional VAE latent-space encoding of MNIST digits, and perform generation in the latent space. We consider $\mathcal{M}$ to be the set corresponding to the digit 4, which comprises a lower-dimensional structure in the latent space. This ground-truth manifold is approximated using all samples of the digit 4, from which we use a subset of 100 samples as our training dataset. We consider a smoothed-score diffusion model using an isotropic Gaussian kernel, and compare with kernel density estimation which corresponds to smoothing at the density level.

**VAE**   We train the VAE on 10,000 samples from the MNIST database. The VAE uses 16 initial feature channels, with scaling multiples of $(1, 2, 2, 2)$ during downsampling, a convolutional kernel size of 3, a dropout rate of 0.1, and 4 groups for group normalisation. It maps into a 32-dimensional latent space. It is trained for 10,000 training steps with a batch size of 64, using the Adam optimiser (Kingma and Ba, 2015) with learning rate 1e-3 and default parameters 0.9, 0.999.

**Dataset construction**   We use the remaining 60,000 points not used to train the VAE. The latent representations of the individual classes comprise lower-dimensional structures in the space, which we verify in Figure 10. To show this, for a particular class we map all points to the latent space. We then pick one of the latent points $z$, and look at its 50 closest neighbours $z_i$. We perform a PCA decomposition on the vectors $z - z_i$ to analyse their local structure. In Figure 10 we plot the cumulative explained variance, and observe that most of the variance is captured by fewer than the full 32 dimensions, suggesting that the latent representations for a particular class lie on a lower-dimensional manifold within the latent space.

We restrict to considering the digit 4, of which there are 5842 samples. We use these points to approximate the 'true' manifold $\mathcal{M}$, and randomly choose 100 points for use as the empirical dataset.

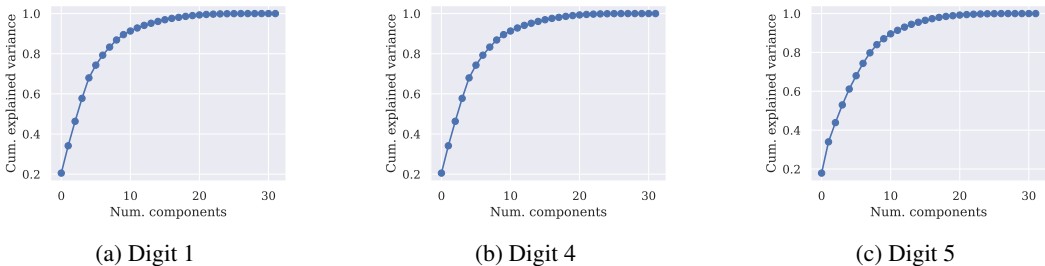

(a) Digit 1        (b) Digit 4        (c) Digit 5

Figure 10: Verifying that latent representations of a digit class lie on a lower-dimensional structure in the latent space, by performing a PCA decomposition on the differences between nearby points.

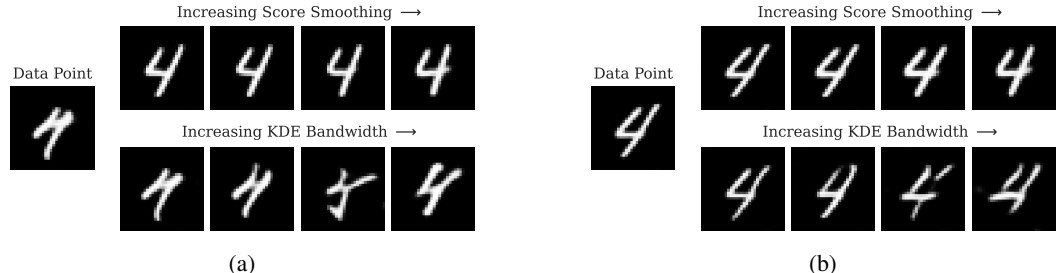

Figure 11: Additional generations, as in Figure 6.

**Experiment hyperparameters**  We use a variance-exploding diffusion model with T = 9.0, a geometric noise schedule, and 100 generation steps with an Euler-Maruyama discretisation scheme. We generate 500 samples to calculate the $L_2$ distances reported in Figure 7. For the isotropic Gaussian kernel, we used smoothing with standard deviations $\sigma \in \{0.4, 0.45, 0.5, 0.6, 0.7, 0.8, 0.9\}$, and we use 1000 smoothing samples at each generation step. For KDE, we use $\sigma \in \{0.07, 0.1, 0.13, 0.16, 0.19, 0.22, 0.25\}$, which were chosen to induce comparable average distances to the dataset from the generated samples (plotted along the $x$-axis).

To obtain the samples plotted in Figure 6, we use Gaussian-smoothing with standard deviations $\sigma \in \{0.3, 0.4, 0.5, 0.7\}$, and KDE scales $\sigma \in \{0.1, 0.2, 0.4, 0.5\}$. These were selected to induce comparable lateral distances along the manifold, computed as $(d(x, \hat{\mu}_{data})^2 - d(x, \mathcal{M})^2)^{\frac{1}{2}}$ and averaged over 500 generated samples, which corresponds to inducing the same degree of 'novelty' relative to the training set. We display the same plot showing more samples in Figure 11. The quality of the reconstruction provides a proxy for how well the manifold structure is preserved. For the score-smoothed diffusion model, as the amount of smoothing increases, the samples become novel images that are not present in the dataset yet nonetheless decode to resemble samples from the class of 4's, suggesting that they remain close to the underlying geometry. For KDE, we see that the quality of the reconstructed samples deteriorates more, indicating that in order to induce the same degree of novelty and difference from the training set, the KDE moves significantly further off-manifold, thereby failing to preserve the geometric structure of the data.

### G.3 Synthetic image manifold

We now describe the experimental setup used in the synthetic image manifold experiment in Section 5.2. As working in pixel space is more challenging, we will focus on a simple one-dimensional image manifold. Now that we are operating in the pixel space, the training datapoints are further apart and thus there are many permissible manifolds that interpolate the data. In Section 4, we considered how the *type* of smoothing can induce different structures in the generated samples, and illustrated this effect with low-dimensional experiments. We therefore aim to assess to what extent this intuition transfers to higher-dimensional settings by also considering smoothing kernels adapted to the manifold structure, and seeing whether this can influence the geometric structure of the sampling distribution.

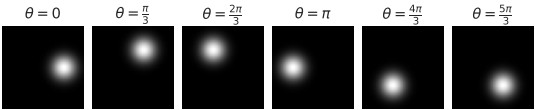

Figure 12: Visualisation of traversing the synthetic image manifold.

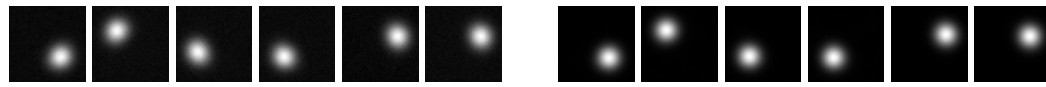

(a) Isotropic Gaussian smoothing, $\sigma = 2.6$        (b) Adapted smoothing, $\sigma = 5.0$

Figure 13: Samples generated using the Gaussian and manifold-adapted smoothing kernels. The manifold-adapted smoothing generates samples that are visually more 'on-manifold', in that the samples appear more spherically symmetric. See also Figure 8 for a quantitative measure of the spherical symmetry of the samples.

**Dataset generation**    We construct the synthetic image dataset using a function $\phi : [0, 2\pi) \to \mathbb{R}^{64 \times 64}$ that maps an angle $\theta$ to an 'image'. The 'image' is constructed as the density of a $\eta^2$-variance Gaussian distribution centred on the point on the circle with radius 0.5 corresponding to the angle $\theta$ (where the overall image corresponds to $[-1, 1] \times [-1, 1]$). The density is scaled to take values between 0 and 1. The resulting manifold in image space therefore consists of a closed curve of 'Gaussian bumps' that move around the 0.5-circle as $\theta$ moves from 0 to $2\pi$. We provide a visualisation of traversing the manifold in Figure 12. We use $\eta = 0.2$, and use 16 equally spaced points along the curve as the training dataset.

The manifold-adapted smoothing kernel is defined as follows. For a point $x$ in the generation procedure, the projection $\Pi_{\mathcal{M}}(x)$ is computed. We define a shifted manifold as $\mathcal{M} + (x - \Pi_{\mathcal{M}}(x))$, which is a translated copy of the manifold that passes through $x$. Gaussian noise of standard deviation $\sigma$ is added to $x$, then we project onto this shifted manifold. All manifold projections are approximated by generating 1024 equally spaced points along the manifold and taking the closest one.

**Experiment hyperparameters**    We use a variance-exploding diffusion model with $T = 9.0$, a geometric noise schedule, and 100 generation steps with an Euler-Maruyama discretisation scheme. For the isotropic Gaussian kernel, we used smoothing with standard deviations $\sigma \in \{1.0, 1.4, 1.8, 2.0, 2.2, 2.4, 2.6\}$. For the manifold-adapted smoothing, we used $\sigma \in \{1.6, 2.4, 3.2, 3.5, 3.8, 4.4, 5.0\}$. These values were chosen to induce comparable average distances to the training dataset in the generated samples (plotted along the $x$-axis).

For the isotropic Gaussian smoothing, we took 50,000 kernel samples at each generation step. For the manifold-adapted smoothing, we take 1000 smoothing samples at each generation step (note that this can be much lower than for Gaussian smoothing, as the manifold along which we smooth is only 1-dimensional). We generate 100 samples, and average the closest distances to the manifold and to the empirical dataset. As with the projections, the closest distance to the manifold is calculated by generating 1000 points on the manifold and taking the minimum $L_2$ distance.

**Assessing the visual quality of samples**    The plot on the left of Figure 8 reports the average $L_2$ distance from the manifold $\mathcal{M}$, relative to the average $L_2$ distance from the training dataset. It is clear that the Gaussian-smoothed samples deviate comparatively further from the manifold than the adaptive-smoothing samples according to this distance. On the right of Figure 8, we additionally report an alternative measure of how 'on-manifold' the generations are, related to the *visual* properties of the generations.

Note that samples from the true manifold consists of renormalised Gaussian density functions. Visually, being 'on-manifold' therefore corresponds to the generated images being spherically symmetric. In Figure 13 we display generated samples for the isotropic Gaussian and manifold-adapted smoothing mechanisms (for the largest smoothing values that were used in Figure 8), and see that the manifold-adapted smoothing generates samples appear visually more spherically symmetric.

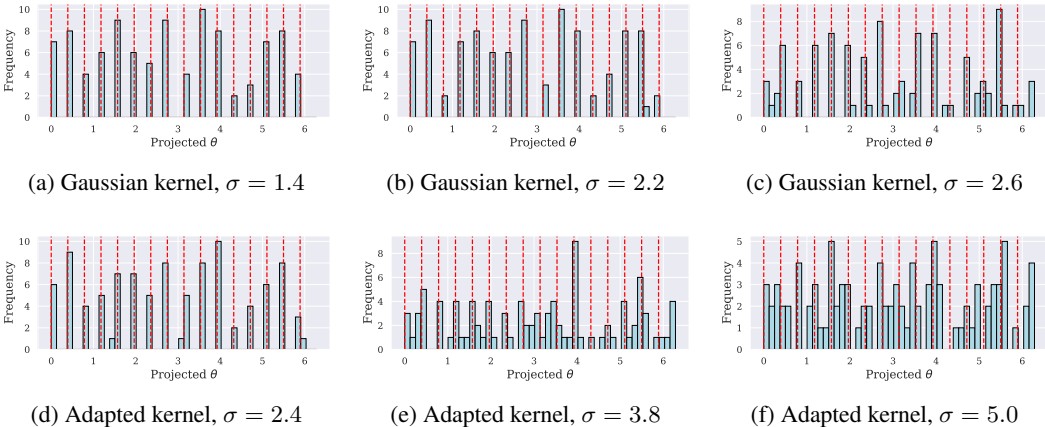

(a) Gaussian kernel, $\sigma = 1.4$    (b) Gaussian kernel, $\sigma = 2.2$    (c) Gaussian kernel, $\sigma = 2.6$

(d) Adapted kernel, $\sigma = 2.4$    (e) Adapted kernel, $\sigma = 3.8$    (f) Adapted kernel, $\sigma = 5.0$

Figure 14: Plots showing the projected $\theta$ values for the generated samples, for different amounts of smoothing. As the smoothing increases, the generated samples spread along the manifold structure, and populate the space between the points in the training dataset (indicated by red vertical lines).

This property is however somewhat difficult to assess by eye, as any such changes can be subtle, so in Figure 8 we also quantitatively measure the spherical symmetry to assess this visual property.

In order to do so, we report the 'anisotropy' of the generated samples. Namely, we consider the renormalised generated samples as a probability density function on $[0, 1] \times [0, 1]$, and record the anisotropy of the corresponding distribution (that is, we compute the covariance matrix $\Sigma \in \mathbb{R}^2$, and report $\frac{\lambda_{max}}{\lambda_{min}}$ for eigenvalues $\lambda_{max}, \lambda_{min}$). Samples that are 'on-manifold' will have values close to 1.0. In the computation, we set values less than 0.1 to zero, so that the noise in the generations does not impact the calculation.

The results are consistent with the pattern of $L_2$ distances reported in Figure 8—as the degree of smoothing increases and the generated samples deviate away from the training datapoints, the generations using the adapted smoothing have lower anisotropy and are therefore more 'round' than those obtained from Gaussian smoothing. Indeed, we know from Scarvelis et al. (2025) that Gaussian smoothing will generate barycentres of training points, which will skew the generated samples away from being perfectly round; it appears that the manifold-adapted smoothing somewhat mitigates this effect by shaping the geometry of the generated samples towards a different interpolation.

### G.3.1 Additional plots

The results in Figure 8 indicate that an adapted smoothing kernel can induce different structure in the generations compared to isotropic Gaussian smoothing—as the degree of smoothing increases, the generated samples deviate away from the training data for both kernels, but remain comparatively closer to the manifold structure when using the adapted smoothing kernel. We here include some additional plots that further elucidate this observed effect.

**Spread along the manifold**    While the $L_2$ distances reported in Figure 8 show that the generations have deviated away from the training data, it is not necessarily clear how the scale of such deviations corresponds to the degree of spreading along the manifold structure. We therefore also examine the extent to which the generated samples become spread along the manifold as the smoothing increases, to confirm that the generations do indeed deviate sufficiently far from the training points to reasonably be considered 'novel'.

In Figure 14, we plot histograms showing the projected $\theta$ values of the generations, in order to see how far the generated distribution has spread along the $1d$ synthetic manifold. We provide histograms for three different smoothing values, for both types of smoothing. For small smoothing levels, we recover only training points as expected, but as the smoothing increases we see that the generations do indeed deviate far from the training datapoints relative to the manifold structure, and spread out to fill the gaps in the manifold between the points in the training dataset.

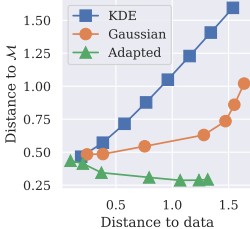

Figure 15: Comparing $L_2$ distance to data and $\mathcal{M}$, for a $2d$ synthetic image manifold.

**2-dimensional synthetic image manifold**   We now consider a similar 2-dimensional image manifold example, so see whether similar effects hold in this setting too. Now, rather than considering Gaussian bump images with the centres located around a circle, we consider the manifold induced by placing the centres of the Gaussian bumps covering the $[-0.5, 0.5] \times [-0.5, 0.5]$ square. As before we take a small training dataset, which now consists of 116 points positioned on a lattice of equilateral triangles covering the square, each with side-length 0.1. We run sampling as in the $1d$ case, now using smoothing with standard deviations $\sigma \in \{0.8, 1.0, 1.2, 1.4, 1.6, 1.8, 2.0\}$ for the Gaussian kernel, and $\sigma \in \{0.4, 0.8, 1.0, 1.2, 1.4, 1.6, 3.0\}$ for the manifold-adapted kernel. In Figure 15 we plot distance to the manifold (approximated with 2879 samples, on a triangular grid with triangle side-length 0.02) versus distance to the training dataset, and observe a similar effect to in the 1-dimensional case.

### G.4   MNIST manifold

We now provide the details for the MNIST manifold experiment in Section 5.2.

**Dataset generation**   Similarly to the synthetic case, we construct a manifold by defining a curve $\phi : [0, 1] \to \mathbb{R}^{32 \times 32}$ in pixel space, which interpolates between samples of the same digit from the MNIST dataset (LeCun et al., 2010). To obtain such an interpolation, we train a convolutional VAE (Kingma and Welling, 2014). We then choose three datapoints from the same digit class (in this case, the digit 4), and draw a triangle between their latent representations. We construct $\phi(t)$ by decoding this triangle, which results in a closed loop in pixel space. We use the decodings of 10 equidistant points along the latent triangular interpolation to define the training dataset. We emphasise that the VAE is only used to construct a manifold structure in pixel-space, and the actual diffusion procedure takes place directly in the pixel-space without any interaction with the VAE.

**Experiment hyperparameters**   We use a variance-exploding diffusion model with $T = 9.0$, a geometric noise schedule, and 100 generation steps with an Euler-Maruyama discretisation scheme. We used smoothing with standard deviations $\sigma \in \{0.0, 0.3, 0.6, 0.8, 0.9, 1.0, 1.05, 1.1\}$ for isotropic Gaussian smoothing, and $\sigma \in \{0.0, 0.5, 1.0, 1.5, 2.0, 2.5, 4.0, 7.0\}$ for manifold-adapted smoothing (which again were chosen to induce similar distances from the data points in Figure 9). For the isotropic Gaussian smoothing, we took 50,000 kernel samples at each generation step. For the manifold-adapted smoothing, we take 1000 smoothing samples at each generation step (this can be much lower than for Gaussian smoothing, as the manifold along which we smooth is only 1-dimensional). As before, we generate 100 samples, and report the average closest distances to the manifold and to the empirical dataset. The closest distance to the manifold is calculated by generating 1000 points on the manifold, and taking the minimum $L_2$ distance to these points.

**FID calculation**   As we work with a 1-dimensional cuve in pixel space, neighbouring points in the empirical dataset look very similar. It is therefore difficult to visually judge the quality of obtained samples from both smoothing mechanisms, so we use FID (Heusel et al., 2017) as measure of similarity to the true manifold that also provides an indication of sample quality. We compute the FID values using the Inception-v3 model, and we stack across the channels and resize to match the network input. We calculate the FID scores of the generated samples relative to the 1000 random samples from the manifold. As we use small dataset sizes for the computations, these values should not be compared to values reported elsewhere; nevertheless, they are still representative of the visible changes in sample quality.

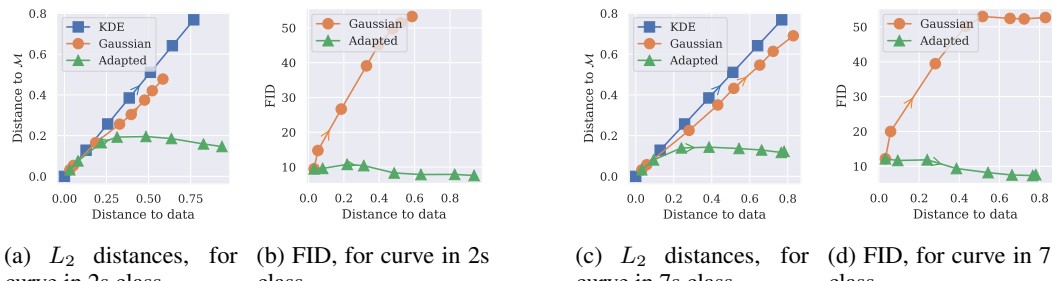

(a) $L_2$ distances, for curve in 2s class.

(b) FID, for curve in 2s class.

(c) $L_2$ distances, for curve in 7s class.

(d) FID, for curve in 7s class.

Figure 16: Comparison of Gaussian and manifold-adapted smoothing kernels, for alternative curves $\phi$ in the manifold of digits 2 and 7. Arrows indicate increasing smoothing.

**Different manifolds**   We also ran the same experiment with manifolds for different digits, and observe similar behaviour. Results for the curves for digits 2 and 7 are plotted in Figure 16. The selected points were generally chosen to be the first three examples of that digit in the dataset (other than when these datapoints induced a poorly-decoded manifold, in which case we used the first that made the constructed manifold of good quality).

**Licenses:**   MNIST digits classification dataset (LeCun et al., 2010), CC BY-SA 3.0 License

