# OpenReview forum: "Diffusion Models and the Manifold Hypothesis: Log-Domain Smoothing is Geometry Adaptive"
_NeurIPS.cc/2025/Conference — NeurIPS 2025 poster_

### Official Review · Reviewer_bgfQ · 2025-06-15

**Clarity:** 3
**Significance:** 3
**Originality:** 3
**Rating:** 4
**Confidence:** 3

**Summary:**

This paper seeks to understand the success of diffusion models by relating them to the manifold hypothesis. After recalling the basics of diffusion models, they argue that such models implicitly perform smoothing of the empirical density.

They provide theoretical results showing that the smoothed density is well approximated by its manifold adapted counterpart and that this counterpart keeps mass near the data manifold. They complement their theoretical analysis with experiments on synthetic data (where they can directly control the curvature) and on simple images from MNIST

**Questions:**

In light of the supposed connection between diffusion and the manifold hypothesis, I was wondering if diffusion models be applied to biomedical data sets which satisfy the manifold hypothesis? If so, this could be a good way of generating synthetic data for testing ML algorithms while preserving privacy

Also, is there a way to related diffusion models to manifold learning algorithms such as Diffusion Maps of Laplacian Eigenmaps? E.g., can one diffusion in the coordinates induced by these algorithms?

In settings where the manifold hypothesis is justified, is it possible to replace B_t in the definition of Eqn's 1 and 2 with a manifold-valued Brownian motion.

**Ethical Concerns:**

["NO or VERY MINOR ethics concerns only"]

**Final Justification:**

I stand by my earlier review, for the reasons outlined there. This is a good paper.

**Limitations:**

Yes

**Quality:**

3

**Strengths And Weaknesses:**

Strengths

The linear warm up is very helpful for gaining intuition

While I did not have time to check in detail, the proofs appear to be carefully written and well organized (unlike many proofs I see in conference submissions with obvious mistakes that jump of the page)

Connection between manifold hypothesis and diffusion is interesting as is the casting of implicit smoothing of the density in the log domain to preserve the manifold structure

Theoretical understanding of diffusion is limited



Weaknesses

The experimental set up is limited to images, in particular synthetic images and MNIST. It would be improved by considering experiments on non-image related manifolds or even on more complex image data sets

Furthermore, the connection between the success of diffusion models and the manifold hypothesis would be strengthened if one were to establish that diffusion models perform poorly on data that does NOT satisfy the manifold hypothesis

It is not entirely clear how the projection onto \Pi_{\mathcal{M}_r(x)} would be implemented if the manifold is unknown

Minor: Should explicitly state the B_t is a Brownian motion (Weiner process) in equations 1 and 2

---

> ### Author Rebuttal · Authors · 2025-07-31
>
> Thank you for the positive review. We appreciate that you found our theoretical contributions to be correct and that you agree we are tackling an important problem. Your main concerns appear to be centered on the experimental scope and the practical implications of our approach to the manifold hypothesis. We hope the following clarifications are helpful.
>
> ## Manifold hypothesis
> The classical manifold hypothesis assumes that there is a single, ground-truth manifold that one aims to recover (for example, a "true" manifold of images or proteins). This is how most of the theoretical analysis of diffusion models up to this point treat the problem, and also the way in which we analyse the setting up to Section 4. Afterwards, we offer an alternative perspective: we argue that in high dimensions there are many plausible interpolating manifolds given sparse observations. The choice of which manifold is learned is therefore determined by the inductive biases of the training procedure.
>
> Practitioners already leverage this kind of intuition, though often in an ad-hoc manner, by experimenting with different model architectures that encode certain invariances. Our work takes a step toward understanding how to design such a bias purposefully if the desired manifold structure were known. We demonstrate this on toy examples (Figures 1 and 3), where different biases lead to different interpolating manifolds for the same set of data points. For a more complex and realistic case like MNIST, we use a VAE to construct a manifold, and guide the smoothing to occur along the manifold structure to examine its induced effects.
>
>
> ## Weaknesses
> - We think that designing the correct inductive biases for specific, complex datasets is a challenging task. We hope our work has helped make a step in this direction. Our primary goal with this work is to deepen the theoretical understanding of what an ideal bias would entail if one already knew the target manifold structure. We believe this is a necessary step before such principles can be used to design application-specific biases. For instance, in scientific machine learning, where data may be solutions to a given PDE, we are hopeful that this framework can be used to design more effective biases/models in follow-up work.
> - To clarify, we are not claiming that diffusion models work only on manifold-structured data. Instead, we argue that they remain effective even in this challenging setting, and we posit that this effectiveness is linked to the implicit log-domain smoothing.
> - You are correct that the projection operator is not implementable without prior knowledge of the target manifold. In our experiments, we use the knowledge of the true manifold (constructed explicitly in Section 5.1, and via a VAE in Section 5.2) to construct the smoothing mechanism, while the function that is being smoothed is the empirical score and thus unaware of the true manifold structure. These experiments are designed to illustrate that the *type* of smoothing can provide a mechanism that impacts the resulting interpolating manifold, as suggested by the results in Section 4. We do not propose this as a practical algorithm or argue that this is necessarily the right way to actually smooth your neural network. We discuss this in lines 316-320, noting that if manifold-adapted smoothing occurs in practice, it will have arose implicitly from the models inductive biases, via architectural choices such as convolutions and attention [1].
>
> [1] Kamb and Ganguli, 2024, "An analytic theory of creativity in convolutional diffusion models"
>
> ## Questions
> - Biomedical datasets can vary greatly, but we do believe that many of them do satisfy some kind of manifold hypothesis, and that in these cases the manifold constraints might even be a bit more explicit than for images, making it possible to think about the right biases. We think an analysis in terms of privacy metrics is an important direction. However, it requiress a considerable amount of theoretical work which we plan to do in future publications. We are very interested in the reviewer's ideas for follow-up works on privacy in case the reviewer wants to elaborate.
> - If one could learn a manifold structure via a global coordinate chart $\mathbb{R}^d \to \mathbb{R}^D$, with $\mathbb{R}^D$ the ambient space, then it would be possible to run a diffusion model in these coordinates (see [1]). However, standard formulations of Diffusion Maps (and to the best of our knowledge Laplacian Eigenmaps as well) do not yield a global feature map $\mathbb{R}^d \to \mathbb{R}^D$. While the inverse map $\mathbb{R}^D \to \mathbb{R}^d$ (from the embedding to latent space) can often be extended, extending the forward direction chart $\mathbb{R}^d \to \mathbb{R}^D$ is typically non-canonical and non-trivial. Still, the reviewer may be interested in [2], which uses diffusion maps in a generative setting with an algorithm similar to diffusion models.
> - Yes indeed, in cases where the manifold is known, one can also run the whole algorithm directly on that manifold which has been explored in prior work, see for example [1]. However, our theoretical model is focused on analyzing why standard diffusion models work so well even when this structure is not explicitly known or enforced. Our work aims to make the connection between an inductive bias (e.g., tangential smoothing) and the recovered manifold more explicit.
>
> [1a] De Bortoli; Mathieu; Hutchinson; Thornton; Teh; Doucet. 2022. “Riemannian Score‑Based Generative Modelling.”
>
> [1b] Huang; Aghajohari; Bose; Panangaden; Courville. 2022. “Riemannian Diffusion Models.” Universität de Montréal & University of Toronto & McGill University
>
> [2] Gottwald; Li; Marzouk; Reich. 2024. “Stable generative modeling using diffusion maps.”
>
> ## Summary
> We thank the reviewer for their questions, as clarifying these points has greatly helped improve the quality of the paper. We have modified the camera-ready version in several places accordingly. We would be more than happy to discuss any questions further or to hear if you feel that any of the points we mentioned would in your opinion help improve our camera-ready version.
>
> In case you think that the above modifications have improved our rating beyond a borderline accept, we would be most appreciative if you would consider raising your score.

---

> > ### Comment · Reviewer_bgfQ · 2025-08-01
> >
> > I confirm the response. I am satisfied with the response.

---

### Official Review · Reviewer_sUDw · 2025-07-02

**Clarity:** 3
**Significance:** 3
**Originality:** 3
**Rating:** 5
**Confidence:** 2

**Summary:**

The paper concerns diffusion models and the relation between inductive biases and the fact that the score is related to the log density of the data. Specifically, the authors consider a specific kind of inductive bias in the form of a smoothing kernel. Assuming spatial stationarity, the convolution with the kernel commutes with the gradient in the score, and the smoothing can be seen to happen on the log density, in contrast to more standard kernel density estimation that relates to the density itself. The authors use this to investigate how the smoothing happens if the data is supported on manifolds of different complexities, and how small sample regimes affect the results. The paper is mostly theoretical, but it contains some experimental validation of the results.

**Questions:**

- "Consequently, if a smoothing kernel extends into regions off the manifold, the resulting smoothed log-density in those off-manifold regions will also equal −∞" ... "it will have exceedingly small values in regions distant from the data manifold" Could this intuitive argument also go the other way around: at a data manifold with positive codimension, the smoothing will extend to -\infty areas close by, and thus give very small values also at the data manifold itself. Essentially setting the probability at the data manifold to zero? (with the same argument) I think what saves it is that t>0 immediately smoothes the manifold so that the probability is positive in all dimensions and this case doesn't appear, but still one should be careful with the intuition in such cases
- reach in assumption 3.4: should the distance be signed? otherwise, for opposing points of a hyperplane, the dist would be equal but the points not. Isn't the reach usually defined as for \tau close points, the projection to M is unique?
- Theorem 3.6: "with high probability". What is the precise statement here?

**Ethical Concerns:**

["NO or VERY MINOR ethics concerns only"]

**Final Justification:**

The authors responded adequately to the questions and concerns I raised in my review. My initial rating was positive, and the response backs this evaluation. I therefore keep my score of the paper.

**Limitations:**

yes

**Paper Formatting Concerns:**

no concerns

**Quality:**

3

**Strengths And Weaknesses:**

Strengths:
- the theoretical exploration of the relation between inductive biases in this specific form is relevant, theoretically interesting, and is useful in understanding the behaviour of diffusion models
- the paper is well-written and clearly presented
- the theoretical results are non-trivial, particularly the extension of the result from linear spaces to manifolds
- the paper contains some experimental backing of the theoretical investigation

Weaknesses:
- the results rely on some assumptions, and it is not always clear how realistic they are in practical settings. E.g. eq (5) is an assumption, and it is exactly clear to me how realistic this actually is
- more generally, one can always be in doubt how relevant theoretical investigations like this are to actual data and training settings where the assumptions may not be satisfied. In any case, I find the investigation interesting in themselves as guides to understanding the relation between log domain smoothing and data supported on low-dimensional manifolds

---

> ### Author Rebuttal · Authors · 2025-07-31
>
> Thank you for the positive feedback on our paper. We are very happy that you appreciate our theoretical contributions. We will address your points and questions below.
>
> ## Weaknesses
> We agree that Equation (5) is an assumption. However, we want to note that it is essentially the assumption that the learned score is some smoothing of the empirical score, rather than a restriction on the type of smoothing that can occur. Since we allow the kernels to be position-dependent (i.e., the kernels $k_x$ can be completely unrelated for different $x$), this assumption posits that neural network takes the target function and at each point $x$, smooth the function arbitrarily and completely independent of other points. This allows for a very large class of smoothing operators (or even non-smoothing operators if $k_x$ is a dirac) to be encompassed in this setting. We then study which kind of assumptions we have to make for $k_x$ to recover a specific target manifold (in particular, Assumption 3.5). It turns out that smoothing in the log-domain is already a very powerful tool to recover manifolds; moreover, by smoothing along tangential directions of a target manifold $\mathcal{M}$, one can induce a biases towards that particular interpolating manifold.
>
> Regarding the practicality of our work, we see its primary contribution as deepening the theoretical understanding of how diffusion models interact with manifold-structured data. Nevertheless, we believe there are two key practical takeaways. First, smoothing empirical densities in the log-domain is more compatible with manifold structures than traditional density-domain smoothing (as illustrated in Figure 2), a principle that could inform the design of novel algorithms. Second, we argue one can induce a geometric bias by smoothing along the tangential directions of a desired manifold. While engineering neural networks with such specific inductive biases is a complex challenge, we hope our work provides a guiding intuition for this direction, which we plan to explore in future work.
>
>
> ## Questions
> - You raise an excellent point regarding the intuition of log-domain smoothing. In the theoretical limit where the log-density is $-\infty$ off-manifold, any smoothing kernel with support extending beyond the manifold would indeed yield a value of $-\infty$. However, for any positive time $t>0$, the noising process ensures the density $p_t(x)$ is positive everywhere. Consequently, the log-density is finite across the space. While smoothing with points far from the manifold will lower the log-density value on the manifold, the same effect occurs for all points. Since the final density will be normalized to mass 1, what matters isn't the absolute/specific values taken by the unnormalized smoothed log-density, but rather their relative differences. Our intuitive argument is that this procedure "punishes" points further away from the manifold more than points on the manifold, thus preserving the structure. We hope this clarifies the intuition. We have made sure to clarify this point in the paper and to emphasize that it is the relative effects that matter. We are open to making more changes if you think it would improve the presentation.
> - You are correct about the definition of reach. We made a small error when finalizing the paper; the definition in the submitted version is incorrect. The proper definition is indeed equivalent to requiring a unique projection onto the manifold. This definition is more intuitive and we have changed it.
> - When we say "with high probability", we mean that for any $\delta>0$, there exists an $N_\delta$ such that for $N>N_\delta$ the statement holds with probability at least $1-\delta$. We have revised the phrasing of the theorem to clarify this and to provide precise dependencies of the relevant quantities on each variable; please let us know if you would like more information regarding this point.
>
> ## Summary
> We want to thank the reviewer again for the great questions and we hope that our answers helped to address your concerns. We would be grateful for any final suggestions on which of these clarifications you feel are most helpful to incorporate into the final manuscript. If you think that our changes justify a change in your score, we would of course be delighted.

---

> > ### Comment · Reviewer_sUDw · 2025-08-01
> >
> > Thank you for the responses, and for addressing the questions in the review.

---

### Official Review · Reviewer_RumM · 2025-07-02

**Clarity:** 2
**Significance:** 3
**Originality:** 3
**Rating:** 5
**Confidence:** 3

**Summary:**

The paper offers an intuitive explanation for why diffusion models naturally apply smoothing in the log-density domain as a consequence of the score-matching objective.

**Questions:**

Regarding the identified weaknesses, I kindly request clarification on the following points to improve the clarity and presentation of the paper:

1. Illustration of Reach: Please consider providing a more intuitive and accessible illustration for the definition of reach. This request is purely aimed at enhancing the presentation and reader understanding, rather than questioning the validity of the concept.

2. Motivation for Assumptions 3.3, 3.4, and 3.5: It would be beneficial if the authors could elaborate on the intuition and motivation underlying 3.4 (lower bound on reach), and 3.5 (control over smoothing in normal directions). A clearer justification of their necessity would aid in understanding how these assumptions influence the theoretical results.

3. Role and Origin of Manifold Assumption in Theorem 4.1: Please clarify the role played by the assumption on the manifold family $\mathcal{M}$ in Theorem 4.1. In particular, how does this assumption arise, and what purpose does it serve in the context of the stated KL divergence bound?

**Ethical Concerns:**

["NO or VERY MINOR ethics concerns only"]

**Final Justification:**

The authors have addressed the concerns I raised in their response.

**Limitations:**

yes

**Quality:**

3

**Strengths And Weaknesses:**

**Strengths.**

- The paper offers a clear and accessible intuition for the connection between smoothing in the log-density domain and the score-matching objective in diffusion models. Despite not working directly in this area, I was able to develop a reasonable understanding of the core ideas.

- The theoretical results and definitions are presented with rigor, particularly in the supplementary material. The related work and limitations are thoroughly discussed.

**Weaknesses.**

While the paper is generally well-written and its sections are individually comprehensible, its theoretical nature calls for a more structured presentation of core concepts. For instance, the definition of manifold reach should be included in the main text; if that is not feasible, the authors should at least provide a clearer and more intuitive illustration. Additionally, the motivations behind key assumptions—such as Assumptions 3.4 and 3.5—should be elaborated upon in greater detail.

---

> ### Author Rebuttal · Authors · 2025-07-31
>
> Thank you for your positive assessment and for providing such constructive feedback!
>
> We appreciate the opportunity to clarify the presentation of our work, especially related to Assumptions 3.4 and 3.5. We have incorporated your feedback (and utilized the additional page) to improve the camera-ready version. We have expanded Section 4 with more intuition and enhanced the presentation in several places. As we cannot upload a revised manuscript at this stage, we will outline the changes below, focusing on the points you highlighted.
>
> ## Assumption 3.4: Manifold Reach
> We agree that the definition of reach would benefit from greater clarity. A minor error in the current definition may have contributed to some confusion. In the revised version, we have replaced this with a more intuitive, equivalent definition: *the reach is the maximal distance from the manifold within which every point has a unique projection onto that manifold*. This is the precise assumption used in our proofs and we believe this more intuitive definition may help clear some confusion.
>
> To offer further intuition on the concept and its necessity: for a linear submanifold, any point in the ambient space has a unique closest point on the submanifold. This is not guaranteed for curved manifolds. Consider a circle, for which the center point is equidistant from every point on its circumference. The reach of this circular manifold is precisely its radius. Our assumption is designed to exclude ill-behaved manifolds with rapidly oscillating curvature (e.g., the graph of y=sin(1/x) near x=0), where the reach can become arbitrarily small locally, posing challenges for analysis. We are excluding this case with this assumption.
>
>
> In the camera-ready version, we have moved the revised, more intuitive definition of reach into the main paper and included some of this intuition.
>
> ## Assumption 3.5: Smoothing tangential to the manifold
> This assumption bounds the extent to which the smoothing kernel operates in directions normal to the manifold. The linear case is again instructive. Imagine the manifold is a linear hyperplane. If a smoothing kernel $k_x$ only distributes mass parallel to this hyperplane, then for any point $Y \sim k_x$, we have $\text{dist}(Y,\mathcal{M})=\text{dist}(x,\mathcal{M})$. In this scenario, the assumption is trivially satisfied. However, if the kernel also smooths along directions normal to the manifold, the value of $\text{dist}(Y,\mathcal{M})$ will vary. The assumption specifies that this variation is bounded.
>
> In the nonlinear case, the ambient space can be locally decomposed into a set of "parallel" manifold slices, at least within the reach (per Assumption 3.4). The same intuition applies: we require a global bound on how much mass the kernel assigns to these parallel slices, which is an important condition for our proof technique.
>
>
> ## Theorem 4.1
> We agree with the reviewer that the definition of the manifold family would have benefited from further discussion. A smooth compact manifold $\mathcal{M}$ is a member of this family $\mathbb{M}_\mu$ if the projection of the data distribution onto $\mathcal{M}$ has a density that is uniformly lower-bounded away from zero:
>
> $$
>     \inf_{x \in \mathcal{M}} p_{\mu, \mathcal{M}}(x) > 0.
> $$
> The main intuition having no areas with $0$ probability is that these areas would never carry any samples, and could not be identified even if we have an infinite amount of samples. The requirement that the inf is strictly bounded away from $0$  is necessary to guarantee that even with a finite number of samples we can recover the whole manifold with a lower bounded probability. This type of assumption is typical in the literature on diffusion models and manifold learning (e.g., [1], [2]).
>
> [1] Aamari. 2017. “Convergence Rates for Geometric Inference.” Université Paris-Saclay.
>
> [2] Azangulov; Deligiannidis; Rousseau. 2024. “Convergence of Diffusion Models under the Manifold Hypothesis in High-Dimensions.”
>
>
> # Summary
> We thank the reviewer again for their concrete feedback, which has helped us identify areas where our presentation could be improved. We have already integrated these changes into our camera-ready version. If any of the above explanations was particularly helpful, we would be very happy if you point that out so that we can elaborate on it in the camera ready version. If you feel like these explanations address your concerns and raise the papers appropriate rating beyond a borderline accept, we would be delighted if you could change your score accordingly.

---

> > ### Comment · Reviewer_RumM · 2025-08-04
> >
> > I thank the authors for their response and am generally satisfied with the clarifications provided.
> >
> > I acknowledge that the definition of *reach* is fundamental and directly tied to the problem of manifold estimation. While I understand that such assumptions are common and often necessary in provable machine learning frameworks, I remain somewhat uncertain about their practical validity. In particular, since manifolds often carry semantic meaning while being embedded in a finite-dimensional space, their curvature can be highly complex - rendering the notion of reach somewhat ambiguous in real-world scenarios. To empirically validate the framework, additional experiments on more complex and diverse datasets - beyond synthetic data and MNIST - *must* be conducted.
> >
> > These are personal observations intended to help the authors further improve the manuscript, rather than criticisms of any fundamental flaw in the work. I therefore raise my overall score to acceptance.

---

### Official Review · Reviewer_bbTq · 2025-07-03

**Clarity:** 4
**Significance:** 4
**Originality:** 4
**Rating:** 6
**Confidence:** 3

**Summary:**

Diffusion models generate data from a target distribution by propagating points along paths given by a vector field.  That vector field is given by the gradient of log(p_t) where p_t is the distribution at time t, as it moves from the source (Gaussian) distribution to the target distribution of the data. A neural network is trained to approximate the time-dependent field \grad \log p_t(x) (known as the score).

The authors argue that since the NN is trained to approximates the log-density, any noise in the training process implicitly smoothes out the log values of the density. Since areas with a low density have log(density) go to minus infinity, this has the interesting effect of smoothing out the data only in the directions that are tangent to the data manifold, as long as the effective smoothing width is small w.r.t. the curvature of the manifold.  The authors explain their reasoning very well and prove this claim for a manifold with bounded reach. They also included an empirical study with both synthetic data and MNIST digits that supports the claim.

**Questions:**

1. The phenomenon discussed in this paper follows from the fact that as the density goes to zero the log-density goes to minus infinity. It seems that one could pick any other monotone function whose limit is minus infinity at zero and get the same behavior. This suggests that diffusion models based on variants of score matching should still be adaptive to the manifold geometry. Could your results be easily extended to such generalized settings? Will you consider running an experiment with a different monotone function?

2. By Conv(M) do you mean the convex hull of the manifold? If so it should be stated.

**Ethical Concerns:**

["NO or VERY MINOR ethics concerns only"]

**Final Justification:**

I have read all the reviews and rebuttals. No major problems were uncovered so I leave my scores unchanged. I am looking forward to see the final version with the additional 2D example and congratulate the authors again for this high quality work.

**Limitations:**

yes

**Quality:**

4

**Strengths And Weaknesses:**

The main observation in this paper is important and novel as far as I can tell (but I am not an expert on diffusion models). The paper is written very well and I enjoyed reading it. Overall, I think this is an excellent work.

Major comments:
1. It seems like the work could be extended beyond the log transformation. See the Questions section of the review.
2. The synthetic and MNIST examples are very nice but both are one-dimensional manifolds. Consider adding an example with 2 or 3 dimensions.

A few minor comments:
* It is well-known that the reach of a manifold controls the difficulty of estimating it is well-known, see for example Section 1.1 of [1].
* The concepts of smoothing, density estimation and local regression along the tangent directions / geodesics of a data manifold have been explored extensively in the literature years before generative diffusion models gained popularity. e.g. [2], [3], [4].

Typos:
Page 3, line 71: "spacial" -> "spatial"
Page 4, line 141: "subspace subspace"
Page 5, line 183: "the the"
Page 8, line 277: "16 uniformly points"


---

[1] Genovese, Christopher R., Marco Perone-Pacifico, Isabella Verdinelli, and Larry Wasserman. "Minimax manifold estimation." The Journal of Machine Learning Research 13, no. 1 (2012): 1263-1291.


[2] Cheng, Ming-Yen, and Hau-tieng Wu. "Local linear regression on manifolds and its geometric interpretation." Journal of the American Statistical Association 108, no. 504 (2013): 1421-1434.

[3] Moscovich, Amit, Ariel Jaffe, and Nadler Boaz. "Minimax-optimal semi-supervised regression on unknown manifolds." In Artificial Intelligence and Statistics, pp. 933-942. PMLR, 2017.

[4] Gao, Jia-Xing, Da-Quan Jiang, and Min-Ping Qian. "Adaptive manifold density estimation." Journal of Statistical Computation and Simulation 92, no. 11 (2022): 2317-2331.

---

> ### Author Rebuttal · Authors · 2025-07-31
>
> We would like to express our sincere gratitude to the reviewer for this rating and for the insightful, encouraging comments. We are very happy that you found our paper interesting! The references you provided are indeed highly relevant, and we have integrated them into the revised manuscript to better contextualize our work for the reader.
>
> ## Other transforms than the logarithm
> This is an excellent observation, and we agree that the core intuition extends beyond the logarithmic transform. We find this prospect very interesting. Our focus on the logarithm was motivated directly by the formulation of contemporary diffusion models: a neural network is trained to approximate $\nabla \log \hat{p}_t$. Since the network's approximation (and thus its implicit smoothing effect) occurs at the logarithmic level, we felt this was the natural choice for our analysis.
>
> Our current paper was primarily focused on trying to help explain part of the success of modern diffusion models, and as such we haven't found a good experiment for more general functions $f$ in explaining this. However, we would greatly appreciate any suggestions or ideas that you may have in this direction.
>
> We do, however, believe your point is especially prescient when considering the design of new algorithms. The principle that any monotonic function where $f(0) = -\infty$ could theoretically confer a similar geometry-adaptive benefit is a powerful one. Training a diffusion model with a different transform would mean that one would need to learn $\nabla f(p(x))$ and then transform this to $\nabla \log p(x)$, which is then used as a drift (which is possible if $f$ is an invertible function). This is an interesting direction to think about.
>
> ## Synthetic and MNIST experiments
> Thank you for the great suggestion. We are working on a 2D experiment to add to our camera-ready version.
>
> ## Minor
> Yes, we meant the convex hull and we have clarified this in the paper. Thank you very much for catching the other typos as well; they have been corrected.
>
>
> ## Summary
>
> We want to warmly thank you for your rating and your great questions! Adding a 2D experiment will be a significant improvement, and your observation about the broader class of applicable transformations is a very pleasing and interesting insight that we will continue to reflect on.

---

### Decision · Program_Chairs · 2025-09-17

**Decision:**

Accept (poster)

**Comment:**

This work provides evidence for the conjecture that the success of diffusion models is related to the manifold hypothesis. The authors' theoretical and empirical results show that smoothing the score function produces smoothing tangential to the data manifold. Diffusion models are not well understood theoretically, and this paper provides a nice contribution in this direction.

All of the reviewers' concerns were addressed to their satisfaction and they all recommend the paper be accepted.